# Pareto-frontier Entropy Search with Variational Lower Bound Maximization

**Masanori Ishikura** [1]    **Masayuki Karasuyama** [1]

## Abstract

This study considers multi-objective Bayesian optimization (MOBO) through the information gain of the Pareto-frontier. To calculate the information gain, a predictive distribution conditioned on the Pareto-frontier plays a key role, which is defined as a distribution truncated by the Pareto-frontier. However, it is usually impossible to obtain the entire Pareto-frontier in a continuous domain, and therefore, the complete truncation cannot be known. We consider an approximation of the truncated distribution by using a mixture distribution consisting of two possible approximate truncations obtainable from a subset of the Pareto-frontier, which we call over- and under-truncation. Since the optimal balance of the mixture is unknown beforehand, we propose optimizing the balancing coefficient through the variational lower bound maximization framework, by which the approximation error of the information gain can be minimized. Our empirical evaluation demonstrates the effectiveness of the proposed method particularly when the number of objective functions is large.

## 1. Introduction

Multi-objective optimization (MOO) of black-box functions is ubiquitous in a variety of fields such as materials science, engineering, drug design, and AutoML. Evolutionary algorithms have been classically studied for MOO, but they require a large number of function evaluations, which is often difficult for practical problems. On the other hand, Bayesian optimization (BO) based approaches to MOO, which use a probabilistic surrogate model (typically, Gaussian process), have been widely studied recently (e.g., Knowles, 2006; Emmerich, 2005; Ponweiser et al., 2008; Belakaria et al., 2019; Suzuki et al., 2020; Qing et al., 2022; Tu et al., 2022).

This study focuses on multi-objective BO (MOBO) based on the information gain of the Pareto-frontier. Since the optimal solution of an MOO problem is not unique in general, the optimal values are represented as a set of output vectors, called the Pareto-frontier $\mathcal{F}^*$. Pareto-frontier entropy search (PFES) (Suzuki et al., 2020) considers the mutual information between the Pareto-frontier and a candidate point as an acquisition function of MOBO. The effectiveness of the basic idea of PFES has been repeatedly shown (Qing et al., 2022; Tu et al., 2022).

In the information-theoretic approaches considering the information of the Pareto-frontier (Suzuki et al., 2020; Qing et al., 2022; Tu et al., 2022), the predictive distribution given the Pareto-frontier $p(\boldsymbol{f}(\boldsymbol{x}) \mid \mathcal{F}^*)$ plays a key role in the information evaluation, where $\boldsymbol{f}(\boldsymbol{x})$ is a vector of objective function values at the input $\boldsymbol{x}$. In this distribution, $\boldsymbol{f}(\boldsymbol{x})$ cannot be better than $\mathcal{F}^*$ because $\mathcal{F}^*$ should be the Pareto-frontier. Therefore, $p(\boldsymbol{f}(\boldsymbol{x}) \mid \mathcal{F}^*)$ becomes a truncated distribution (see Fig. 1(a) for which details will be discussed in Section 3.2). However, it is practically impossible to obtain the entire $\mathcal{F}^*$ in a continuous space, and we only obtain a finite size subset $\mathcal{F}_S^* \subseteq \mathcal{F}^*$ (red stars in Fig. 1(a)). This means that the exact truncation by $\mathcal{F}^*$ shown in Fig. 1(a) cannot be calculated. To avoid this issue, all the existing studies use approximations based on an overly truncated distribution created by $\mathcal{F}_S^*$ (Fig. 1(d)). A drawback of this approach is that the effect of truncation by $\mathcal{F}^*$ on $\boldsymbol{f}(\boldsymbol{x})$ is always estimated stronger than the true truncation, because of which we call it over-truncation.

In this study, we introduce the variational lower bound maximization approach into the mutual information (MI) estimation of information-theoretic MOBO, which is called Pareto-frontier Entropy search with Variational lower bound maximization (PFEV). In addition to the conventional over-truncation, we also consider a conservative approach, called under-truncation (shown as Fig. 1(c)). The under-truncation estimates the effect of truncation by $\mathcal{F}^*$ weaker than the true truncation. Therefore, to balance two opposite approaches, we combine the over and the under-

[1]Department of Computer Science, Nagoya Institute of Technology, Aichi, Japan. Correspondence to: Masayuki Karasuyama <karasuyama@nitech.ac.jp>.

*Proceedings of the $42^{nd}$ International Conference on Machine Learning*, Vancouver, Canada. PMLR 267, 2025. Copyright 2025 by the author(s).

truncation in such a way that a mixture of them (Fig. 1(e)) defines a variational distribution of the MI approximation. We show that the mixture weight can be optimized through the variational lower bound maximization. This means that the optimal balance of the over and the under-truncation can be determined so that the MI approximation error is minimized.

Our contributions are summarized as follows:

- PFEV is the first approach to continuous space MOBO that is based on a general lower bound of MI (existing work only shows a lower bound under a restrictive condition of two objective problems, for which we discuss in Section 4).

- We newly introduce an under-truncation approximation for $p(\boldsymbol{f}(\boldsymbol{x}) \mid \mathcal{F}^*)$. Further, we define variational distribution as a mixture of distributions with the over and the under-truncation, and show how to optimize the mixture weight.

- We also discuss properties and extensions of PFEV. For example, we show that our MI lower bound can be further lower bounded by PI (probability of improvement). We also discuss a Monte-Carlo approximation tailored to our lower bound. We further discuss several extended settings such as parallel querying.

- Through empirical evaluation on Gaussian process generated functions, benchmark functions, and an application to machine learning hyper-parameter optimization, we demonstrate effectiveness of PFEV. We empirically observed that PFEV shows a particular difference from existing over-truncation based methods when output dimension $\geq 3$, in which the difference of two truncation becomes more apparent.

## 2. Multi-Objective Optimization and Gaussian Process Model

We consider Bayesian optimization (BO) for a multi-objective optimization (MOO) problem of maximizing $L \geq 2$ objective functions $f^l : \mathcal{X} \to \mathbb{R}$ ($l = 1, \ldots, L$), where $\mathcal{X} \subseteq \mathbb{R}^d$ is an input space. Let $\boldsymbol{f}_{\boldsymbol{x}} := (f_{\boldsymbol{x}}^1, \ldots, f_{\boldsymbol{x}}^L)^\top$, where $f_{\boldsymbol{x}}^l := f^l(\boldsymbol{x})$. The optimal solutions of MOO are characterized by the Pareto optimality. For given $\boldsymbol{f}_{\boldsymbol{x}}$ and $\boldsymbol{f}_{\boldsymbol{x}'}$, if $f_{\boldsymbol{x}}^l \geq f_{\boldsymbol{x}'}^l$ for $\forall l \in \{1, \ldots, L\}$ and there exists $l$ that satisfies $f_{\boldsymbol{x}}^l > f_{\boldsymbol{x}'}^l$, then "$\boldsymbol{f}_{\boldsymbol{x}}$ dominates $\boldsymbol{f}_{\boldsymbol{x}'}$", written as $\boldsymbol{f}_{\boldsymbol{x}} \succ \boldsymbol{f}_{\boldsymbol{x}'}$. When $\boldsymbol{f}_{\boldsymbol{x}}$ is not dominated by any other $\boldsymbol{f}_{\boldsymbol{x}'}$, then $\boldsymbol{f}_{\boldsymbol{x}}$ is Pareto-optimal. The Pareto-frontier $\mathcal{F}^*$ is a set of Pareto-optimal $\boldsymbol{f}_{\boldsymbol{x}}$ that can be defined as $\mathcal{F}^* := \{\boldsymbol{f}_{\boldsymbol{x}} \in \mathcal{F}_{\mathcal{X}} \mid \boldsymbol{f}_{\boldsymbol{x}'} \not\succ \boldsymbol{f}_{\boldsymbol{x}}, \forall \boldsymbol{f}_{\boldsymbol{x}'} \in \mathcal{F}_{\mathcal{X}}\}$, where $\mathcal{F}_{\mathcal{X}} := \{\boldsymbol{f}_{\boldsymbol{x}} \in \mathbb{R}^L \mid \forall \boldsymbol{x} \in \mathcal{X}\}$.

Each objective function is represented by the Gaussian process (GP) regression. The observation of the $l$-th objective function is $y_i^l = f_{\boldsymbol{x}_i}^l + \varepsilon$, where $\varepsilon \sim \mathcal{N}(0, \sigma_{\mathrm{noise}}^2)$. The training dataset with $n$ observations is written as $\mathcal{D} := \{(\boldsymbol{x}_i, \boldsymbol{y}_i)\}_{i=1}^n$, where $\boldsymbol{y}_i = (y_i^1, \ldots, y_i^L)^\top$. We use independent $L$ GPs with a kernel function $k(\boldsymbol{x}, \boldsymbol{x}')$. Let $\boldsymbol{k}(\boldsymbol{x}) := (k(\boldsymbol{x}, \boldsymbol{x}_1), \ldots, k(\boldsymbol{x}, \boldsymbol{x}_n))^\top$, $\boldsymbol{y}^l := (y_1^l, \ldots, y_n^l)^\top$, and $\boldsymbol{K}$ be the matrix whose $(i, j)$-element is $k(\boldsymbol{x}_i, \boldsymbol{x}_j)$. The posterior $p(f_{\boldsymbol{x}}^l \mid \mathcal{D})$ of the $l$-th GP (with 0 prior mean) is written as $\mathcal{N}(\mu_l(\boldsymbol{x}), \sigma_l^2(\boldsymbol{x}))$, where $\mu_l(\boldsymbol{x}) = \boldsymbol{k}(\boldsymbol{x})^\top (\boldsymbol{K} + \sigma_{\mathrm{noise}}^2 \boldsymbol{I})^{-1} \boldsymbol{y}^l$ and $\sigma_l^2(\boldsymbol{x}) = k(\boldsymbol{x}, \boldsymbol{x}) - \boldsymbol{k}(\boldsymbol{x})^\top (\boldsymbol{K} + \sigma_{\mathrm{noise}}^2 \boldsymbol{I})^{-1} \boldsymbol{k}(\boldsymbol{x})$. From independence, we have $p(\boldsymbol{f}_{\boldsymbol{x}} \mid \mathcal{D}) = \prod_{l \in [L]} p(f_{\boldsymbol{x}}^l \mid \mathcal{D})$. For notational brevity, conditioning on $\mathcal{D}$ is omitted (e.g., $p(\boldsymbol{f}_{\boldsymbol{x}} \mid \mathcal{D})$ is written as $p(\boldsymbol{f}_{\boldsymbol{x}})$).

## 3. Pareto-frontier Entropy Search with Variational Lower Bound Maximization

We consider multi-objective Bayesian optimization (MOBO) based on mutual information $\mathrm{MI}(\boldsymbol{f}_{\boldsymbol{x}}; \mathcal{F}^*)$ between an objective function value $\boldsymbol{f}_{\boldsymbol{x}}$ and the Pareto-frontier $\mathcal{F}^*$ (Note that, throughout the paper, $\mathcal{F}^*$ is a random variable determined via the predictive distribution of $\boldsymbol{f}_{\boldsymbol{x}}$). An intuition behind this criterion is to select $\boldsymbol{x}$ that provides the maximum information gain of the Pareto-frontier $\mathcal{F}^*$. The effectiveness of this approach is shown by (Suzuki et al., 2020), but it is known that accurate evaluation of $\mathrm{MI}(\boldsymbol{f}_{\boldsymbol{x}}; \mathcal{F}^*)$ is difficult. Our proposed method is the first method introducing the variational lower bound maximization to evaluate $\mathrm{MI}(\boldsymbol{f}_{\boldsymbol{x}}; \mathcal{F}^*)$. We call our proposed method Pareto-frontier Entropy search with Variational lower bound maximization (PFEV).

### 3.1. Lower Bound of Mutual Information

A lower bound $\mathrm{LB}(\boldsymbol{x})$ of $\mathrm{MI}(\boldsymbol{f}_{\boldsymbol{x}}; \mathcal{F}^*)$ can be derived as

$$
\begin{aligned}
&\mathrm{MI}(\boldsymbol{f}_{\boldsymbol{x}}; \mathcal{F}^*) \\
&= \int p(\mathcal{F}^*) \int p(\boldsymbol{f}_{\boldsymbol{x}} \mid \mathcal{F}^*) \log \frac{p(\boldsymbol{f}_{\boldsymbol{x}} \mid \mathcal{F}^*)}{p(\boldsymbol{f}_{\boldsymbol{x}})} \mathrm{d}\boldsymbol{f}_{\boldsymbol{x}} \mathrm{d}\mathcal{F}^* \\
&= \int p(\mathcal{F}^*) \left[ \int p(\boldsymbol{f}_{\boldsymbol{x}} \mid \mathcal{F}^*) \log \frac{q(\boldsymbol{f}_{\boldsymbol{x}} \mid \mathcal{F}^*)}{p(\boldsymbol{f}_{\boldsymbol{x}})} \mathrm{d}\boldsymbol{f}_{\boldsymbol{x}} \right. \\
&\qquad \left. + D_{\mathrm{KL}} \left( p(\boldsymbol{f}_{\boldsymbol{x}} \mid \mathcal{F}^*) \,\|\, q(\boldsymbol{f}_{\boldsymbol{x}} \mid \mathcal{F}^*) \right) \right] \mathrm{d}\mathcal{F}^* \\
&\geq \mathbb{E}_{\mathcal{F}^*} \left[ \int p(\boldsymbol{f}_{\boldsymbol{x}} \mid \mathcal{F}^*) \log \frac{q(\boldsymbol{f}_{\boldsymbol{x}} \mid \mathcal{F}^*)}{p(\boldsymbol{f}_{\boldsymbol{x}})} \mathrm{d}\boldsymbol{f}_{\boldsymbol{x}} \right] \\
&= \mathbb{E}_{\mathcal{F}^*, \boldsymbol{f}_{\boldsymbol{x}}} \left[ \log \frac{q(\boldsymbol{f}_{\boldsymbol{x}} \mid \mathcal{F}^*)}{p(\boldsymbol{f}_{\boldsymbol{x}})} \right] =: \mathrm{LB}(\boldsymbol{x}), \quad (1)
\end{aligned}
$$

where $D_{\mathrm{KL}}$ is Kullback-Leibler (KL) divergence, and $q(\boldsymbol{f}_{\boldsymbol{x}} \mid \mathcal{F}^*)$ is a density function called a variational distribution. A similar lower bound was first derived in the context of constrained BO (Takeno et al., 2022). This lower bound holds for any distribution $q(\boldsymbol{f}_{\boldsymbol{x}} \mid \mathcal{F}^*)$ that satisfies

the following support condition:

$$\text{supp}(q(\boldsymbol{f_x} \mid \mathcal{F}^*)) \supseteq \text{supp}(p(\boldsymbol{f_x} \mid \mathcal{F}^*)), \qquad (2)$$

where supp is a set of non-zero points of a given density, defined as $\text{supp}(p(\boldsymbol{f} \mid \mathcal{F}^*)) = \{\boldsymbol{f} \in \mathbb{R}^L \mid p(\boldsymbol{f} \mid \mathcal{F}^*) \neq 0\}$. In addition to this condition, the lower bound $\text{LB}(\boldsymbol{x})$ is derived by using the convention $0 \log 0 = 0$ (Cover & Thomas, 2006). We call the condition (2) variational distribution condition (VDC). Note that if VDC is not satisfied, $\text{LB}(\boldsymbol{x})$ is not defined, because then, the probability measure induced by $p(\boldsymbol{f_x} \mid \mathcal{F}^*)$ is not absolutely continuous with respect to those of $q(\boldsymbol{f_x} \mid \mathcal{F}^*)$.

### 3.2. Variational Lower Bound Maximization by Combining Over- and Under-Truncation

If $q(\boldsymbol{f_x} \mid \mathcal{F}^*) = p(\boldsymbol{f_x} \mid \mathcal{F}^*)$ for $\mathcal{F}^*$ almost everywhere, the lower bound $\text{LB}(\boldsymbol{x})$ is equal to $\text{MI}(\boldsymbol{f_x}; \mathcal{F}^*)$. However, the analytical representation of $p(\boldsymbol{f_x} \mid \mathcal{F}^*)$ is not known. This stems from a common well-known difficulty of information-theoretic BO, and effectiveness of the truncated distribution based approximation has been repeatedly shown (e.g., Wang & Jegelka, 2017; Takeno et al., 2020; Perrone et al., 2019; Suzuki et al., 2020) not only for MOBO, but also for a variety contexts of BO problems (such as standard single-objective problems, constraint problems, and multi-fidelity problems).

Define $\boldsymbol{f} \preceq \mathcal{F}^*$ as that $\boldsymbol{f}$ is dominated by or equal to at least one element in $\mathcal{F}^*$. The truncation-based approximation replaces the conditioning in $p(\boldsymbol{f_x} \mid \mathcal{F}^*)$ with $\boldsymbol{f_x} \preceq \mathcal{F}^*$, by which we obtain $q(\boldsymbol{f_x} \mid \mathcal{F}^*) = p(\boldsymbol{f_x} \mid \boldsymbol{f_x} \preceq \mathcal{F}^*)$:

$$q(\boldsymbol{f_x} \mid \mathcal{F}^*) = \begin{cases} p(\boldsymbol{f_x})/Z^{\mathcal{F}^*}(\boldsymbol{x}) & \text{if } \boldsymbol{f_x} \in \mathcal{A}^{\mathcal{F}^*}, \\ 0 & \text{otherwise,} \end{cases}$$

where $\mathcal{A}^{\mathcal{F}^*} := \{\boldsymbol{f} \in \mathbb{R}^L \mid \boldsymbol{f} \preceq \boldsymbol{f}', \exists \boldsymbol{f}' \in \mathcal{F}^*\}$ and $Z^{\mathcal{F}^*}(\boldsymbol{x}) := p(\boldsymbol{f_x} \in \mathcal{A}^{\mathcal{F}^*})$ is the normalization constant. As shown in Fig. 1(a), $q(\boldsymbol{f_x} \mid \mathcal{F}^*)$ is a truncated normal distribution in which only the region dominated by $\mathcal{F}^*$ remains. Suzuki et al. (2020) call this distribution PFTN (Pareto-Frontier Truncated Normal distribution). PFTN is derived by the fact that if $\mathcal{F}^*$ is given, any $\boldsymbol{f_x}$ dominating $\mathcal{F}^*$ cannot exist. However, $\mathcal{F}^*$ cannot be obtained in practice in the continuous domain, and we only obtain limited discrete points, such as those indicated by the red star points in Fig. 1(a). A subset of $\mathcal{F}^*$ defined by these discrete points is written as $\mathcal{F}_S^* \subseteq \mathcal{F}^*$ (we discuss how to calculate $\mathcal{F}_S^*$ in Section 3.3). In this study, we consider combining two truncated distributions derived from $\mathcal{F}_S^*$, instead of using $\mathcal{F}^*$ that is not obtainable.

In Fig. 1(b), the orange dashed lines are examples of possible truncation behind $\mathcal{F}_S^*$ and the red dot line is the true $\mathcal{F}^*$, which is unknown. The first truncated distribution is based

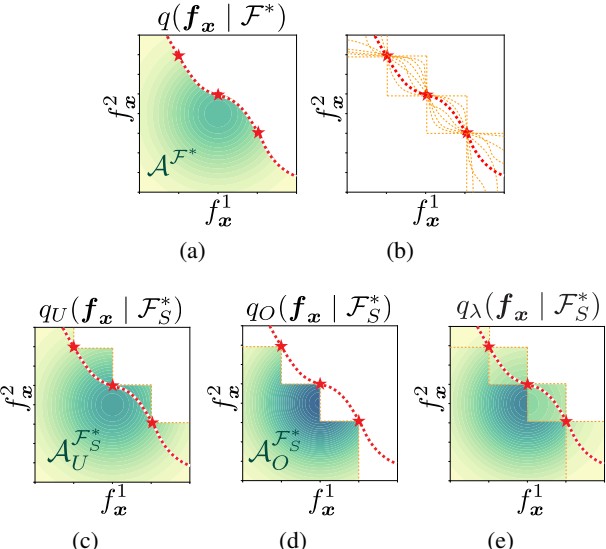

*Figure 1.* (a) Predictive distribution truncated by $\mathcal{F}^*$. (b) Examples of possible truncation given $\mathcal{F}_S^*$. (c) Under-truncation based on $\tilde{\mathcal{F}}_S^*$. (d) Over-truncation based on $\tilde{\mathcal{F}}_S^*$. (e) Mixture of $q_U(\boldsymbol{f_x} \mid \tilde{\mathcal{F}}_S^*)$ and $q_O(\boldsymbol{f_x} \mid \tilde{\mathcal{F}}_S^*)$.

on the most conservative truncation shown in Fig. 1(c). This truncation is defined by removing "the region that dominates $\mathcal{F}_S^*$" (any point that dominates $\mathcal{F}_S^*$ cannot exist). The remaining region is written as $\mathcal{A}_U^{\mathcal{F}_S^*} := \mathbb{R}^L \setminus \{\boldsymbol{f} \in \mathbb{R}^L \mid \boldsymbol{f}' \preceq \boldsymbol{f}, \exists \boldsymbol{f}' \in \mathcal{F}_S^*\}$. The resulting truncated normal distribution is

$$q_U(\boldsymbol{f_x} \mid \mathcal{F}_S^*) := \begin{cases} p(\boldsymbol{f_x})/Z_U^{\mathcal{F}_S^*}(\boldsymbol{x}) & \text{if } \boldsymbol{f_x} \in \mathcal{A}_U^{\mathcal{F}_S^*}, \\ 0 & \text{otherwise,} \end{cases}$$

where $Z_U^{\mathcal{F}_S^*}(\boldsymbol{x}) := p(\boldsymbol{f_x} \in \mathcal{A}_U^{\mathcal{F}_S^*})$. We call this distribution PFTN-U (PFTN with under-truncation). PFTN-U has a larger support $\text{supp}(q_U(\boldsymbol{f_x} \mid \mathcal{F}_S^*)) \supseteq \text{supp}(p(\boldsymbol{f_x} \mid \mathcal{F}^*))$, and thus, VDC is satisfied. This truncation is conservative in the sense that the region where the density function becomes zero is smaller than $q(\boldsymbol{f_x} \mid \mathcal{F}^*)$, by which it underestimates the effect of the condition $\boldsymbol{f_x} \preceq \mathcal{F}^*$ on $\boldsymbol{f_x}$.

The second truncated distribution is based on the over-truncation shown in Fig 1(d). This is "the region dominated by $\mathcal{F}_S^*$", defined as $\mathcal{A}_O^{\mathcal{F}_S^*} := \{\boldsymbol{f} \in \mathbb{R}^L \mid \boldsymbol{f} \preceq \boldsymbol{f}', \exists \boldsymbol{f}' \in \mathcal{F}_S^*\}$. The resulting truncated normal distribution is

$$q_O(\boldsymbol{f_x} \mid \mathcal{F}_S^*) := \begin{cases} p(\boldsymbol{f_x})/Z_O^{\mathcal{F}_S^*}(\boldsymbol{x}) & \text{if } \boldsymbol{f_x} \in \mathcal{A}_O^{\mathcal{F}_S^*}, \\ 0 & \text{otherwise,} \end{cases}$$

where $Z_O^{\mathcal{F}_S^*}(\boldsymbol{x}) := p(\boldsymbol{f_x} \in \mathcal{A}_O^{\mathcal{F}_S^*})$. In contrast to $q_U$, $q_O$ overly truncates the distribution in the sense that the region where the density function value becomes zero is larger than $q(\boldsymbol{f_x} \mid \mathcal{F}^*)$, by which it overestimates the effect of

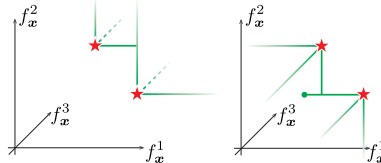

*Figure 2.* An example of truncation in three-dimensional output space. The resulting truncate regions have a large difference.

the condition $\boldsymbol{f_x} \preceq \mathcal{F}^*$ on $\boldsymbol{f_x}$. We call this distribution PFTN-O (PFTN with over-truncation). It is important to note that PFTN-O may not satisfy VDC (2) because $\mathrm{supp}(q_O(\boldsymbol{f_x} \mid \mathcal{F}_S^*)) \subseteq \mathrm{supp}(p(\boldsymbol{f_x} \mid \mathcal{F}^*))$.

As we have already discussed, $q_O$ and $q_U$ over and under estimates the truncation by the true $\mathcal{F}^*$. Therefore, instead of using one of them, we consider the mixture defined as follows:

$$
q_\lambda(\boldsymbol{f_x} \mid \mathcal{F}_S^*) = \lambda q_U(\boldsymbol{f_x} \mid \mathcal{F}_S^*) + (1-\lambda)q_O(\boldsymbol{f_x} \mid \mathcal{F}_S^*),
$$

$$
= \begin{cases} \left(\frac{\lambda}{Z_U^{\mathcal{F}_S^*}(\boldsymbol{x})} + \frac{1-\lambda}{Z_O^{\mathcal{F}_S^*}(\boldsymbol{x})}\right) p(\boldsymbol{f_x}) & \text{if } \boldsymbol{f_x} \in \mathcal{A}_O^{\mathcal{F}_S^*}, \\ \lambda p(\boldsymbol{f_x})/Z_U^{\mathcal{F}_S^*}(\boldsymbol{x}) & \text{if } \boldsymbol{f_x} \in \mathcal{A}_{U\setminus O}^{\mathcal{F}_S^*}, \\ 0 & \text{otherwise,} \end{cases} \quad (3)
$$

where $\mathcal{A}_{U\setminus O}^{\mathcal{F}_S^*} := \mathcal{A}_U^{\mathcal{F}_S^*} \setminus \mathcal{A}_O^{\mathcal{F}_S^*}$, and $\lambda \in (0, 1]$ is a weight of the mixture. Because of $\lambda \neq 0$, we have $\mathrm{supp}(q_\lambda(\boldsymbol{f_x} \mid \mathcal{F}_S^*)) \supseteq \mathrm{supp}(p(\boldsymbol{f_x} \mid \mathcal{F}^*))$, meaning that the mixture satisfies VDC. Figure 1(e) shows an example of the mixture. The effect of PFTN-U and PFTN-O can be controlled by $\lambda$. In the 2D illustration in Fig. 1, the difference between the two truncations might appear small. However, as shown in Fig. 2, in three or more dimensions, the two truncations are significantly different, obviously (further discussion is in Appendix L). Therefore, we conjecture that the balance of them can have a strong effect, particularly for problems with $L \geq 3$.

By substituting $q_\lambda(\boldsymbol{f_x} \mid \mathcal{F}_S^*)$ into $\mathrm{LB}(\boldsymbol{x})$, we define $\mathrm{LB}(\boldsymbol{x}, \lambda)$ as

$$
\mathrm{LB}(\boldsymbol{x}, \lambda)
$$
$$
= \mathbb{E}_{\mathcal{F}^*, \boldsymbol{f_x}}\left[\log\left(\frac{\lambda q_U(\boldsymbol{f_x} \mid \mathcal{F}_S^*) + (1-\lambda)q_O(\boldsymbol{f_x} \mid \mathcal{F}_S^*)}{p(\boldsymbol{f_x})}\right)\right]
$$
$$
= \mathbb{E}_{\mathcal{F}^*, \boldsymbol{f_x}}\big[\log\big\{\zeta_\lambda^{\mathcal{F}_S^*}(\boldsymbol{x})\mathbb{I}(\boldsymbol{f_x} \in \mathcal{A}_O^{\mathcal{F}_S^*})
$$
$$
+ \eta_\lambda^{\mathcal{F}_S^*}(\boldsymbol{x})\mathbb{I}(\boldsymbol{f_x} \in \mathcal{A}_{U\setminus O}^{\mathcal{F}_S^*})\}\big], \quad (4)
$$

where $\zeta_\lambda^{\mathcal{F}_S^*}(\boldsymbol{x}) = \frac{\lambda}{Z_U^{\mathcal{F}_S^*}(\boldsymbol{x})} + \frac{1-\lambda}{Z_O^{\mathcal{F}_S^*}(\boldsymbol{x})}$, $\eta_\lambda^{\mathcal{F}_S^*}(\boldsymbol{x}) = \frac{\lambda}{Z_U^{\mathcal{F}_S^*}(\boldsymbol{x})}$, and $\mathbb{I}$ is the indicator function. Importantly, the weight parameter $\lambda$ can be estimated by maximizing the lower

bound:

$$
\max_{\lambda \in (0,1]} \mathrm{LB}(\boldsymbol{x}, \lambda) \quad (5)
$$

This maximization implies the following property, which is a well-known advantage of the variational lower bound maximization (Bishop & Bishop, 2023):

**Remark 3.1.** *From (1),*

$$
\mathrm{MI}(\boldsymbol{f_x}; \mathcal{F}^*) = \mathrm{LB}(\boldsymbol{x}, \lambda)
$$
$$
+ \mathbb{E}_{\mathcal{F}^*}\left[D_{\mathrm{KL}}\left(p(\boldsymbol{f_x} \mid \mathcal{F}^*) \parallel q_\lambda(\boldsymbol{f_x} \mid \mathcal{F}^*)\right)\right].
$$

*Therefore, (5) is equivalent to*

$$
\min_{\lambda \in (0,1]} \mathbb{E}_{\mathcal{F}^*}\left[D_{\mathrm{KL}}\left(p(\boldsymbol{f_x} \mid \mathcal{F}^*) \parallel q_\lambda(\boldsymbol{f_x} \mid \mathcal{F}^*)\right)\right].
$$

*As a result, (5) can be seen as the minimization of difference between true $p(\boldsymbol{f_x} \mid \mathcal{F}^*)$ and $q_\lambda$.*

Further, we have the following property:

**Remark 3.2.** *(5) can be bounded from below (the proof is in Appendix A):*

$$
\max_{\lambda \in (0,1]} \mathrm{LB}(\boldsymbol{x}, \lambda) \geq \mathbb{E}_{\mathcal{F}^*}\left[p(\boldsymbol{f_x} \notin \mathcal{A}_U^{\mathcal{F}_S^*})\right] > 0.
$$

$p(\boldsymbol{f_x} \notin \mathcal{A}_U^{\mathcal{F}_S^*})$ *can be seen as the probability of improvement (PI) from the region $\mathcal{A}_U^{\mathcal{F}_S^*}$, from which positivity of (5) is also directly derived.*

While MI has a trivial lower bound $0$ in general, this remark guarantees that (5) is a larger lower bound than this trivial bound. Further, we can also see that if the candidate $\boldsymbol{x}$ is promising in a sense of PI, (5) should have substantially larger value than $0$.

As a result, the selection of $\boldsymbol{x}$ is formulated as

$$
\max_{\boldsymbol{x}, \lambda \in (0,1]} \mathrm{LB}(\boldsymbol{x}, \lambda),
$$

in which $\boldsymbol{x}$ and $\lambda$ can be simultaneously optimized ($(d+1)$-dimensional maximization).

### 3.3. Computations

We employ the Monte-Carlo (MC) estimation to calculate the expectation in $\mathrm{LB}(\boldsymbol{x}, \lambda)$:

$$
\mathrm{LB}(\boldsymbol{x}, \lambda) \approx \frac{1}{K} \sum_{(\tilde{\mathcal{F}}_S^*, \tilde{\boldsymbol{f}}) \in F} \log\left\{\zeta_\lambda^{\tilde{\mathcal{F}}_S^*}(\boldsymbol{x})\mathbb{I}(\tilde{\boldsymbol{f}_x} \in \mathcal{A}_O^{\tilde{\mathcal{F}}_S^*})\right.
$$
$$
\left. + \eta_\lambda^{\tilde{\mathcal{F}}_S^*}(\boldsymbol{x})\mathbb{I}(\tilde{\boldsymbol{f}_x} \in \mathcal{A}_{U\setminus O}^{\tilde{\mathcal{F}}_S^*})\right\}, \quad (6)
$$

where $F$ is a set of sampled pairs of $(\mathcal{F}_S^*, \boldsymbol{f})$, for which a sample pair is denoted as $(\tilde{\mathcal{F}}_S^*, \tilde{\boldsymbol{f}})$, and $K = |F|$ is the

number of samples. Henceforth, variables with ' ~ ' indicate sampled values. For $\mathcal{F}_S^*$, the same sampling strategy can be used as existing information-theoretic MOBO methods (Suzuki et al., 2020; Hernandez-Lobato et al., 2016), in which a sample path $\tilde{f}^l$ is approximately generated by using random feature map (RFM) (Rahimi & Recht, 2008). We obtain $\mathcal{F}_S^*$ by solving the MOO maximizing sampled $\tilde{f}^l(\boldsymbol{x})$ for $l = 1, \ldots, L$. This maximization can be performed by general MOO solvers such as the well-known NSGA-II (Deb et al., 2002). In the case of NSGA-II, $|\mathcal{F}_S^*|$ can be specified, typically less than 100. Computations of $Z_O^{\tilde{\mathcal{F}}_S^*}$ and $Z_U^{\tilde{\mathcal{F}}_S^*}$ can be easily performed by the cell (hyper-rectangle) decomposition-based approach as shown by (Suzuki et al., 2020) for which details are in Appendix B. This decomposition is only required once for each iteration of BO because it is common for all candidate $\boldsymbol{x}$.

We also consider another way to numerically approximate $\text{LB}(\boldsymbol{x}, \lambda)$ based on the following transformation:

$$
\begin{aligned}
&\text{LB}(\boldsymbol{x}, \lambda) \\
&= \mathbb{E}_{\mathcal{F}^*, \boldsymbol{f}_{\boldsymbol{x}}} \Big[ \mathbb{I}(\boldsymbol{f}_{\boldsymbol{x}} \in \mathcal{A}_O^{\mathcal{F}_S^*}) \log \zeta_\lambda^{\tilde{\mathcal{F}}_S^*}(\boldsymbol{x}) + \mathbb{I}(\boldsymbol{f}_{\boldsymbol{x}} \in \mathcal{A}_{U \setminus O}^{\mathcal{F}_S^*}) \log \eta_\lambda^{\tilde{\mathcal{F}}_S^*}(\boldsymbol{x}) \Big] \\
&= \mathbb{E}_{\mathcal{F}^*} \Big[ p(\boldsymbol{f}_{\boldsymbol{x}} \in \mathcal{A}_O^{\mathcal{F}_S^*} \mid \mathcal{F}^*) \log \zeta_\lambda^{\tilde{\mathcal{F}}_S^*}(\boldsymbol{x}) \\
&\qquad\qquad + p(\boldsymbol{f}_{\boldsymbol{x}} \in \mathcal{A}_{U \setminus O}^{\mathcal{F}_S^*} \mid \mathcal{F}^*) \log \eta_\lambda^{\tilde{\mathcal{F}}_S^*}(\boldsymbol{x}) \Big] \\
&\approx \frac{1}{K} \sum_{(\tilde{\mathcal{F}}_S^*, \tilde{\boldsymbol{f}}) \in F} p(\boldsymbol{f}_{\boldsymbol{x}} \in \mathcal{A}_O^{\tilde{\mathcal{F}}_S^*} \mid \tilde{\mathcal{F}}^*) \log \zeta_\lambda^{\tilde{\mathcal{F}}_S^*}(\boldsymbol{x}) \\
&\qquad\qquad + p(\boldsymbol{f}_{\boldsymbol{x}} \in \mathcal{A}_{U \setminus O}^{\tilde{\mathcal{F}}_S^*} \mid \tilde{\mathcal{F}}^*) \log \eta_\lambda^{\tilde{\mathcal{F}}_S^*}(\boldsymbol{x}).
\end{aligned} \tag{7}
$$

Note that the second line is from $\mathbb{I}(\boldsymbol{f}_{\boldsymbol{x}} \in \mathcal{A}_O^{\mathcal{F}_S^*}) = 1 - \mathbb{I}(\boldsymbol{f}_{\boldsymbol{x}} \in \mathcal{A}_{U \setminus O}^{\mathcal{F}_S^*})$, and the third line is obtained by replacing $\mathbb{E}_{\mathcal{F}^*, \boldsymbol{f}_{\boldsymbol{x}}}$ with $\mathbb{E}_{\mathcal{F}^*} \mathbb{E}_{\boldsymbol{f}_{\boldsymbol{x}} \mid \mathcal{F}^*}$. Since (6) can be rewritten as $\frac{1}{K} \sum_{(\tilde{\mathcal{F}}_S^*, \tilde{\boldsymbol{f}}) \in F} \mathbb{I}(\tilde{\boldsymbol{f}}_{\boldsymbol{x}} \in \mathcal{A}_O^{\tilde{\mathcal{F}}_S^*}) \log \zeta_\lambda^{\tilde{\mathcal{F}}_S^*}(\boldsymbol{x}) + \mathbb{I}(\tilde{\boldsymbol{f}}_{\boldsymbol{x}} \in \mathcal{A}_{U \setminus O}^{\tilde{\mathcal{F}}_S^*}) \log \eta_\lambda^{\tilde{\mathcal{F}}_S^*}(\boldsymbol{x})$, by comparing this re-written expression with (7), we can interpret that (6) performs one-sample approximations $p(\boldsymbol{f}_{\boldsymbol{x}} \in \mathcal{A}_O^{\tilde{\mathcal{F}}_S^*} \mid \tilde{\mathcal{F}}^*) \approx \mathbb{I}(\tilde{\boldsymbol{f}}_{\boldsymbol{x}} \in \mathcal{A}_O^{\tilde{\mathcal{F}}_S^*})$ and $p(\boldsymbol{f}_{\boldsymbol{x}} \in \mathcal{A}_{U \setminus O}^{\tilde{\mathcal{F}}_S^*} \mid \tilde{\mathcal{F}}^*) \approx \mathbb{I}(\tilde{\boldsymbol{f}}_{\boldsymbol{x}} \in \mathcal{A}_{U \setminus O}^{\tilde{\mathcal{F}}_S^*})$. We consider improving this approximations by introducing prior knowledge about $p(\boldsymbol{f}_{\boldsymbol{x}} \in \mathcal{A}_O^{\tilde{\mathcal{F}}_S^*} \mid \tilde{\mathcal{F}}^*)$ and $p(\boldsymbol{f}_{\boldsymbol{x}} \in \mathcal{A}_{U \setminus O}^{\tilde{\mathcal{F}}_S^*} \mid \tilde{\mathcal{F}}^*)$.

Let $\theta := p(\boldsymbol{f}_{\boldsymbol{x}} \in \mathcal{A}_O^{\tilde{\mathcal{F}}_S^*} \mid \tilde{\mathcal{F}}^*)$. We use an approximation $\theta \approx p(\boldsymbol{f}_{\boldsymbol{x}} \in \mathcal{A}_O^{\tilde{\mathcal{F}}_S^*} \mid \boldsymbol{f}_{\boldsymbol{x}} \in \mathcal{A}_U^{\tilde{\mathcal{F}}_S^*}) = Z_O^{\tilde{\mathcal{F}}_S^*}(\boldsymbol{x}) / Z_U^{\tilde{\mathcal{F}}_S^*}(\boldsymbol{x}) =: \hat{p}$ as our prior knowledge. The approximation is replacement of the conditioning by $\tilde{\mathcal{F}}^*$ with the under-truncation $\boldsymbol{f}_{\boldsymbol{x}} \in \mathcal{A}_U^{\tilde{\mathcal{F}}_S^*}$. Note that $Z_O^{\tilde{\mathcal{F}}_S^*}(\boldsymbol{x})$ and $Z_U^{\tilde{\mathcal{F}}_S^*}(\boldsymbol{x})$ are required even in the naïve MC (6), $\hat{p}$ can be obtained without additional computations. A prior distribution having the mode at $\hat{p}$ is introduced to estimate $\theta$. We use the beta distribution by

which MAP (maximum a posteriori) becomes

$$
\theta_{\text{MAP}}(\boldsymbol{f}_{\boldsymbol{x}}) = \frac{\hat{p} + \mathbb{I}(\boldsymbol{f}_{\boldsymbol{x}} \in \mathcal{A}_O^{\tilde{\mathcal{F}}_S^*})}{2}.
$$

The detailed derivation is in Appendix D.1. $\theta_{\text{MAP}}$ can be seen as an average of $\hat{p}$ and $\mathbb{I}(\boldsymbol{f}_{\boldsymbol{x}} \in \mathcal{A}_O^{\tilde{\mathcal{F}}_S^*})$.

As a result, we obtain

$$
\begin{aligned}
\text{LB}(\boldsymbol{x}, \lambda) &\approx \frac{1}{K} \sum_{(\tilde{\mathcal{F}}_S^*, \tilde{\boldsymbol{f}}) \in F} \theta_{\text{MAP}}(\tilde{\boldsymbol{f}}_{\boldsymbol{x}}) \log \zeta_\lambda^{\tilde{\mathcal{F}}_S^*}(\boldsymbol{x}) \\
&+ (1 - \theta_{\text{MAP}}(\tilde{\boldsymbol{f}}_{\boldsymbol{x}})) \log \eta_\lambda^{\tilde{\mathcal{F}}_S^*}(\boldsymbol{x}) =: \widehat{\text{LB}}(\boldsymbol{x}, \lambda).
\end{aligned} \tag{8}
$$

We empirically observe that the estimation variance can be improved by the prior. Instead, the bias to $\hat{p}$ can occur, but because the number of samples is usually small (our default setting is $K = 10$ in the experiments, which is same as existing information-theoretic BO studies such as (Wang & Jegelka, 2017)), variance reduction has a stronger benefit in practice. Further discussion on the accuracy of this estimator is in Appendices D.2 and D.3.

The procedure of PFEV is shown in Algorithm 1. Assume that we already have the posterior mean and variance of the GPs. Then, computations of (8) is $O(KCL)$, where $C$ is the number of hyper-rectangle cells in the decomposition. For the cell decomposition, we employ the quick hypervolume (QHV) algorithm (Russo & Francisco, 2014) as indicated by (Suzuki et al., 2020). Sampling of $F$ requires $O(D^3)$ for RFM with $D$ basis functions and the cost of NSGA-II is also required. We empirically see that, for small $L$, the computational cost of NSGA-II is dominant, and for large $L$, QHV becomes dominant, both of which are commonly required for several information-theoretic MOBO (Suzuki et al., 2020; Qing et al., 2022; Tu et al., 2022). Compared with them, the practical cost of the lower bound calculation in CalcPFEV of Algorithm 1 is often relatively small (see Appendix K.3 for details). As discussed in the end of Section 3.2, the acquisition function maximization is formulated as $(d + 1)$-dimensional optimization (line 10 of Algorithm 1). On the other hand, it is also possible to optimize $\lambda$ for each given $\boldsymbol{x}$ as an inner one-dimensional optimization. In the latter case, efficient computations can be performed by considering (8) (or (6)) is concave with respect to $\lambda$. Further, we can show that the maximizer of $\lambda$ exists though the candidate $\lambda$ is defined as a left open interval $(0, 1]$. Details of these discussions about the maximization problem of $\lambda$ are shown in Appendix C.

# 4. Related Work

For MOBO, a variety of approaches have been proposed, typically by extending single-objective acquisition function such as expected improvement and upper confidence bound

---

**Algorithm 1** Pseudo-code of PFEV

1: **Function** PFEV( Initial dataset $\mathcal{D}_0$ ):
2: **for** $t = 0, \ldots, T$ **do**
3: $\quad F \leftarrow \{\}$
4: $\quad$ **for** $k = 1, \ldots, K$ **do**
5: $\quad\quad$ Generate a sample path $\tilde{\boldsymbol{f}}$ from the current posterior of $\boldsymbol{f}$
6: $\quad\quad$ Obtain $\tilde{\mathcal{F}}_S^*$ by applying NSGA-II to $\tilde{\boldsymbol{f}}_{\boldsymbol{x}}$
7: $\quad\quad$ Create cell-decomposition by QHV
8: $\quad\quad$ $F \leftarrow F \cup (\tilde{\mathcal{F}}_S^*, \tilde{\boldsymbol{f}})$
9: $\quad$ **end for**
10: $\quad (\boldsymbol{x}_{t+1}, \lambda_{t+1}) \leftarrow \text{argmax}_{\boldsymbol{x}, \lambda} \text{ CALCPFEV}(\boldsymbol{x}, \lambda; F)$
11: $\quad$ Evaluate $\boldsymbol{f}_{\boldsymbol{x}_{t+1}}$ and observe $\boldsymbol{y}_{t+1}$
12: $\quad \mathcal{D}_{t+1} \leftarrow \mathcal{D}_t \cup (\boldsymbol{x}_{t+1}, \boldsymbol{y}_{t+1})$
13: **end for**
14: **Function** CALCPFEV( $\boldsymbol{x}, \lambda; F$ ):
15: Calculate $Z_O^{\tilde{F}_S^*}(\boldsymbol{x}), Z_U^{\tilde{F}_S^*}(\boldsymbol{x})$, and $\mathbb{I}(\tilde{\boldsymbol{f}}_{\boldsymbol{x}} \in \mathcal{A}_O^{\tilde{F}_S^*})$ for $(\tilde{\mathcal{F}}_S^*, \tilde{\boldsymbol{f}}) \in F$
16: **return** Approximate lower bound (8)

---

(e.g., Emmerich, 2005; Shah & Ghahramani, 2016; Zuluaga et al., 2016; Daulton et al., 2020; Ament et al., 2023). Information-theoretic approaches (e.g., Hernandez-Lobato et al., 2016; Belakaria et al., 2019) also have been extended from its counterpart of the single-objective BO (Hennig & Schuler, 2012; Hernández-Lobato et al., 2014; Wang & Jegelka, 2017). Among these, the most closely related method to the proposed method is PFES (Suzuki et al., 2020). PFES can be seen as a multi-objective extension of the max value based information-theoretic BO, called max-value entropy search (MES) (Wang & Jegelka, 2017). Information-theoretic MOBO before PFES and other approaches are reviewed in (Suzuki et al., 2020). Here, we mainly focus on information-theoretic MOBO methods after PFES. A more comprehensive related work including other information-theoretic methods and other criteria are discussed in Appendix I.

The MI approximation with PFTN was first introduced by PFES. On the other hand, the MI approximation is based on a decomposition into difference of the entropy, which is a classical approach in information-theoretic BO (Hernández-Lobato et al., 2014). In the context of constrained BO, Takeno et al. (2022) revealed that the entropy difference based decomposition can cause a critical issue originated from a negative value of the MI approximation. The positivity of PFES has not been clarified. Further, PFES only uses over-truncation (PFTN-O). Therefore, effect of the truncation by $\mathcal{F}^*$ on $\boldsymbol{f}_{\boldsymbol{x}}$ is overly estimated.

After PFES, Joint Entropy Search (JES) (Tu et al., 2022) was proposed in which the joint entropy of the optimal $\boldsymbol{x}$ and $\mathcal{F}^*$ is considered. This approach is also only uses over-truncation. Another approach considering the lower

bound of MI is Parallel Feasible Pareto Frontier Entropy Search ($\{PF\}^2ES$) (Qing et al., 2022), which also pointed out the problem of over-truncation. However, $\{PF\}^2ES$ is still based only on over-truncation. A shifting parameter $\varepsilon \in \mathbb{R}^L$ of the Pareto-frontier is heuristically introduced to mitigate over-truncation, for which theoretical justification is not clarified. Further, their criterion is not guaranteed as a lower bound in general (only when $L = 2$ with the assumption $|\mathcal{F}_S^*| \to \infty$).

## 5. Discussion on Extensions

PFEV is a general framework so that we can drive extensions for the following four scenarios:

**Parallel querying:** In parallel querying, we consider querying multiple points at one iteration, which is an important practical setting. Suppose that we consider $Q > 1$ points selection. Let $\mathcal{X}_q := \{\boldsymbol{x}^{(1)}, \ldots, \boldsymbol{x}^{(q)}\}$ and $\mathcal{H}_q = \{\boldsymbol{f}_{\boldsymbol{x}^{(1)}}, \ldots, \boldsymbol{f}_{\boldsymbol{x}^{(q)}}\}$ for $q \leq Q$. The MI for all $Q$ points $\text{MI}(\mathcal{H}_Q; \mathcal{F}^*)$ represents the benefit of selecting $\mathcal{X}_q$, but this results in a $(Q \times d)$-dimensional optimization problem, which can be unstable. Instead, we follow the approach in (Takeno et al., 2022), which is a greedy selection based on conditional mutual information (CMI). When we select the $(q + 1)$-th $\boldsymbol{x}$ after determining $\boldsymbol{x}^{(1)}, \ldots, \boldsymbol{x}^{(q)}$, the MI can be decomposed into

$$\text{MI}(\mathcal{H}_q \cup \boldsymbol{f}_{\boldsymbol{x}}; \mathcal{F}^*) = \text{MI}(\mathcal{H}_q; \mathcal{F}^*) + \text{CMI}(\boldsymbol{f}_{\boldsymbol{x}}; \mathcal{F}^* \mid \mathcal{H}_q),$$

where $\text{CMI}(\boldsymbol{f}_{\boldsymbol{x}}; \mathcal{F}^* \mid \mathcal{H}_q) = \mathbb{E}_{\mathcal{H}_q}[\text{MI}(\boldsymbol{f}_{\boldsymbol{x}}; \mathcal{F}^* \mid \mathcal{H}_q)]$ is the MI conditioned on $\mathcal{H}_q$. Since the first term does not depend on $\boldsymbol{x}$, we only need to optimize CMI to select $\boldsymbol{x}$. The calculation of CMI can be performed by adding sampled $\mathcal{H}_q$ into the training dataset of the GPs, after which evaluation of the lower bound is almost same as the single querying. See Appendix E for details.

**Decoupled setting:** In the decoupled setting, we can observe only one selected objective function instead of querying all $L$ objective functions simultaneously (Hernandez-Lobato et al., 2016). In PFEV, the criterion for the decoupled setting can be defined as $\text{MI}(f_{\boldsymbol{x}}^l; \mathcal{F}^*)$, which is the information gain from only one objective function $f_{\boldsymbol{x}}^l$. The lower bound of $\text{MI}(f_{\boldsymbol{x}}^l; \mathcal{F}^*)$ can be derived by the same approach shown in Section 3. In this case, we need to approximate $p(f_{\boldsymbol{x}}^l \mid \mathcal{F}^*)$ instead of $p(\boldsymbol{f}_{\boldsymbol{x}} \mid \mathcal{F}^*)$, for which we can use the marginal distribution of $q_\lambda(\boldsymbol{f}_{\boldsymbol{x}} \mid \mathcal{F}^*)$. See Appendix F for detail.

**Joint entropy search:** For PFEV, not only the information gain of the Pareto-frontier $\mathcal{F}^*$ but also the corresponding input $\boldsymbol{x}$ can be considered as $\text{MI}(\boldsymbol{f}_{\boldsymbol{x}}; \mathcal{X}^*, \mathcal{F}^*)$, where $\mathcal{X}^* = \{\boldsymbol{x} \mid \boldsymbol{f}_{\boldsymbol{x}} \in \mathcal{F}^*\}$. This can be seen as a JES extension of PFEV. When we derive the lower bound for

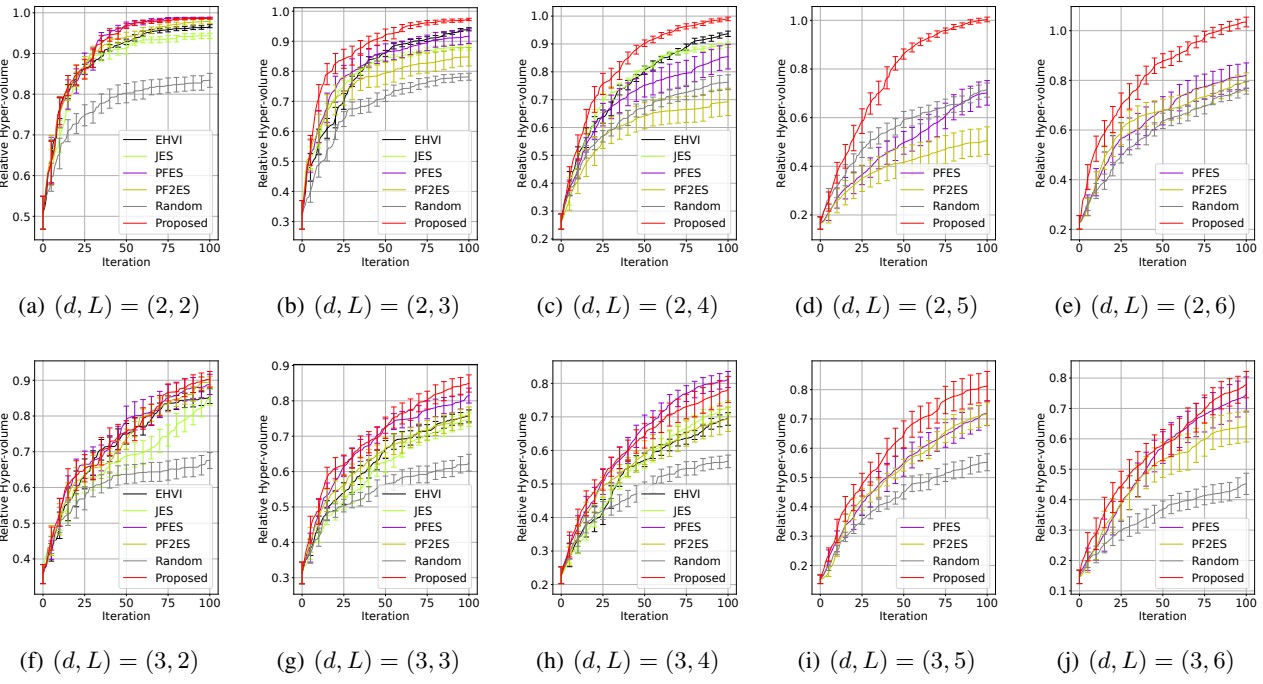

*Figure 3.* Performance comparison in GP-derived synthetic functions (average and standard deviation of 10 runs).

$\mathrm{MI}(\boldsymbol{f}_{\boldsymbol{x}}; \mathcal{X}^*, \mathcal{F}^*)$ by the same approach as Section 3, a variational approximation for $p(\boldsymbol{f} \mid \mathcal{X}^*, \mathcal{F}^*)$ is required. A basic idea of JES (Tu et al., 2022) is to simply add $(\mathcal{X}^*, \mathcal{F}^*)$ into the training data of the GPs. In the case of PFEV, we can define the variational distribution $q(\boldsymbol{f} \mid \mathcal{X}^*, \mathcal{F}^*)$ by using the same mixture as (3). The only difference is that $(\mathcal{X}^*, \mathcal{F}^*)$ is added to the training data of the GPs. Since we have not observed particular performance improvement by this additional conditioning, we employ $\mathrm{MI}(\boldsymbol{f}_{\boldsymbol{x}}; \mathcal{F}^*)$ as the default setting of PFEV. See Appendix G for detail.

**Noisy observations:** It is also possible to derive the information gain obtained from noisy observation $\mathrm{MI}(\boldsymbol{y}_{\boldsymbol{x}}; \mathcal{F}^*)$, where $\boldsymbol{y}_{\boldsymbol{x}} = \boldsymbol{f}_{\boldsymbol{x}} + \boldsymbol{\varepsilon}$ with $\boldsymbol{\varepsilon} \sim \mathcal{N}(\boldsymbol{0}, \sigma_{\mathrm{noise}}^2 \boldsymbol{I})$, though we mainly consider $\mathrm{MI}(\boldsymbol{f}_{\boldsymbol{x}}; \mathcal{F}^*)$ for brevity. In this case, instead of $p(\boldsymbol{f}_{\boldsymbol{x}} \mid \mathcal{F}^*)$, we need to consider $p(\boldsymbol{y}_{\boldsymbol{x}} \mid \mathcal{F}^*)$. Through the relation $p(\boldsymbol{y}_{\boldsymbol{x}} \mid \mathcal{F}^*) = \int p(\boldsymbol{y}_{\boldsymbol{x}} \mid \boldsymbol{f}_{\boldsymbol{x}}) p(\boldsymbol{f}_{\boldsymbol{x}} \mid \mathcal{F}^*) \mathrm{d}\boldsymbol{f}_{\boldsymbol{x}}$, we can derive the lower bound based on the same approximation of $p(\boldsymbol{f}_{\boldsymbol{x}} \mid \mathcal{F}^*)$ by using $q_\lambda$. See Appendix H for detail.

## 6. Experiments

We empirically verify the performance of PFEV by comparing mainly with EHVI (Emmerich, 2005), PFES (Suzuki et al., 2020), {PF}²ES (Qing et al., 2022), JES (Tu et al., 2022), and random search. In Appendix K, we added other methods such as ParEGO (Knowles, 2006),

MOBO-RS (Paria et al., 2020), and MESMO (Belakaria et al., 2019), which are mostly omitted in the main text for brevity of plots. We show results on GP-based synthetic functions, benchmark functions, and hyper-parameter optimization problems.

Each evaluation run 10 times. As a performance metric, RHV (relative hyper-volume) was used. RHV is defined as the hyper-volume of the observed Pareto-frontier divided by the volume of the reference Pareto-frontier. The reference Pareto-frontier was obtained by $10,000$ iterations of NSGA-II. All methods used GPs for $f_{\boldsymbol{x}}^l$ with a kernel function $k(\boldsymbol{x}, \boldsymbol{x}') = \exp(-\|\boldsymbol{x} - \boldsymbol{x}'\|_2^2 / (2\ell_{\mathrm{RBF}}^2))$, where $\ell_{\mathrm{RBF}} \in \mathbb{R}$ is a hyper-parameter. The marginal likelihood maximization was performed at every iteration to optimize $\ell_{\mathrm{RBF}}$. The number of samples of the optimal value or the Pareto-frontier in MESMO, PFES, {PF}²ES, and PFEV was 10, each of which was performed by NSGA-II ($1,000$ generations and population size 50). We used the DIRECT algorithm (Jones et al., 1993) for the acquisition function maximization. We selected 5 random $\boldsymbol{x}$ as the initial observations in $\mathcal{D}$. Depending on problems, EHVI and JES were performed up to $L = 4$ due to computational issues. Other settings are described in Appendix J.

### 6.1. Synthetic Functions Generated by GPs

Here, we consider synthetic functions from GPs as true objective functions, i.e., $f^l \sim \mathcal{GP}(0, k)$, in which $\ell_{\mathrm{RBF}} = 0.1$

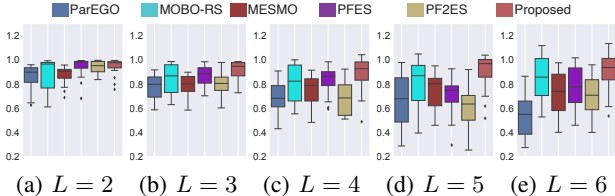

(a) $L = 2$ (b) $L = 3$ (c) $L = 4$ (d) $L = 5$ (e) $L = 6$

Figure 4. Boxplots of RHV at 100-th iteration. Note that, here, methods omitted in Fig. 3 (ParEGO, MOBO-RS, and MESMO) are also included, which are shown Appendix K.1.

was used in the kernel $k$. Since we require objective functions in a continuous domain, we used the RFM-based approximation (the number of RFM basis is $D = 1000$). The input dimensions are $d \in \{2, 3\}$, and the domain $\mathcal{X}$ is $[0, 1]^d$. The output dimensions are $L \in \{2, 3, 4, 5, 6\}$.

Figure 3 shows the results. In all (a)-(j), PFEV shows superior or comparable performance to existing methods. In these experiments, we empirically see that the performance of PFEV is often similar to PFES and $\{PF\}^2ES$ for $L = 2$, and differences become clearer for $L \geq 3$. This can also be confirmed by Fig. 4, which shows boxplots of RHV at the 100-th iteration in all trials of Fig. 3. This result is consistent with our conjecture about the difference of two types of PFTN described in Section 3.2. In Appendix K.1, additional results on different $\ell_{RBF}$ and $d = 4$ are shown, in which similar tendency was confirmed.

### 6.2. Benchmark Functions

We used benchmark problems called Fonseca-Fleming $(d, L) = (2, 2)$, Kursawe $(d, L) = (3, 2)$, Viennet $(d, L) = (2, 3)$ and FES3 $(d, L) = (3, 4)$ problems (for details, see Appendix J). Further, we combine multiple problems having the same input dimensions, i.e., we created Fonseca+Viennet $(d, L) = (2, 5)$ and FES3+Kursawe $(d, L) = (3, 6)$. Since the input domain is shared (which is scaled to $[0, 1]^d$ beforehand), only the output dimension increases with the original problems. The results are shown in Fig. 5. Overall, the performance of proposed PFEV is high among the compared methods. Here again, we see that in problems with the output dimension $\geq 3$, PFEV tends to show its advantage. Additional results on benchmark functions are also shown in Appendix K.2.

### 6.3. Hyper-parameter Optimization of LightGBM

As an application example of hyper-parameter optimization problems, we consider optimizing class weights in multi-class classification problems using LightGBM (Ke et al., 2017) as the base model (this setting is from (Ozaki et al., 2024)). We focused solely on optimizing the class weight parameters, making the input dimension of BO equal to the number of classes. The objective functions are the test

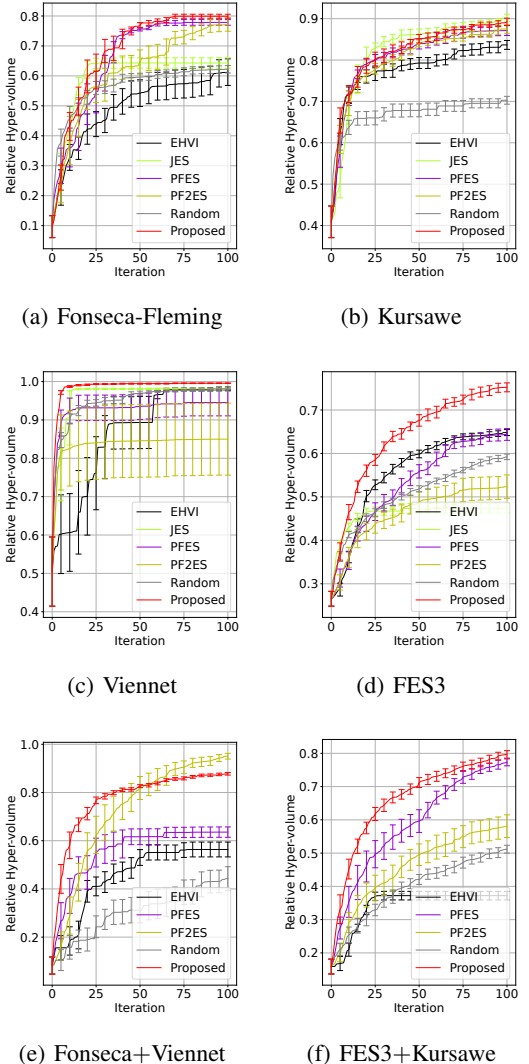

(a) Fonseca-Fleming (b) Kursawe

(c) Viennet (d) FES3

(e) Fonseca+Viennet (f) FES3+Kursawe

Figure 5. Performance comparison in benchmark functions (average and standard deviation of 10 runs).

classification accuracies for each class, which means the output dimension also equals to the number of classes. Experiments were conducted on four datasets: Abalone and Waveform (both with 3 classes), and Pageblocks and Gesturephase (both with 5 classes). For each dataset, we split the original data into training and test sets with an 8:2 ratio. Figure 6 shows the results, where the vertical axis represents the hyper-volume of test accuracies across all classes, calculated using the reference point at $\mathbf{0}$. Note that each objective function is in $[0, 1]$ (classification accuracy of each class), and therefore, the volume is also in $[0, 1]$ (if all classes achieve accuracy 1, the volume becomes 1). As observed in the figure, overall, PFEV shows sufficiently high performance among the compared methods.

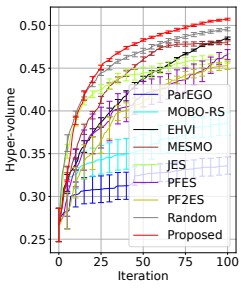
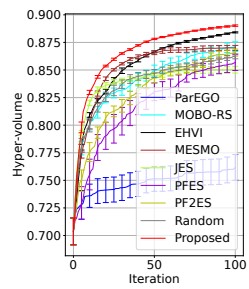

(a) Abalone (4177 samples, 8 features, 3 classes)

(b) Waveform (5000 samples, 40 features, 3 classes)

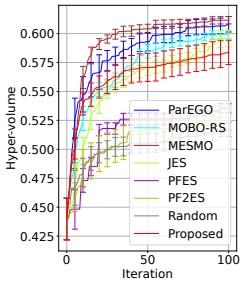
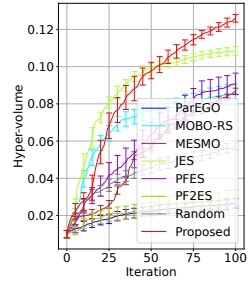

(c) Pageblocks (5473 samples, 10 features, 5 classes)

(d) Gesturephase (9873 samples, 33 features, 5 classes)

*Figure 6.* Results on hyper-parameter (class-weights) optimization.

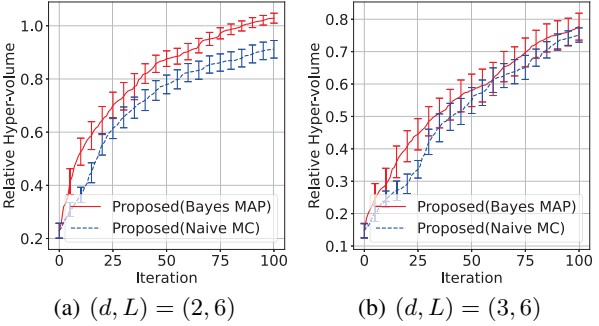

(a) $(d, L) = (2, 6)$

(b) $(d, L) = (3, 6)$

*Figure 7.* Comparison between two MC estimators (6) and (8) of PFEV using GP-derived synthetic functions.

### 6.4. Comparison of PFEV with Different MC Estimators

By using GP-derived synthetic functions, two estimators of PFEV (6) and (8) are compared. The same settings as in Section 6.1 were used. The results of $(d, L) = (2, 6)$ and $(d, L) = (3, 6)$ are shown in Fig. 7. We see that the MAP-based approximation (8) improves the performance compared with (6). We confirmed similar results on other GP-derived functions, which is summarized in Fig. 21 in Appendix K.1.

## 7. Discussion on Gradient-based Optimization of Acquisition Function

We employed DIRECT as an optimizer of the acquisition function because it does not require 'initial points' unlike the gradient descent that requires the appropriate setting of the initial points (the number of initial points and locations). Our purpose is to focus more on differences of the acquisition functions and to reduce the other factors affecting the performance. On the other hand, evaluation using gradient-based approaches is also important future work because it is widely used in BO and should have high performance in general. We partially show comparison with a baseline using the gradient optimization (qLogNEHVI in BoTorch) in Appendix M.

Note that the proposed acquisition function (8) is mostly differentiable, except for the indicator $\mathbb{I}(\tilde{\boldsymbol{f}}_{\boldsymbol{x}} \in \mathcal{A}_O^{\tilde{\mathcal{F}}_S^*})$ in $\theta_{\mathrm{MAP}}(\tilde{\boldsymbol{f}}_{\boldsymbol{x}})$. We consider that possible approaches are simply ignoring this term in the gradient (regarding the gradient of the indicator as 0) or using a continuous approximation of the gradient. In the continuous approximation, we replace $\mathbb{I}(\tilde{\boldsymbol{f}}_{\boldsymbol{x}} \in \mathcal{A}_O^{\tilde{\mathcal{F}}_S^*}) \approx p(\boldsymbol{f} \in \mathcal{A}_O^{\tilde{\mathcal{F}}_S^*})$, where $\boldsymbol{f} \sim \mathcal{N}(\tilde{\boldsymbol{f}}_{\boldsymbol{x}}, \rho \boldsymbol{I})$ in which $\rho > 0$ is a fixed smoothing parameter. The right hand side of the approximation is differentiable with respect to $\boldsymbol{x}$ through the similar decomposition to (9) in Appendix B (note that $\tilde{\boldsymbol{f}}_{\boldsymbol{x}}$ is differential because it is generated from RFM). This can be interpreted as a counter-part of the standard CDF-based smoothing approximation of an indicator function, extended to the indicator of the Pareto dominated region. For the calculation, although the cell-based decomposition is required for $\mathcal{A}_O^{\tilde{\mathcal{F}}_S^*}$, we can reuse the cells created in line 7 of Algorithm 1.

## 8. Conclusions

We proposed a multi-objective Bayesian optimization acquisition function that is based on variational lower bound maximization. By combining two normal distributions, defined by under- and over-truncation of Pareto-frontier, we introduced a variational distribution as mixture of these two distributions based on which a lower bound of mutual information can be constructed. Performance superiority was shown by GP-generated functions and benchmark functions. A current limitation includes theoretical guarantee of the MI approximation and convergence, which is important open problems for information-theoretic BO. Another possible future direction is to use a more complicated variational distribution. We employ the simple one parameter ($\lambda$) distribution because $q$ should be estimated based on the $K$ samples, which is usually quite small (10 in the experiments). However, to make the lower bound tight, a more flexible distribution may be required.

## Acknowledgement

This work is partially supported by JSPS KAKENHI (Grant Number 23K21696, 25K03182, 22H00300, and 23K17817), and Data Creation and Utilization Type Material Research and Development Project (Grant No. JP-MXP1122712807) of MEXT.

## Impact Statement

This paper presents work whose goal is to advance the field of Bayesian optimization. There are many potential societal consequences of our work, none which we feel must be specifically highlighted here.

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

## A. Proof of Remark 3.2

The lower bound of (5) is derived by using the case of $\lambda = 1$:

$$
\begin{aligned}
&\max_{\lambda \in (0,1]} \mathrm{LB}(\boldsymbol{x}, \lambda) \\
&\geq \mathrm{LB}(\boldsymbol{x}, 1) \\
&= \mathbb{E}_{\mathcal{F}^*, \boldsymbol{f_x}} \left[ \log \left( \frac{q_U(\boldsymbol{f_x} \mid \mathcal{F}_S^*)}{p(\boldsymbol{f_x})} \right) \right] \\
&= -\mathbb{E}_{\mathcal{F}^*} \left[ \int_{\mathcal{A}_U^{\mathcal{F}_S^*}} p(\boldsymbol{f_x} \mid \mathcal{F}^*) \log Z_U^{\mathcal{F}_S^*}(\boldsymbol{x}) \mathrm{d}\boldsymbol{f_x} \right] \\
&= -\mathbb{E}_{\mathcal{F}^*} \left[ \log Z_U^{\mathcal{F}_S^*}(\boldsymbol{x}) \right] \\
&= -\mathbb{E}_{\mathcal{F}^*} \left[ \log(1 - p(\boldsymbol{f_x} \notin \mathcal{A}_U^{\mathcal{F}_S^*})) \right] \\
&\geq \mathbb{E}_{\mathcal{F}^*} \left[ p(\boldsymbol{f_x} \notin \mathcal{A}_U^{\mathcal{F}_S^*}) \right] > 0
\end{aligned}
$$

## B. Computations of $Z_O^{\tilde{\mathcal{F}}_S^*}$ and $Z_U^{\tilde{\mathcal{F}}_S^*}$

First, we consider the normalization constant of PFTN-O $Z_O^{\tilde{\mathcal{F}}_S^*} = p(\boldsymbol{f_x} \in \mathcal{A}_O^{\mathcal{F}_S^*})$. We decompose $\mathcal{A}_O^{\tilde{\mathcal{F}}_S^*}$ into disjoint hyper-rectangles, denoted as $\mathcal{A}_O^{\tilde{\mathcal{F}}_S^*} = \mathcal{C}_1 \cup \mathcal{C}_2 \cup \ldots \cup \mathcal{C}_C$, where $\mathcal{C}_i$ is a hyper-rectangle and $C$ is the number of hyper-rectangles. Algorithms decomposing a dominated region have been studied in the context of the Pareto hyper-volume computation. We use quick hypervolume (QHV) (Russo & Francisco, 2014), used also in PFES. Each hyper-rectangle $\mathcal{C}_i$ is written as

$$
\mathcal{C}_i = \left( \ell_i^1, u_i^1 \right] \times \left( \ell_i^2, u_i^2 \right] \times \ldots \times \left( \ell_i^L, u_i^L \right],
$$

where $\ell_i^l$ and $u_i^l$ are the smallest and largest values in the $l$-th dimension of the $i$-th hyper-rectangle. From the independence of the objective functions, $Z_O^{\tilde{\mathcal{F}}_S^*}$ can be easily computed by

$$
\begin{aligned}
Z_O^{\tilde{\mathcal{F}}_S^*} &= \sum_{i=1}^{C} \int_{\mathcal{C}_i} p(\boldsymbol{f_x}) \mathrm{d}\boldsymbol{f_x} \\
&= \sum_{i=1}^{C} \prod_{l=1}^{L} \int_{\ell_i^l}^{u_i^l} p(f_{\boldsymbol{x}}^l) \mathrm{d}f_{\boldsymbol{x}}^l \\
&= \sum_{i=1}^{C} \prod_{l=1}^{L} \left( \Phi(\bar{\alpha}_{i,l}) - \Phi(\underline{\alpha}_{i,l}) \right), \quad (9)
\end{aligned}
$$

where $\underline{\alpha}_{i,l} = (\ell_i^l - \mu_l(\boldsymbol{x}))/\sigma_l(\boldsymbol{x})$ and $\bar{\alpha}_{i,l} = (u_i^l - \mu_l(\boldsymbol{x}))/\sigma_l(\boldsymbol{x})$, and $\Phi$ is the cumulative distribution function of the standard normal distribution.

Next, we consider $Z_U^{\tilde{\mathcal{F}}_S^*}$ that can be re-written as

$$
\begin{aligned}
Z_U^{\tilde{\mathcal{F}}_S^*} &= p \left( \boldsymbol{f_x} \in \mathcal{A}_U^{\tilde{\mathcal{F}}_S^*} \right) \\
&= 1 - p \left( \boldsymbol{f_x} \notin \mathcal{A}_U^{\tilde{\mathcal{F}}_S^*} \right) \\
&= 1 - p \left( -\boldsymbol{f_x} \preceq \{ -\boldsymbol{f} \mid \boldsymbol{f} \in \tilde{\mathcal{F}}_S^* \} \right).
\end{aligned}
$$

In the last equation, the region $\boldsymbol{f_x} \notin \mathcal{A}_U^{\tilde{\mathcal{F}}_S^*}$ is re-written by flipping the sign of $\tilde{\mathcal{F}}_S^*$ in such a way that the region is written as a "dominated region". This enables us to use QHV, which decomposes a "dominated region" into hyper-rectangles, for the computation of $p(-\boldsymbol{f_x} \preceq \{ -\boldsymbol{f} \mid \boldsymbol{f} \in \tilde{\mathcal{F}}_S^* \})$ based on almost the same way as (9).

## C. Properties about Maximization of $\lambda$

We first show the concavity. To simplify the notation, we unify (8) and (6) as

$$
\begin{aligned}
\widehat{\mathrm{LB}}(\boldsymbol{x}, \lambda) &= \frac{1}{K} \sum_{(\tilde{\mathcal{F}}_S^*, \tilde{\boldsymbol{f}}) \in F} \log \left( \frac{\lambda}{Z_U} + \frac{1 - \lambda}{Z_O} \right) \xi(\tilde{\boldsymbol{f}}_{\boldsymbol{x}}) \\
&\quad + \log \left( \frac{\lambda}{Z_U} \right) (1 - \xi(\tilde{\boldsymbol{f}}_{\boldsymbol{x}})),
\end{aligned}
$$

where $Z_U = Z_U^{\tilde{\mathcal{F}}_S^*}(\boldsymbol{x})$, $Z_O = Z_O^{\tilde{\mathcal{F}}_S^*}(\boldsymbol{x})$, and $\xi(\tilde{\boldsymbol{f}}_{\boldsymbol{x}}) = \theta_{\mathrm{MAP}}(\tilde{\boldsymbol{f}}_{\boldsymbol{x}})$ when (8), and $\xi(\tilde{\boldsymbol{f}}_{\boldsymbol{x}}) = \mathbb{I}(\tilde{\boldsymbol{f}}_{\boldsymbol{x}} \in \mathcal{A}_O^{\tilde{\mathcal{F}}_S^*})$ when (6). The second derivative is

$$
\begin{aligned}
\frac{\partial^2 \widehat{\mathrm{LB}}(\boldsymbol{x}, \lambda)}{\partial \lambda^2} &= \frac{1}{K} \sum_{(\tilde{\mathcal{F}}_S^*, \tilde{\boldsymbol{f}}) \in F} - \left( \frac{Z_O - Z_U}{\lambda(Z_O - Z_U) + Z_U} \right)^2 \xi(\tilde{\boldsymbol{f}}_{\boldsymbol{x}}) \\
&\quad - \frac{1}{\lambda^2} (1 - \xi(\tilde{\boldsymbol{f}}_{\boldsymbol{x}})) \leq 0.
\end{aligned}
$$

Thus, we see concavity.

Further, in the case of (8), we can show that the optimal $\lambda$ should exist for $\max_{\lambda \in (0,1]} \widehat{\mathrm{LB}}(\boldsymbol{x}, \lambda)$ though $\lambda$ has a left open interval as a domain. First, we can derive $\hat{p} \in (0, 1)$ from its definition $\hat{p} = Z_O^{\tilde{\mathcal{F}}_S^*}(\boldsymbol{x})/Z_U^{\tilde{\mathcal{F}}_S^*}$. Since $\boldsymbol{f_x}$ is the Gaussian distribution, $Z_O^{\tilde{\mathcal{F}}_S^*}(\boldsymbol{x}) = p(\boldsymbol{f_x} \in \mathcal{A}_O^{\tilde{\mathcal{F}}_S^*}) \in (0, 1)$ and $Z_O^{\tilde{\mathcal{F}}_S^*}(\boldsymbol{x}) < Z_U^{\tilde{\mathcal{F}}_S^*}(\boldsymbol{x}) = p(\boldsymbol{f_x} \in \mathcal{A}_U^{\tilde{\mathcal{F}}_S^*}) \in (0, 1)$. Here, we used $\emptyset \neq \mathcal{A}_O^{\tilde{\mathcal{F}}_S^*} \subset \mathcal{A}_U^{\tilde{\mathcal{F}}_S^*} \subset \mathbb{R}^L$ (neither $\mathcal{A}_O^{\tilde{\mathcal{F}}_S^*}$ nor $\mathcal{A}_U^{\tilde{\mathcal{F}}_S^*}$ is the empty set or the entire output space, and $\mathcal{A}_O^{\tilde{\mathcal{F}}_S^*} \subset \mathcal{A}_U^{\tilde{\mathcal{F}}_S^*}$ holds as far as the sampled Pareto frontier set $\tilde{\mathcal{F}}_S^*$ is finite while the true Pareto frontier is continuous which is our problem setting). Therefore, we see $\theta_{\mathrm{MAP}} \in (0, 1)$. When $\lambda \to 0$, the second term of (8) goes to $\log 0$ (from the definition of $\eta_\lambda^{\tilde{\mathcal{F}}_S^*}$). On the other hand, the first term is finite even when $\lambda \to 0$. As a result, if $\lambda \to 0$, we see (8) goes to $-\infty$.

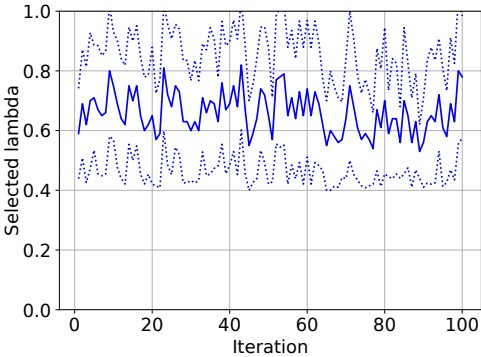

*Figure 8.* Transition of selected $\lambda$ for GP-derived synthetic function ($d = 3, L = 4$).

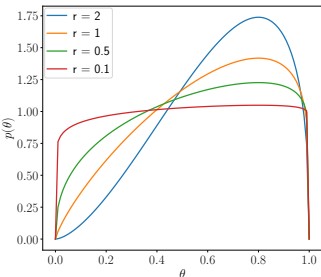

*Figure 9.* Density functions of the beta distribution when $\hat{p} = 0.8$. For all $r$, the mode is 0.8.

Figure 8 shows the transition of $\lambda$ during BO iterations (the average and standard deviation of 10 runs) for a GP-derived synthetic function ($d = 3, L = 4$). We see that $\lambda$ takes intermediate values in $(0, 1]$, indicating that under- and over-truncated distributions are indeed mixed during BO iterations. As far as we have examined, no consistent increasing or decreasing tendency has been observed during iterations.

# D. Combining Prior Knowledge in MC Estimator

## D.1. Derivation of MAP

We use the beta distribution for the prior $\theta \sim \text{Beta}(a, b)$. The density function is

$$p(\theta) = \frac{\theta^{a-1}(1-\theta)^{b-1}}{B(a, b)}.$$

Using the approximation $\theta \approx \hat{p}$, we set $a - 1 = r\hat{p}$ and $b - 1 = r(1 - \hat{p})$, where $r \geq 0$ is a parameter. This sets the mode of $p(\theta)$ as $\hat{p}$. As shown in Fig. 9, a larger $r$ has a stronger peak at $\hat{p}$. When $r = 0$, $p(\theta)$ becomes uniform in $[0, 1]$.

The posterior of the beta distribution with the Bernoulli distribution likelihood is $\text{Beta}(a + m, b + \ell)$, where $m$ is

the number of 'success' and $\ell$ is the number of 'fail' in Bernoulli trials. We can interpret that $\mathbb{I}(\boldsymbol{f_x} \in \mathcal{A}_O^{\mathcal{F}_S^*})$ is a sample from the Bernoulli distribution with the probability $p(\boldsymbol{f_x} \in \mathcal{A}_O^{\tilde{\mathcal{F}}_S^*} \mid \mathcal{F}^*)$. Therefore, we set $m = \mathbb{I}(\boldsymbol{f_x} \in \mathcal{A}_O^{\mathcal{F}_S^*})$ and $\ell = 1 - \mathbb{I}(\boldsymbol{f_x} \in \mathcal{A}_O^{\mathcal{F}_S^*})$, from which the mode of the posterior $\text{Beta}(a + m, b + \ell)$ can be derived as

$$
\begin{aligned}
\theta_{\text{MAP}} &= \frac{a - 1 + m}{a - 1 + m + b - 1 + \ell} \\
&= \frac{r\hat{p} + \mathbb{I}(\boldsymbol{f_x} \in \mathcal{A}_O^{\mathcal{F}_S^*})}{r\hat{p} + \mathbb{I}(\boldsymbol{f_x} \in \mathcal{A}_O^{\mathcal{F}_S^*}) + r(1 - \hat{p}) + (1 - \mathbb{I}(\boldsymbol{f_x} \in \mathcal{A}_O^{\mathcal{F}_S^*}))} \\
&= \frac{r\hat{p} + \mathbb{I}(\boldsymbol{f_x} \in \mathcal{A}_O^{\mathcal{F}_S^*})}{r + 1}.
\end{aligned}
$$

In the main text, we employ $r = 1$.

The lower bound estimation by MAP (8) can have an estimation bias caused by the approximation $\hat{p}$, though it is almost negligible when $K$ is a typical setting (such as $K = 10$). This bias occurs because we only have one $\tilde{\boldsymbol{f}}_x$ for each corresponding $\tilde{\mathcal{F}}_S^*$. Therefore, in each $\theta_{\text{MAP}}$, the effect of the prior remains even when $K$ is large. We can easily avoid this bias by setting $r$ so that it decreases $r \to 0$ when $K$ increases, by which the estimator (8) converges to the usual MC estimator (6). In Appendix D.3, we show that, in practice, the MAP-based approach has an advantage for small $K$ setting.

## D.2. Analyzing Variance

We re-write the estimator of the lower bound (8) as

$$\frac{1}{K} \sum_{(\tilde{\mathcal{F}}_S^*, \tilde{\boldsymbol{f}}) \in F} a(\tilde{\mathcal{F}}_S^*)\xi(\tilde{\boldsymbol{f}}_x) + b(\tilde{\mathcal{F}}_S^*).$$

where $a(\tilde{\mathcal{F}}_S^*) = \log\left(\frac{\lambda}{Z_U^{\tilde{\mathcal{F}}_S^*}(\boldsymbol{x})} + \frac{1-\lambda}{Z_O^{\tilde{\mathcal{F}}_S^*}(\boldsymbol{x})}\right) - \log\left(\frac{\lambda}{Z_U^{\tilde{\mathcal{F}}_S^*}(\boldsymbol{x})}\right)$, $b(\tilde{\mathcal{F}}_S^*) = \log\left(\frac{\lambda}{Z_U^{\tilde{\mathcal{F}}_S^*}(\boldsymbol{x})}\right)$, and $\xi(\tilde{\boldsymbol{f}}_x) \in [0, 1]$ is $\theta_{\text{MAP}}(\tilde{\boldsymbol{f}}_x)$ when (8) and is $\mathbb{I}(\tilde{\boldsymbol{f}}_x \in \mathcal{A}_O^{\tilde{\mathcal{F}}_S^*})$ when (6). By using independence of the MC samples and law of total variance, the variance of this estimator is decomposed as follows:

$$
\begin{aligned}
&\mathbb{V}_{\tilde{\mathcal{F}}^*, \tilde{\boldsymbol{f}}}\left[\frac{1}{K}\sum_{(\tilde{\mathcal{F}}_S^*, \tilde{\boldsymbol{f}}) \in F} a(\tilde{\mathcal{F}}_S^*)\xi(\tilde{\boldsymbol{f}}_x) + b(\tilde{\mathcal{F}}_S^*)\right] \\
&= \frac{1}{K}\mathbb{E}_{\tilde{\mathcal{F}}^*}\mathbb{V}_{\tilde{\boldsymbol{f}}|\tilde{\mathcal{F}}^*}\left[a(\tilde{\mathcal{F}}_S^*)\xi(\tilde{\boldsymbol{f}}_x) + b(\tilde{\mathcal{F}}_S^*)\right] \\
&\quad + \frac{1}{K}\mathbb{V}_{\tilde{\mathcal{F}}^*}\mathbb{E}_{\tilde{\boldsymbol{f}}|\tilde{\mathcal{F}}^*}\left[a(\tilde{\mathcal{F}}_S^*)\xi(\tilde{\boldsymbol{f}}_x) + b(\tilde{\mathcal{F}}_S^*)\right] \\
&= \frac{1}{K}\Bigg\{\mathbb{E}_{\tilde{\mathcal{F}}^*}\left[a(\tilde{\mathcal{F}}_S^*)^2\mathbb{V}_{\tilde{\boldsymbol{f}}|\tilde{\mathcal{F}}^*}\left[\xi(\tilde{\boldsymbol{f}}_x)\right]\right] + \mathbb{E}_{\tilde{\mathcal{F}}^*}\left[b(\tilde{\mathcal{F}}_S^*)\right] \\
&\quad + \mathbb{V}_{\tilde{\mathcal{F}}^*}\left[a(\tilde{\mathcal{F}}_S^*)\mathbb{E}_{\tilde{\boldsymbol{f}}|\tilde{\mathcal{F}}^*}\left[\xi(\tilde{\boldsymbol{f}}_x)\right] + b(\tilde{\mathcal{F}}_S^*)\right]\Bigg\} \quad (10)
\end{aligned}
$$

In the first term of (10), only $\mathbb{V}_{\tilde{\boldsymbol{f}}|\tilde{\mathcal{F}}^*}\left[\xi(\tilde{\boldsymbol{f}}_{\boldsymbol{x}})\right]$ changes depending on $\xi$. From

$$\mathbb{V}_{\tilde{\boldsymbol{f}}|\tilde{\mathcal{F}}^*}\left[\theta_{\mathrm{MAP}}(\tilde{\boldsymbol{f}}_{\boldsymbol{x}})\right] = \mathbb{V}_{\tilde{\boldsymbol{f}}|\tilde{\mathcal{F}}^*}\left[\mathbb{I}(\tilde{\boldsymbol{f}}_{\boldsymbol{x}} \in \mathcal{A}_O^{\tilde{\mathcal{F}}^*})\right]/4,$$

we see that MAP makes the variance of the first term $1/4$. The second term in (10) does not change depending on $\xi$. In the case of the MAP estimate, the third term in (10) is

$$\begin{aligned}
&\mathbb{V}_{\tilde{\mathcal{F}}^*}\left[a(\tilde{\mathcal{F}}_S^*)\mathbb{E}_{\tilde{\boldsymbol{f}}|\tilde{\mathcal{F}}^*}\left[\xi(\tilde{\boldsymbol{f}}_{\boldsymbol{x}})\right] + b(\tilde{\mathcal{F}}_S^*)\right] \\
&= \mathbb{V}_{\tilde{\mathcal{F}}^*}\left[a(\tilde{\mathcal{F}}_S^*)\mathbb{E}_{\tilde{\boldsymbol{f}}|\tilde{\mathcal{F}}^*}\left[\frac{\hat{p} + \mathbb{I}(\tilde{\boldsymbol{f}}_{\boldsymbol{x}} \in \mathcal{A}_O^{\tilde{\mathcal{F}}_S^*})}{2}\right] + b(\tilde{\mathcal{F}}_S^*)\right] \\
&= \mathbb{V}_{\tilde{\mathcal{F}}^*}\left[a(\tilde{\mathcal{F}}_S^*)\frac{\hat{p} + p(\tilde{\boldsymbol{f}}_{\boldsymbol{x}} \in \mathcal{A}_O^{\tilde{\mathcal{F}}_S^*} \mid \tilde{\mathcal{F}}^*)}{2} + b(\tilde{\mathcal{F}}_S^*)\right].
\end{aligned}$$

If $\hat{p} \approx p(\tilde{\boldsymbol{f}}_{\boldsymbol{x}} \in \mathcal{A}_O^{\tilde{\mathcal{F}}_S^*} \mid \tilde{\mathcal{F}}^*)$, this should be similar to $\mathbb{V}_{\tilde{\mathcal{F}}^*}\left[a(\tilde{\mathcal{F}}_S^*)p(\tilde{\boldsymbol{f}}_{\boldsymbol{x}} \in \mathcal{A}_O^{\tilde{\mathcal{F}}_S^*} \mid \tilde{\mathcal{F}}^*) + b(\tilde{\mathcal{F}}_S^*)\right]$, which is the variance in the case of $\xi(\tilde{\boldsymbol{f}}_{\boldsymbol{x}}) = \mathbb{I}(\tilde{\boldsymbol{f}}_{\boldsymbol{x}} \in \mathcal{A}_O^{\tilde{\mathcal{F}}_S^*})$. Therefore, under the assumption of $\hat{p} \approx p(\tilde{\boldsymbol{f}}_{\boldsymbol{x}} \in \mathcal{A}_O^{\tilde{\mathcal{F}}_S^*} \mid \tilde{\mathcal{F}}^*)$ the variance reduction is expected because of the variance reduction in the first term of (10).

### D.3. Empirical Verification of MC Estimator of Lower Bound

Using a two objective problem generated by GPs ($d = 1$), we examine the accuracy of the MC estimator compared with the true lower bound. Hereafter, Naïve MC indicates the calculation by (6), and MC with Bayes MAP indicates the calculation by (8). We calculate the lower bound at 100 grid points in $x \in [0, 1]$ with random 5 training points. Regarding the result by Naïve MC with $K = 10^5$ samples as a pseudo ground-truth, compared with which we evaluate the estimation error. In addition to the fixed $r$ setting, we here examine the setting $r = \sqrt{10/K}$, by which $r$ reduces with $O(1/\sqrt{K})$ (same as the general convergence rate of the MC estimator) and $r = 1$ when $K = 10$ in this setting.

Figure 10 shows the results by mean squared error (MSE). MC with Bayes MAP shows $r = 1$ and $r = \sqrt{10/K}$, for which they have the same result when $K = 10$. When the sample size $K$ is small, MC with Bayes MAP has smaller errors for both the $r$ settings compared with Naïve MC. With the increase of $K$, all methods decrease MSE, but in the right log scale plot, the decrease of MC with Bayes MAP ($r = 1$) stagnates at around $K = 10^2$–$10^3$. On the other hand, MC with Bayes MAP ($r = \sqrt{10/K}$) continues to decrease MSE because it can diminish the effect of the approximation in the prior.

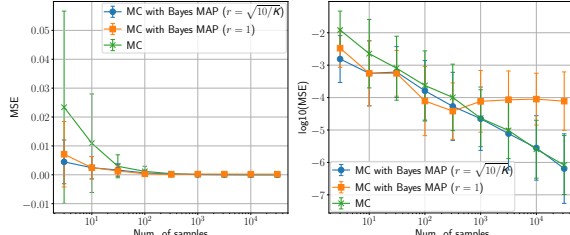

*Figure 10.* MSE compared with the pseudo ground truth, created by Naïve MC with $K = 10^5$. The plot is the original scale, and the right plot is the $\log_{10}$ scale.

## E. Parallel Querying

We here consider parallel querying in which $Q$ queries should be selected every iteration. Let $\mathcal{X}_q := \{\boldsymbol{x}^{(1)}, \ldots, \boldsymbol{x}^{(q)}\}$ and $\mathcal{H}_q = \{\boldsymbol{f}_{\boldsymbol{x}^{(1)}}, \ldots, \boldsymbol{f}_{\boldsymbol{x}^{(q)}}\}$ for $q \leq Q$. Then, $\mathrm{MI}(\mathcal{H}_Q; \mathcal{F}^*)$ can be a selection criterion for determining $Q$ points simultaneously. However, this leads to $(Q \times d)$-dimensional optimization. Instead, we employ a greedy strategy shown by (Takeno et al., 2022), in which the MI approximation can be reduced to a similar computation to the case of single querying.

Assume that we have already selected $q < Q$ points $\mathcal{X}_q$, for which observations are not obtained yet, and consider determining the $(q + 1)$-th point $\boldsymbol{x}$. In this case, $\mathrm{MI}(\mathcal{H}_q \cup \boldsymbol{f}_{\boldsymbol{x}}; \mathcal{F}^*)$ should be maximized with respect to the additional $\boldsymbol{x}$. We see

$$\mathrm{MI}(\mathcal{H}_q \cup \boldsymbol{f}_{\boldsymbol{x}}; \mathcal{F}^*) = \mathrm{MI}(\mathcal{H}_q; \mathcal{F}^*) + \mathrm{CMI}(\boldsymbol{f}_{\boldsymbol{x}}; \mathcal{F}^* \mid \mathcal{H}_q),$$

where $\mathrm{CMI}(\boldsymbol{f}_{\boldsymbol{x}}; \mathcal{F}^* \mid \mathcal{H}_q) = \mathbb{E}_{\mathcal{H}_q}[\mathrm{MI}(\boldsymbol{f}_{\boldsymbol{x}}; \mathcal{F}^* \mid \mathcal{H}_q)]$ is the MI conditioned on $\mathcal{H}_q$. Since the first term does not depend on $\boldsymbol{x}$, we only need to consider the second term $\mathrm{CMI}(\boldsymbol{f}_{\boldsymbol{x}}; \mathcal{F}^* \mid \mathcal{H}_q)$.

The lower bound of $\mathrm{MI}(\boldsymbol{f}_{\boldsymbol{x}}; \mathcal{F}^* \mid \mathcal{H}_q)$ is derived in the same way as the lower bound of $\mathrm{MI}(\boldsymbol{f}_{\boldsymbol{x}}; \mathcal{F}^*)$, i.e., (4). The only difference is that the GPs have additional training data consisting of $\mathcal{X}_q$ and $\mathcal{H}_q$. Let $\zeta_{\lambda,q}^{\mathcal{F}_S^*}(\boldsymbol{x})$ and $\eta_{\lambda,q}^{\mathcal{F}_S^*}(\boldsymbol{x})$ be $\zeta_{\lambda}^{\mathcal{F}_S^*}(\boldsymbol{x})$ and $\eta_{\lambda}^{\mathcal{F}_S^*}(\boldsymbol{x})$ in which the GPs with additional $q$ observations are used to calculate $Z_U^{\mathcal{F}_S^*}(\boldsymbol{x})$ and $Z_O^{\mathcal{F}_S^*}(\boldsymbol{x})$ as follows:

$$\begin{aligned}
\zeta_{\lambda,q}^{\mathcal{F}_S^*}(\boldsymbol{x}) &= \frac{\lambda}{Z_U^{\mathcal{F}_S^*}(\boldsymbol{x} \mid \mathcal{H}_q)} + \frac{1 - \lambda}{Z_O^{\mathcal{F}_S^*}(\boldsymbol{x} \mid \mathcal{H}_q)}, \\
\eta_{\lambda,q}^{\mathcal{F}_S^*}(\boldsymbol{x}) &= \frac{\lambda}{Z_U^{\mathcal{F}_S^*}(\boldsymbol{x} \mid \mathcal{H}_q)},
\end{aligned}$$

where $Z_U^{\mathcal{F}_S^*}(\boldsymbol{x} \mid \mathcal{H}_q) := p(\boldsymbol{f}_{\boldsymbol{x}} \in \mathcal{A}_U^{\mathcal{F}_S^*} \mid \mathcal{H}_q)$ and $Z_O^{\mathcal{F}_S^*}(\boldsymbol{x} \mid$

$\mathcal{H}_q) := p(\boldsymbol{f}_{\boldsymbol{x}} \in \mathcal{A}_O^{\mathcal{F}_S^*} \mid \mathcal{H}_q)$. Then, we can write

$$\mathbb{E}_{\mathcal{H}_q}\left[\mathrm{MI}(\boldsymbol{f}_{\boldsymbol{x}}; \mathcal{F}^* \mid \mathcal{H}_q)\right]$$

$$\geq \mathbb{E}_{\mathcal{H}_q}\mathbb{E}_{\mathcal{F}^*, \boldsymbol{f}_{\boldsymbol{x}}|\mathcal{H}_q}\left[\log\{\zeta_{\lambda,q}^{\mathcal{F}_S^*}(\boldsymbol{x})\mathbb{I}(\boldsymbol{f}_{\boldsymbol{x}} \in \mathcal{A}_O^{\mathcal{F}_S^*})\right.$$

$$\left. + \eta_{\lambda,q}^{\mathcal{F}_S^*}(\boldsymbol{x})\mathbb{I}(\boldsymbol{f}_{\boldsymbol{x}} \in \mathcal{A}_{U\backslash O}^{\mathcal{F}_S^*})\}\right]$$

$$= \mathbb{E}_{\mathcal{F}^*, \boldsymbol{f}_{\boldsymbol{x}}, \mathcal{H}_q}\left[\log\{\zeta_{\lambda,q}^{\mathcal{F}_S^*}(\boldsymbol{x})\mathbb{I}(\boldsymbol{f}_{\boldsymbol{x}} \in \mathcal{A}_O^{\mathcal{F}_S^*})\right.$$

$$\left. + \eta_{\lambda,q}^{\mathcal{F}_S^*}(\boldsymbol{x})\mathbb{I}(\boldsymbol{f}_{\boldsymbol{x}} \in \mathcal{A}_{U\backslash O}^{\mathcal{F}_S^*})\}\right] \quad (11)$$

Since the expectation $\mathbb{E}_{\mathcal{H}_q}\mathbb{E}_{\mathcal{F}^*, \boldsymbol{f}_{\boldsymbol{x}}|\mathcal{H}_q}$ can be seen as the joint expectation $\mathbb{E}_{\mathcal{F}^*, \boldsymbol{f}_{\boldsymbol{x}}, \mathcal{H}_q}$, the MC approximation can be performed by using samples from the joint distribution of $\mathcal{F}^*$, $\boldsymbol{f}_{\boldsymbol{x}}$ and $\mathcal{H}_q$. Let $F_q$ be a set of samples $(\tilde{\mathcal{F}}_S^*, \tilde{\boldsymbol{f}}_{\boldsymbol{x}}, \tilde{\mathcal{H}}_q)$ from the joint distribution. Then, the MC approximation of the lower bound (11) is

$$\frac{1}{K}\sum_{(\tilde{\mathcal{F}}_S^*, \tilde{\boldsymbol{f}}_{\boldsymbol{x}}, \tilde{\mathcal{H}}_q)\in F_q}\log\{\zeta_{\lambda,q}^{\tilde{\mathcal{F}}_S^*}(\boldsymbol{x})\mathbb{I}(\tilde{\boldsymbol{f}}_{\boldsymbol{x}} \in \mathcal{A}_O^{\tilde{\mathcal{F}}_S^*})$$

$$+ \eta_{\lambda,q}^{\tilde{\mathcal{F}}_S^*}(\boldsymbol{x})\mathbb{I}(\tilde{\boldsymbol{f}}_{\boldsymbol{x}} \in \mathcal{A}_{U\backslash O}^{\tilde{\mathcal{F}}_S^*})\}$$

The sampling of $F_q$ can be performed by almost the same procedure as the single querying. First, we generate the "entire function $\tilde{\boldsymbol{f}}$" by using RFM. $\tilde{\mathcal{F}}_S^*$ is obtained by applying NSGA-II to $\tilde{\boldsymbol{f}}$. $\tilde{\boldsymbol{f}}_{\boldsymbol{x}}$ and $\tilde{\mathcal{H}}_q$ can be immediately obtained from RFM. Note that even when we perform the next $(q+2)$-th point selection, we can reuse $\tilde{\mathcal{F}}_S^*$ and $\tilde{\boldsymbol{f}}$, from which $\tilde{\mathcal{H}}_{q+1}$ can also be immediately obtained.

The MAP-based approximation can also be applied to the parallel setting. The lower bound (11) can be re-written as

$$\mathbb{E}_{\mathcal{F}^*, \mathcal{H}_q}\mathbb{E}_{\boldsymbol{f}_{\boldsymbol{x}}|\mathcal{F}^*, \mathcal{H}_q}\left[\mathbb{I}(\boldsymbol{f}_{\boldsymbol{x}} \in \mathcal{A}_O^{\mathcal{F}_S^*})\log\zeta_{\lambda,q}^{\mathcal{F}_S^*}(\boldsymbol{x})\right.$$

$$\left. + \mathbb{I}(\boldsymbol{f}_{\boldsymbol{x}} \in \mathcal{A}_{U\backslash O}^{\mathcal{F}_S^*})\log\eta_{\lambda,q}^{\mathcal{F}_S^*}(\boldsymbol{x})\right]$$

$$= \mathbb{E}_{\mathcal{F}^*, \mathcal{H}_q}\left[p(\boldsymbol{f}_{\boldsymbol{x}} \in \mathcal{A}_O^{\mathcal{F}_S^*} \mid \mathcal{F}^*, \mathcal{H}_q)\log\zeta_{\lambda,q}^{\mathcal{F}_S^*}(\boldsymbol{x})\right.$$

$$\left. + p(\boldsymbol{f}_{\boldsymbol{x}} \in \mathcal{A}_{U\backslash O}^{\mathcal{F}_S^*} \mid \mathcal{F}^*, \mathcal{H}_q)\log\eta_{\lambda,q}^{\mathcal{F}_S^*}(\boldsymbol{x})\}\right].$$

The prior approximation for $p(\boldsymbol{f}_{\boldsymbol{x}} \in \mathcal{A}_O^{\mathcal{F}_S^*} \mid \mathcal{F}^*, \mathcal{H}_q)$ becomes $p(\boldsymbol{f}_{\boldsymbol{x}} \in \mathcal{A}_O^{\mathcal{F}_S^*} \mid \mathcal{F}^*, \mathcal{H}_q) \approx Z_O^{\mathcal{F}_S^*}(\boldsymbol{x} \mid \mathcal{H}_q)/Z_U^{\mathcal{F}_S^*}(\boldsymbol{x} \mid \mathcal{H}_q)$. Then, $\theta_{\mathrm{MAP}}$ can be obtained by the same procedure shown in Section 3.3. As a result, we have an approximation of the lower bound (11)

$$\frac{1}{K}\sum_{(\tilde{\mathcal{F}}_S^*, \tilde{\boldsymbol{f}}_{\boldsymbol{x}}, \tilde{\mathcal{H}}_q)\in F_q}\theta_{\mathrm{MAP}}\log\zeta_{\lambda,q}^{\tilde{\mathcal{F}}_S^*}(\boldsymbol{x})$$

$$+ (1-\theta_{\mathrm{MAP}})\log\eta_{\lambda,q}^{\tilde{\mathcal{F}}_S^*}(\boldsymbol{x}).$$

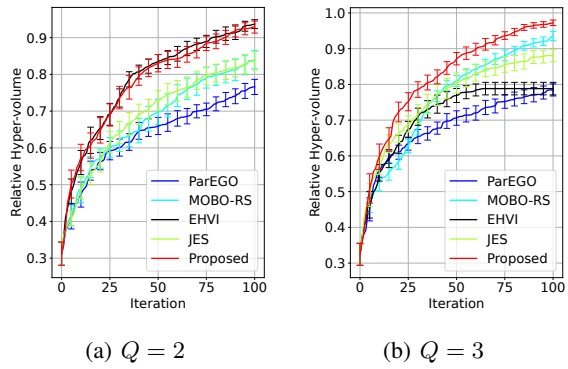

(a) $Q = 2$      (b) $Q = 3$

*Figure 11.* Results on parallel querying.

Figure 11 shows the empirical evaluation for which the setting is same as in Section 6.1. Here, we set $Q = 2$ and 3, and we used the $(d, L) = (3, 3)$ GP derived functions. For comparison, ParEGO, EHVI, MOBO-RS and JES were used. ParEGO and EHVI consider the expected improvement when $Q$ points are simultaneously selected, which is a well known general strategy (Shahriari et al., 2016). Unlike the single querying, the expectation is approximated by the MC estimation for which the number of samples was 100 and 10 for ParEGO and EHVI, respectively. For MOBO-RS, $Q$ points are selected by repeating Thompson sampling with different sample paths $Q$ times (Kandasamy et al., 2018), in which the weights in the Tchebycheff (also known as Chebyshev) scalarization were also re-sampled. JES approximates the simultaneous information gain by $Q$ points as indicated by (Tu et al., 2022). From the results, we can see that the proposed method shows sufficiently high performance in the parallel querying.

## F. Decoupled Setting

For the decoupled setting, MI and its lower bound are

$$\mathrm{MI}(f_{\boldsymbol{x}}^l; \mathcal{F}^*)$$

$$= \mathbb{E}_{\mathcal{F}^*, f_{\boldsymbol{x}}^l}\left[\log\frac{q(f_{\boldsymbol{x}}^l \mid \mathcal{F}^*)}{p(f_{\boldsymbol{x}}^l)}\right] + D_{\mathrm{KL}}\left(p(f_{\boldsymbol{x}}^l \mid \mathcal{F}^*) \parallel q(f_{\boldsymbol{x}}^l \mid \mathcal{F}^*)\right)$$

$$\geq \mathbb{E}_{\mathcal{F}^*, f_{\boldsymbol{x}}^l}\left[\log\frac{q(f_{\boldsymbol{x}}^l \mid \mathcal{F}^*)}{p(f_{\boldsymbol{x}}^l)}\right] =: L_{\mathrm{Dec}}(\boldsymbol{x}, l, \lambda).$$

We define $q(f_{\boldsymbol{x}}^l \mid \mathcal{F}^*)$ as the marginal distribution of $q_\lambda(\boldsymbol{f_x} \mid \mathcal{F}^*)$:

$$q(f_{\boldsymbol{x}}^l \mid \mathcal{F}^*)$$

$$= \int q_\lambda(\boldsymbol{f_x} \mid \mathcal{F}^*)\mathrm{d}\boldsymbol{f_x}^{-l}$$

$$= \int \lambda q_U(\boldsymbol{f_x} \mid \mathcal{F}_S^*) + (1-\lambda)q_O(\boldsymbol{f_x} \mid \mathcal{F}_S^*)\mathrm{d}\boldsymbol{f_x}^{-l}$$

$$= \int p(\boldsymbol{f_x})\zeta_\lambda^{\mathcal{F}_S^*}(\boldsymbol{x})\mathbb{I}(\boldsymbol{f_x} \in \mathcal{A}_O^{\mathcal{F}_S^*})\mathrm{d}\boldsymbol{f_x}^{-l}$$

$$\qquad + \int p(\boldsymbol{f_x})\eta_\lambda^{\mathcal{F}_S^*}(\boldsymbol{x})\mathbb{I}(\boldsymbol{f_x} \in \mathcal{A}_{U \backslash O}^{\mathcal{F}_S^*})\mathrm{d}\boldsymbol{f_x}^{-l}$$

$$= p(f_{\boldsymbol{x}}^l)\Big\{\zeta_\lambda^{\mathcal{F}_S^*}(\boldsymbol{x}) \int p(\boldsymbol{f_x}^{-l} \mid f_{\boldsymbol{x}}^l)\mathbb{I}(\boldsymbol{f_x} \in \mathcal{A}_O^{\mathcal{F}_S^*})\mathrm{d}\boldsymbol{f_x}^{-l}$$

$$\qquad + \eta_\lambda^{\mathcal{F}_S^*}(\boldsymbol{x}) \int p(\boldsymbol{f_x}^{-l} \mid f_{\boldsymbol{x}}^l)\mathbb{I}(\boldsymbol{f_x} \in \mathcal{A}_{U \backslash O}^{\mathcal{F}_S^*})\mathrm{d}\boldsymbol{f_x}^{-l}\Big\}$$

$$= p(f_{\boldsymbol{x}}^l)\Big\{\zeta_\lambda^{\mathcal{F}_S^*}(\boldsymbol{x})p(\boldsymbol{f_x} \in \mathcal{A}_O^{\mathcal{F}_S^*} \mid f_{\boldsymbol{x}}^l)$$

$$\qquad + \eta_\lambda^{\mathcal{F}_S^*}(\boldsymbol{x})p(\boldsymbol{f_x} \in \mathcal{A}_{U \backslash O}^{\mathcal{F}_S^*} \mid f_{\boldsymbol{x}}^l)\Big\}, \qquad (12)$$

where $\boldsymbol{f_x}^{-l}$ is the $(L-1)$-dimensional subvector of $\boldsymbol{f_x}$ in which $f_{\boldsymbol{x}}^l$ is removed. Then, the lower bound and its MC approximation is obtained as

$$L_{\mathrm{Dec}}(\boldsymbol{x}, l, \lambda)$$

$$= \mathbb{E}_{\mathcal{F}^*, f_{\boldsymbol{x}}^l}\Bigg[\log\Big\{\zeta_\lambda^{\mathcal{F}_S^*}(\boldsymbol{x})p(\boldsymbol{f_x} \in \mathcal{A}_O^{\mathcal{F}_S^*} \mid f_{\boldsymbol{x}}^l)$$

$$\qquad + \eta_\lambda^{\mathcal{F}_S^*}(\boldsymbol{x})p(\boldsymbol{f_x} \in \mathcal{A}_{U \backslash O}^{\mathcal{F}_S^*} \mid f_{\boldsymbol{x}}^l)\Big\}\Bigg]$$

$$\approx \frac{1}{K} \sum_{(\tilde{\mathcal{F}}_S^*, \tilde{\boldsymbol{f}}) \in F} \log\Big\{\zeta_\lambda^{\tilde{\mathcal{F}}_S^*}(\boldsymbol{x})p(\boldsymbol{f_x} \in \mathcal{A}_O^{\tilde{\mathcal{F}}_S^*} \mid \tilde{f}_{\boldsymbol{x}}^l)$$

$$\qquad + \eta_\lambda^{\tilde{\mathcal{F}}_S^*}(\boldsymbol{x})p(\boldsymbol{f_x} \in \mathcal{A}_{U \backslash O}^{\tilde{\mathcal{F}}_S^*} \mid \tilde{f}_{\boldsymbol{x}}^l)\Big\}.$$

The probabilities $p(\boldsymbol{f_x} \in \mathcal{A}_O^{\tilde{\mathcal{F}}_S^*} \mid \tilde{f}_{\boldsymbol{x}}^l)$ and $p(\boldsymbol{f_x} \in \mathcal{A}_{U \backslash O}^{\tilde{\mathcal{F}}_S^*} \mid \tilde{f}_{\boldsymbol{x}}^l)$ can be analytically calculated. Let $\mathcal{C}_i = (\ell_i^1, u_i^1] \times (\ell_i^2, u_i^2] \times \cdots \times (\ell_i^L, u_i^L]$ be the cell partitions of the over-truncated region $\mathcal{A}_O^{\tilde{\mathcal{F}}_S^*}$. Then, from the independence of $\boldsymbol{f_x}$, we see

$$p(\boldsymbol{f_x} \in \mathcal{A}_O^{\tilde{\mathcal{F}}_S^*} \mid \tilde{f}_{\boldsymbol{x}}^l)$$

$$= \int p(\boldsymbol{f_x}^{-l})\mathbb{I}(\boldsymbol{f_x} \in \mathcal{A}_O^{\tilde{\mathcal{F}}_S^*})\mathrm{d}\boldsymbol{f_x}^{-l}$$

$$= \sum_{c=1}^C \mathbb{I}(f_{\boldsymbol{x}}^l \in (\ell_c^l, u_c^l]) \int_{\mathcal{C}_c^{-l}} p(\boldsymbol{f_x}^{-l})\mathrm{d}\boldsymbol{f_x}^{-l}$$

$$= \sum_{c=1}^C \mathbb{I}(f_{\boldsymbol{x}}^l \in (\ell_c^l, u_c^l]) \prod_{l' \neq l} \big(\Phi(\bar{\alpha}_{c,l'}) - \Phi(\underline{\alpha}_{c,l'})\big)$$

$$= \sum_{c=1}^C \mathbb{I}(f_{\boldsymbol{x}}^l \in (\ell_c^l, u_c^l]) \prod_{l' \neq l} Z_{cl'}, \qquad (13)$$

where $Z_{cl'} = \Phi(\bar{\alpha}_{c,l'}) - \Phi(\underline{\alpha}_{c,l'})$ and $\mathcal{C}_c^{-l}$ is the $(L-1)$-dimensional hyper-rectangle created by removing the $l$-th dimension of $\mathcal{C}_c$. For $p(\boldsymbol{f_x} \in \mathcal{A}_{U \backslash O}^{\tilde{\mathcal{F}}_S^*} \mid \tilde{f}_{\boldsymbol{x}}^l)$, we use the following relation:

$$p(\boldsymbol{f_x} \in \mathcal{A}_{U \backslash O}^{\tilde{\mathcal{F}}_S^*} \mid \tilde{f}_{\boldsymbol{x}}^l)$$

$$= 1 - p(\boldsymbol{f_x} \in \mathcal{A}_O^{\tilde{\mathcal{F}}_S^*} \mid \tilde{f}_{\boldsymbol{x}}^l) - p(\boldsymbol{f_x} \notin \mathcal{A}_U^{\tilde{\mathcal{F}}_S^*} \mid \tilde{f}_{\boldsymbol{x}}^l).$$

Since $\boldsymbol{f_x} \notin \mathcal{A}_U^{\tilde{\mathcal{F}}_S^*}$ in the last term can be written as "dominated region", shown in the end of Appendix B. We can calculate $p(\boldsymbol{f_x} \notin \mathcal{A}_U^{\tilde{\mathcal{F}}_S^*} \mid \tilde{f}_{\boldsymbol{x}}^l)$ by the same dominated region decomposition as (13).

Figure 12 shows the empirical evaluation for which the setting is same as in Section 6.1. Here, we used the $(d, L) = (3, 3)$ GP derived functions for three length scales. As a baseline, the hypervolume-based KG is used. Since KG is defined as the hyper-volume defined by the 'one-step ahead' posterior mean, it is easy to extend to the decoupled setting as shown by (Daulton et al., 2023). We used BoTorch's qHypervolumeKnowledgeGradient implementation for the HVKG baseline, which is based on the 'one shot' optimization of KG in the decoupled setting. Due to the long computational time, we set the number of Pareto-optimal points in the one-step ahead posterior mean as 10 in HVKG. For fair comparison, our proposed method also set the size of the Pareto-optimal points in the sampled function (i.e., the NSGA-II population size) as 10. The number of samples (so called fantasy points) in HVKG is also set as 10. For the final evaluation of the performance, since the decoupled setting observes only one objective function $f^l(\boldsymbol{x}_t)$ in each iteration, the hyper-volume consisting of observed points is difficult to define. Instead, we employ an approach similar to so-called inference regret. At each iteration $t$, we apply NSGA-II to the posterior mean $\boldsymbol{\mu}(\boldsymbol{x})$, and obtain a set of the Pareto-optimal points $\mathcal{X}^*$ for $\boldsymbol{\mu}(\boldsymbol{x})$. We evaluate the hyper-volume defined by the ground-truth objective function values on $\mathcal{X}^*$, and the plots are the maximum values of the volumes identified until each iteration. From results, we see that the decoupled extension of PFEV has reasonable performance (Note that we only have about $1/3$ observations compared with the same number of iterations of the coupled setting because only one of the objective functions is observed). Due to the package differences (PFEV is based on GPy, while HVKG is BoTorch), this comparison is not a completely fair comparison. However, it demonstrates that the proposed method can achieve comparable or superior performance to a standard package. A more fair and thorough comparison is our future work.

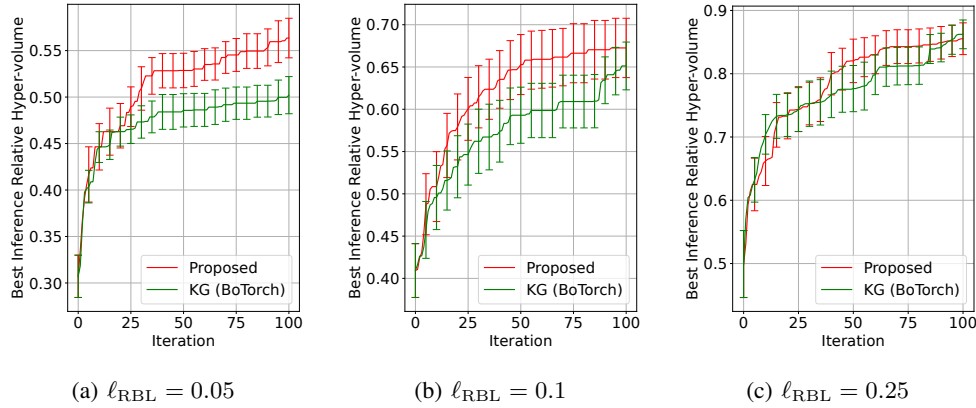

(a) $\ell_{\mathrm{RBL}} = 0.05$         (b) $\ell_{\mathrm{RBL}} = 0.1$         (c) $\ell_{\mathrm{RBL}} = 0.25$

*Figure 12.* Results on decoupled setting.

## G. Joint Entropy Extension

Let $\mathcal{X}^* = \{\boldsymbol{x} \mid \boldsymbol{f_x} \in \mathcal{F}^*\}$. The lower bound is

$$\mathrm{MI}(\boldsymbol{f_x}; \mathcal{X}^*, \mathcal{F}^*) \geq \mathbb{E}_{\mathcal{X}^*, \mathcal{F}^*, \boldsymbol{f_x}}\left[\log \frac{q(\boldsymbol{f_x} \mid \mathcal{X}^*, \mathcal{F}^*)}{p(\boldsymbol{f_x})}\right].$$

In practice, we only obtain a subset $\mathcal{X}_S^* \subseteq \mathcal{X}^*$, by which $\mathcal{F}_S^* = \{\boldsymbol{f_x}\}_{\boldsymbol{x} \in \mathcal{X}_S^*}$ and $\mathcal{H}_S = \{(\boldsymbol{x}, \boldsymbol{f_x})\}_{\boldsymbol{x} \in \mathcal{X}_S^*}$ are defined. In the variational distribution $q(\boldsymbol{f_x} \mid \mathcal{X}^*, \mathcal{F}^*)$, $\mathcal{H}_S$ can be seen as an additional training data of the GPs $\boldsymbol{f_x}$. Then, a natural extension of the variational distribution (3) is

$$
\begin{aligned}
&q_\lambda(\boldsymbol{f_x} \mid \mathcal{X}_S^*, \mathcal{F}_S^*) \\
&= \begin{cases}
\left(\frac{\lambda}{Z_U^{\mathcal{F}_S^*}(\boldsymbol{x}|\mathcal{H}_S)} + \frac{1-\lambda}{Z_O^{\mathcal{F}_S^*}(\boldsymbol{x}|\mathcal{H}_S)}\right) p(\boldsymbol{f_x} \mid \mathcal{H}_S) & \text{if } \boldsymbol{f_x} \in \mathcal{A}_O^{\mathcal{F}_S^*}, \\
\frac{\lambda}{Z_U^{\mathcal{F}_S^*}(\boldsymbol{x}|\mathcal{H}_S)} p(\boldsymbol{f_x} \mid \mathcal{H}_S) & \text{if } \boldsymbol{f_x} \in \mathcal{A}_{U \setminus O}^{\mathcal{F}_S^*}, \\
0 & \text{otherwise},
\end{cases}
\end{aligned}
$$

where $Z_U^{\mathcal{F}_S^*}(\boldsymbol{x} \mid \mathcal{H}_S) := p(\boldsymbol{f_x} \in \mathcal{A}_U^{\mathcal{F}_S^*} \mid \mathcal{H}_S)$ and $Z_O^{\mathcal{F}_S^*}(\boldsymbol{x} \mid \mathcal{H}_S) := p(\boldsymbol{f_x} \in \mathcal{A}_O^{\mathcal{F}_S^*} \mid \mathcal{H}_S)$. Note that here the conditioning on $\mathcal{H}_S$ is interpreted as a simple addition of the training data, and does not impose the conditions that "$\mathcal{H}_S$ is the optimal solutions", which is represented by the truncation (Because of this reason, we do not give '*' to $\mathcal{H}_S$). The resulting lower bound is

$$
\begin{aligned}
\mathbb{E}_{\mathcal{X}^*, \mathcal{F}^*, \boldsymbol{f_x}}\Big[&\log\big\{\zeta_\lambda^{\mathcal{F}_S^*}(\boldsymbol{x} \mid \mathcal{H}_S)\mathbb{I}(\boldsymbol{f_x} \in \mathcal{A}_O^{\mathcal{F}_S^*}) \\
&+ \eta_\lambda^{\mathcal{F}_S^*}(\boldsymbol{x} \mid \mathcal{H}_S)\mathbb{I}(\boldsymbol{f_x} \in \mathcal{A}_{U \setminus O}^{\mathcal{F}_S^*})\big\}\Big], \quad (14)
\end{aligned}
$$

where $\zeta_\lambda^{\mathcal{F}_S^*}(\boldsymbol{x} \mid \mathcal{H}_S) = \frac{\lambda}{Z_U^{\mathcal{F}_S^*}(\boldsymbol{x}|\mathcal{H}_S)} + \frac{1-\lambda}{Z_O^{\mathcal{F}_S^*}(\boldsymbol{x}|\mathcal{H}_S)}$ and $\eta_\lambda^{\mathcal{F}_S^*}(\boldsymbol{x} \mid \mathcal{H}_S) = \frac{\lambda}{Z_U^{\mathcal{F}_S^*}(\boldsymbol{x}|\mathcal{H}_S)}$. The same MC approximation as (6) can be used to evaluate (14).

The MAP-based approximation is also possible based on transformation of (14):

$$
\begin{aligned}
\mathbb{E}_{\mathcal{X}^*, \mathcal{F}^*}\Big[&p(\boldsymbol{f_x} \in \mathcal{A}_O^{\mathcal{F}_S^*} \mid \mathcal{X}^*, \mathcal{F}^*) \log \zeta_\lambda^{\mathcal{F}_S^*}(\boldsymbol{x} \mid \mathcal{H}_S) \\
&+ p(\boldsymbol{f_x} \in \mathcal{A}_{U \setminus O}^{\mathcal{F}_S^*} \mid \mathcal{X}^*, \mathcal{F}^*) \log \eta_\lambda^{\mathcal{F}_S^*}(\boldsymbol{x} \mid \mathcal{H}_S)\Big].
\end{aligned}
$$

Based on the same idea shown in Section 3.3, we approximate $p(\boldsymbol{f_x} \in \mathcal{A}_O^{\mathcal{F}_S^*} \mid \mathcal{X}^*, \mathcal{F}^*) \approx Z_O^{\mathcal{F}_S^*}(\boldsymbol{x} \mid \mathcal{H}_S)/Z_U^{\mathcal{F}_S^*}(\boldsymbol{x} \mid \mathcal{H}_S)$, from which the MAP estimator can be defined.

Figure 13 shows the empirical evaluation for which the setting is same as in Section 6.1. Here, we used the $(d, L) = (3, 4)$ GP derived functions for three length scales. We see that the original proposed method shows slightly better performance than the JES extension, though their behaviors are similar.

## H. Noisy Observation

Let $y_{\boldsymbol{x}}^l = f_{\boldsymbol{x}}^l + \varepsilon$ and $\boldsymbol{y_x} = (y_{\boldsymbol{x}}^1, \ldots, y_{\boldsymbol{x}}^L)^\top$, where $\varepsilon \sim \mathcal{N}(0, \sigma_{\mathrm{noise}}^2)$. The mutual information for noisy observation is $\mathrm{MI}(\boldsymbol{y_x}; \mathcal{F}^*)$, for which the lower bound can be derived as

$$
\begin{aligned}
&\mathrm{MI}(\boldsymbol{y_x}; \mathcal{F}^*) \\
&= \mathbb{E}_{\mathcal{F}^*, \boldsymbol{y_x}}\left[\log \frac{q(\boldsymbol{y_x} \mid \mathcal{F}^*)}{p(\boldsymbol{y_x})}\right] \\
&\quad + D_{\mathrm{KL}}\left(p(\boldsymbol{y_x} \mid \mathcal{F}^*) \,\|\, q(\boldsymbol{y_x} \mid \mathcal{F}^*)\right) \\
&\geq \mathbb{E}_{\mathcal{F}^*, \boldsymbol{y_x}}\left[\log \frac{q(\boldsymbol{y_x} \mid \mathcal{F}^*)}{p(\boldsymbol{y_x})}\right].
\end{aligned}
$$

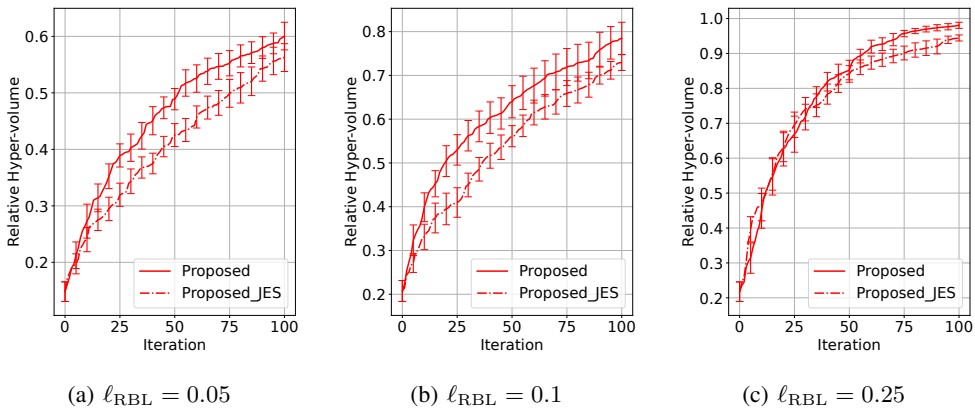

(a) $\ell_{\mathrm{RBL}} = 0.05$      (b) $\ell_{\mathrm{RBL}} = 0.1$      (c) $\ell_{\mathrm{RBL}} = 0.25$

*Figure 13.* Comparison with JES extension of proposed method.

Since $\mathcal{F}^*$ and $\boldsymbol{y_x}$ are independent if $\boldsymbol{f_x}$ is given, we define $q(\boldsymbol{y_x} \mid \mathcal{F}^*)$ by using $q_\lambda(\boldsymbol{f_x} \mid \mathcal{F}^*)$ as follows.

$$
\begin{aligned}
q(&\boldsymbol{y_x} \mid \mathcal{F}^*) \\
&= \int p(\boldsymbol{y_x} \mid \boldsymbol{f_x}) q_\lambda(\boldsymbol{f_x} \mid \mathcal{F}^*) \mathrm{d}\boldsymbol{f_x} \\
&= \int p(\boldsymbol{y_x} \mid \boldsymbol{f_x})(\lambda q_U(\boldsymbol{f_x} \mid \mathcal{F}_S^*) \\
&\qquad + (1-\lambda) q_O(\boldsymbol{f_x} \mid \mathcal{F}_S^*)) \mathrm{d}\boldsymbol{f_x} \\
&= \int p(\boldsymbol{y_x} \mid \boldsymbol{f_x}) p(\boldsymbol{f_x})\{\zeta_\lambda^{\mathcal{F}_S^*}(\boldsymbol{x})\mathbb{I}(\boldsymbol{f_x} \in \mathcal{A}_O^{\mathcal{F}_S^*}) \\
&\qquad + \eta_\lambda^{\mathcal{F}_S^*}(\boldsymbol{x})\mathbb{I}(\boldsymbol{f_x} \in \mathcal{A}_{U\setminus O}^{\mathcal{F}_S^*})\} \mathrm{d}\boldsymbol{f_x} \\
&= \int_{\boldsymbol{f_x} \in \mathcal{A}_O^{\mathcal{F}_S^*}} p(\boldsymbol{y_x}, \boldsymbol{f_x})\zeta_\lambda^{\mathcal{F}_S^*}(\boldsymbol{x})\mathrm{d}\boldsymbol{f_x} \\
&\qquad + \int_{\boldsymbol{f_x} \in \mathcal{A}_{U\setminus O}^{\mathcal{F}_S^*}} p(\boldsymbol{y_x}, \boldsymbol{f_x})\eta_\lambda^{\mathcal{F}_S^*}(\boldsymbol{x})\mathrm{d}\boldsymbol{f_x} \\
&= p(\boldsymbol{y_x})\Big\{\zeta_\lambda^{\mathcal{F}_S^*}(\boldsymbol{x}) \int_{\boldsymbol{f_x} \in \mathcal{A}_O^{\mathcal{F}_S^*}} p(\boldsymbol{f_x} \mid \boldsymbol{y_x})\mathrm{d}\boldsymbol{f_x} \\
&\qquad + \eta_\lambda^{\mathcal{F}_S^*}(\boldsymbol{x}) \int_{\boldsymbol{f_x} \in \mathcal{A}_{U\setminus O}^{\mathcal{F}_S^*}} p(\boldsymbol{f_x} \mid \boldsymbol{y_x})\mathrm{d}\boldsymbol{f_x}\Big\} \\
&= p(\boldsymbol{y_x})\Big\{\zeta_\lambda^{\mathcal{F}_S^*}(\boldsymbol{x})p(\boldsymbol{f_x} \in \mathcal{A}_O^{\mathcal{F}_S^*} \mid \boldsymbol{y_x}) \\
&\qquad + \eta_\lambda^{\mathcal{F}_S^*}(\boldsymbol{x})p(\boldsymbol{f_x} \in \mathcal{A}_{U\setminus O}^{\mathcal{F}_S^*} \mid \boldsymbol{y_x})\Big\}.
\end{aligned}
$$

Then, the MC approximation becomes

$$
\begin{aligned}
\mathbb{E}_{\mathcal{F}^*, \boldsymbol{y_x}}&\Big[\log\Big\{\zeta_\lambda^{\mathcal{F}_S^*}(\boldsymbol{x})p(\boldsymbol{f_x} \in \mathcal{A}_O^{\mathcal{F}_S^*} \mid \boldsymbol{y_x}) \\
&\quad + \eta_\lambda^{\mathcal{F}_S^*}(\boldsymbol{x})p(\boldsymbol{f_x} \in \mathcal{A}_{U\setminus O}^{\mathcal{F}_S^*} \mid \boldsymbol{y_x})\Big\}\Big] \\
&\approx \frac{1}{KK_\varepsilon} \sum_{(\tilde{\mathcal{F}}_S^*, \tilde{\boldsymbol{f}}) \in F} \sum_{\tilde{\varepsilon} \in E} \log\Big\{\zeta_\lambda^{\tilde{\mathcal{F}}_S^*}(\boldsymbol{x})p(\boldsymbol{f_x} \in \mathcal{A}_O^{\tilde{\mathcal{F}}_S^*} \mid \tilde{\boldsymbol{y}}_{\boldsymbol{x}}) \\
&\qquad + \eta_\lambda^{\tilde{\mathcal{F}}_S^*}(\boldsymbol{x})p(\boldsymbol{f_x} \in \mathcal{A}_{U\setminus O}^{\tilde{\mathcal{F}}_S^*} \mid \tilde{\boldsymbol{y}}_{\boldsymbol{x}})\Big\}.
\end{aligned}
$$

where $E$ is a set of $L$-dimension noise samples $\tilde{\varepsilon}$ from $\mathcal{N}(0, \sigma_{\mathrm{noise}}^2 I)$, $\tilde{\boldsymbol{y}}_{\boldsymbol{x}} = \tilde{\boldsymbol{f}}_{\boldsymbol{x}} + \tilde{\varepsilon}$ is a sample of $\boldsymbol{y_x}$, and $K_\varepsilon = |E|$. Note that, in this approximation, we transform $\mathbb{E}_{\mathcal{F}^*, \boldsymbol{y_x}}$ to $\mathbb{E}_{\mathcal{F}^*, \boldsymbol{f_x}}\mathbb{E}_{\boldsymbol{\varepsilon}}$, which is possible because of the independence of the noise term. To evaluate this MC approximation, we need the conditional distribution $p(\boldsymbol{f_x} \mid \tilde{\boldsymbol{y}}_{\boldsymbol{x}})$, written as

$$
\boldsymbol{f_x} \mid \tilde{\boldsymbol{y}}_{\boldsymbol{x}} \sim \mathcal{N}(\boldsymbol{\nu}(\boldsymbol{x}), \mathrm{diag}(\boldsymbol{s}(\boldsymbol{x}))),
$$

where

$$
\begin{aligned}
\nu_l(\boldsymbol{x}) &= \mu_l(\boldsymbol{x}) + \frac{\sigma_l^2(\boldsymbol{x})}{\sigma_l^2(\boldsymbol{x}) + \sigma_{\mathrm{noise}}^2}(\tilde{y}_{\boldsymbol{x}}^l - \mu_l(\boldsymbol{x})), \\
s_l(\boldsymbol{x}) &= \sigma_l^2(\boldsymbol{x}) - \frac{\sigma_l^4(\boldsymbol{x})}{\sigma_l^2(\boldsymbol{x}) + \sigma_{\mathrm{noise}}^2}.
\end{aligned}
$$

Therefore, $p(\boldsymbol{f_x} \in \mathcal{A}_O^{\tilde{\mathcal{F}}_S^*} \mid \tilde{\boldsymbol{y}}_{\boldsymbol{x}})$ can be calculated by using the same procedure as $Z_O^{\tilde{\mathcal{F}}_S^*} = p(\boldsymbol{f_x} \in \mathcal{A}_O^{\tilde{\mathcal{F}}_S^*})$ shown in (9):

$$
\begin{aligned}
p(\boldsymbol{f_x} \in \mathcal{A}_O^{\tilde{\mathcal{F}}_S^*} \mid \tilde{\boldsymbol{y}}_{\boldsymbol{x}}) &= \sum_{i=1}^C \int_{C_i} p(\boldsymbol{f_x} \mid \tilde{\boldsymbol{y}}_{\boldsymbol{x}})\mathrm{d}\boldsymbol{f_x} \\
&= \sum_{i=1}^C \prod_{l=1}^L \int_{\ell_i^l}^{u_i^l} p(f_{\boldsymbol{x}}^l \mid \tilde{y}_{\boldsymbol{x}}^l)\mathrm{d}f_{\boldsymbol{x}}^l \\
&= \sum_{i=1}^C \prod_{l=1}^L \Big(\Phi(\bar{\beta}_{i,l}) - \Phi(\underline{\beta}_{i,l})\Big)
\end{aligned}
$$

where $\underline{\beta}_{i,l} = (\ell_i^l - \nu_l(\boldsymbol{x}))/s_l(\boldsymbol{x})$ and $\bar{\beta}_{i,l} = (u_i^l - \nu_l(\boldsymbol{x}))/s_l(\boldsymbol{x})$. For $p(\boldsymbol{f_x} \in \mathcal{A}_{U\setminus O}^{\tilde{\mathcal{F}}_S^*} \mid \tilde{\boldsymbol{y}}_{\boldsymbol{x}})$, we can use a relation $p(\boldsymbol{f_x} \in \mathcal{A}_{U\setminus O}^{\tilde{\mathcal{F}}_S^*} \mid \tilde{\boldsymbol{y}}_{\boldsymbol{x}}) = p(\boldsymbol{f_x} \in \mathcal{A}_U^{\tilde{\mathcal{F}}_S^*} \mid \tilde{\boldsymbol{y}}_{\boldsymbol{x}}) - p(\boldsymbol{f_x} \in \mathcal{A}_O^{\tilde{\mathcal{F}}_S^*} \mid \tilde{\boldsymbol{y}}_{\boldsymbol{x}})$. The last term $p(\boldsymbol{f_x} \in \mathcal{A}_O^{\tilde{\mathcal{F}}_S^*} \mid \tilde{\boldsymbol{y}}_{\boldsymbol{x}})$ can be calculated by the same approach as $Z_U^{\tilde{\mathcal{F}}_S^*} = p(\boldsymbol{f_x} \in \mathcal{A}_U^{\tilde{\mathcal{F}}_S^*})$, described in the end of Appendix B.

Figure 14 shows the empirical evaluation for which the setting is same as in Section 6.1. Here, we used the $(d, L) = (3, 3)$ GP derived functions for three length scales. We added the independent noise $\mathcal{N}(0, \sigma_{\text{noise}}^2)$ with $\sigma_{\text{noise}} = 0.1$ to all the observations. The GPs have the same value of the noise parameter $\sigma_{\text{noise}} = 0.1$. We compared with EHVI, MOBO-RS, and JES. EHVI and MOBO-RS can handle the noise just by incorporating it into the surrogate GPs. JES (Tu et al., 2022) considers the information gain from noisy observations. PFEV shows similar results to its noisy observation counterpart. We empirically see that $\mathrm{MI}(\boldsymbol{f}(\boldsymbol{x}); \mathcal{F}^*)$ sufficiently works even when the observation contains a moderate level of noise. The investigation under stronger noise is our future work.

## I. Additional Discussion on Related Work

Although our main focus is on information-theoretic approaches, here, other criteria are also reviewed. A classical approach is the scalarization that transforms multiple objective functions into a scalar value, among which ParEGO (Knowles, 2006) is a well-known method based on a random scalarization. However, the information of the Pareto-frontier may be lost by the scalarization. The standard expected improvement (EI) has been extended based on measuring the improvement of the hyper-volume called EHVI (expected hyper-volume improvement) (Emmerich, 2005; Shah & Ghahramani, 2016). The GP upper confidence bound (UCB) is another well-known general approach to BO (Srinivas et al., 2010). UCB based MOBO methods have been studied (Ponweiser et al., 2008; Zuluaga et al., 2013; 2016), but the setting of the confidence interval sometimes becomes practically difficult. SUR (Picheny, 2015) is based on the reduction of PI after the querying, which is computationally quite expensive. A hypervolume-based multi-objective extension of knowledge gradient (KG) is considered by (Daulton et al., 2023). Naïve computations of the hypervolume KG are computationally intractable, and several approximation and acceleration strategies have been studied. For example, so-called one-shot strategy transforms the nested optimization into simultaneous optimization which makes computation much faster, but the dimension of the acquisition function optimization becomes high.

We focus on the information-theoretic approach, which was first proposed for single-objective BO (Hennig & Schuler, 2012; Hernández-Lobato et al., 2014; Wang & Jegelka, 2017). Recently, (Cheng & Becker, 2024) proposed a different variational lower bound approach to single-objective BO, which is only for single-objective problems. A seminal work in the information-theoretic approach to MOBO is the Predictive Entropy Search for Multi-Objective Optimization (PESMO) (Hernandez-Lobato et al., 2016). PESMO defines an acquisition function through the entropy of the set of Pareto-optimal solutions $\boldsymbol{x}$, which is based on complicated approximation by expectation propagation (EP) (Minka, 2001). On the other hand, Belakaria et al. (2019) proposed using the individual max-value entropy of each objective function, called max-value entropy search for multi-objective optimization (MESMO). This largely simplifies the computations, but obviously, information of the Pareto-frontier is lost. Another JES based approach has been recently proposed (Fernández-Sánchez & Hernández-Lobato, 2024), which is in a more general formulation including multi-fidelity and constrained problems. Their computations are based on the Gaussian based entropy approximation, whose validity remains unclear.

## J. Detail of Experimental Settings

Bayesian optimization was implemented by a Python package called GPy (GPy, since 2012). PFEV, PFES, and $\{PF\}^2$ES require $\mathcal{C}_i$ that are hyper-rectangles decomposing the dominated region. To obtain $\mathcal{C}_i$, we used quick hypervolume (QHV) (Russo & Francisco, 2014), which is an efficient recursive algorithm originally proposed for the Pareto hyper-volume calculation. In PFEV, we maximize $\lambda$ for each given $\boldsymbol{x}$ by calculating $\widehat{\mathrm{LB}}(\boldsymbol{x}, \lambda)$ for 11 grid points of $\lambda$ ($10^{-3}, 0.1, 0.2, \ldots, 1.0$). In RFM used for sampling $\mathcal{F}^*$ (required in PFEV, PFES, MESMO, $\{PF\}^2$ES, and JES), the number of bases $D$ was 500. The GP hyper-parameter $\sigma_{\text{noise}}^2$ is fixed as $10^{-4}$. In ParEGO, the coefficient parameter in the augmented Tchebycheff function was set $0.05$ as shown in (Knowles, 2006). In EHVI, the two reference points are required, shown as $\boldsymbol{v}_{\text{ref}}$ and $\boldsymbol{w}_{\text{ref}}$ in (Shah & Ghahramani, 2016). The worst point vector $\boldsymbol{v}_{\text{ref}}$ is defined by subtracting $10^{-4}$ from the vector consisting of the minimum value of each dimension of $\boldsymbol{y}_i$ in the training data. On the other hand, the ideal point vector $\boldsymbol{w}_{\text{ref}}$ is defined by adding 1 to the vector consisting of the maximum value of each dimension of $\boldsymbol{y}_i$.

The definition of each benchmark function is as follows.

- Fonseca-Fleming (Fonseca & Fleming, 1995)

$$f_1(\boldsymbol{x}) = 1 - \exp\left[-\sum_{i=1}^{d}\left(x_i - \frac{1}{\sqrt{d}}\right)^2\right]$$

$$f_2(\boldsymbol{x}) = 1 - \exp\left[-\sum_{i=1}^{d}\left(x_i + \frac{1}{\sqrt{d}}\right)^2\right]$$

subject to $\quad -4 \leq x_i \leq 4, \quad i \in [d]$.

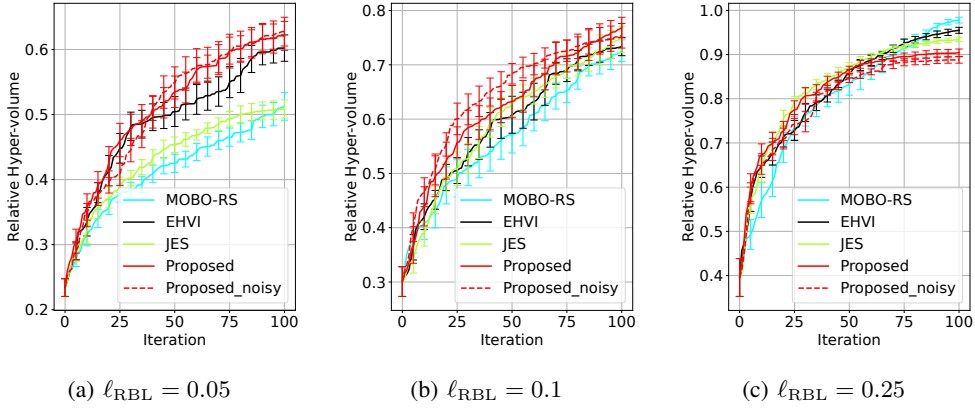

(a) $\ell_{\mathrm{RBL}} = 0.05$      (b) $\ell_{\mathrm{RBL}} = 0.1$      (c) $\ell_{\mathrm{RBL}} = 0.25$

*Figure 14.* Results with noisy observations in GP-derived synthetic functions (average and standard error of 10 runs)

- Kursawe (Kursawe, 1990)

$$f_1(\boldsymbol{x}) = \sum_{i=1}^{2} \left[ -10 \exp\left( -0.2 \sqrt{x_i^2 + x_{i+1}^2} \right) \right]$$

$$f_2(\boldsymbol{x}) = \sum_{i=1}^{3} \left[ |x_i|^{0.8} + 5 \sin(x_i^3) \right]$$

subject to $\quad -5 \le x_i \le 5, \quad 1 \le i \le 3.$

- Viennet (Vlennet et al., 1996)

$$f_1(x,y) = 0.5\left( x^2 + y^2 \right) + \sin\left( x^2 + y^2 \right)$$

$$f_2(x,y) = \frac{(3x - 2y + 4)^2}{8} + \frac{(x - y + 1)^2}{27} + 15$$

$$f_3(x,y) = \frac{1}{x^2 + y^2 + 1} - 1.1 \exp\left( -(x^2 + y^2) \right)$$

subject to $\quad -3 \le x, y \le 3.$

- FES1 (Fieldsend et al., 2003)

$$f_1(\boldsymbol{x}) = \sum_{i=1}^{d} \left| x_i - \exp\left( \frac{(i/d)^2}{3} \right) \right|^{0.5}$$

$$f_2(\boldsymbol{x}) = \sum_{i=1}^{d} \left( x_i - 0.5 \cos\left( \frac{10\pi i}{d} \right) - 0.5 \right)^2$$

subject to $\quad 0 \le x_i \le 1, \quad i \in [d].$

- FES2 (Fieldsend et al., 2003)

$$f_1(\boldsymbol{x}) = \sum_{i=1}^{d} \left( x_i - 0.5 \cos\left( \frac{10\pi i}{d} \right) - 0.5 \right)^2$$

$$f_2(\boldsymbol{x}) = \sum_{i=1}^{d} \left| x_i - \sin^2(i-1) \cos^2(i-1) \right|^{0.5}$$

$$f_3(\boldsymbol{x}) = \sum_{i=1}^{d} \left| x_i - 0.25 \cos(i-1) \cos(2i-2) - 0.5 \right|^{0.5}$$

subject to $\quad 0 \le x_i \le 1, \quad i \in [d].$

- FES3 (Fieldsend et al., 2003)

$$f_1(\boldsymbol{x}) = -\sum_{i=1}^{d} \left| x_i - \frac{\exp\left( \left( \frac{i}{d} \right)^2 \right)}{3} \right|^{0.5},$$

$$f_2(\boldsymbol{x}) = -\sum_{i=1}^{d} \left| x_i - \sin(i-1)^2 \cos(i-1)^2 \right|^{0.5},$$

$$f_3(\boldsymbol{x}) = -\sum_{i=1}^{d} | x_i - (0.25 \cos(i-1) \cos(2i-1) \\ - 0.5)|^{0.5},$$

$$f_4(\boldsymbol{x}) = -\sum_{i=1}^{d} \left( x_i - \frac{1}{2} \sin\left( 1000\pi \frac{i}{d} \right) - \frac{1}{2} \right)^2$$

subject to $\quad 0 \le x_i \le 1, \quad i \in [d].$

## K. Additional Results of Empirical Evaluation

### K.1. Additional Results on GP-derived Synthetic Functions

Additional results on GP-derived synthetic functions are shown in Fig. 15-17. The results are all combinations of $\ell_{\mathrm{RBF}} \in \{0.05, 0.1, 0.25\}$, $L \in \{2, 3, 4, 5, 6\}$, and $d \in \{2, 3, 4\}$ (Note that the results in the main text Fig. 3 are also included). We here also show results of ParEGO, MOBO-RS, and MESMO, which are omitted in the main text. The same format of boxplots as Fig 4 created from all the results are shown in Fig. 18. We also examined an input dimension wise boxplot in Fig. 19 in which differences of all the methods gradually decrease with the increase of the input dimension. We further plot separately for each length scale setting of the true objective function in Fig 20. We see that, particularly for small length scale problems (which tends to be multi-modal functions), differences become small. We speculate that the differences were less

apparent for challenging problems with high-dimensional and highly multi-modal functions.

Further, Fig. 21 is performance difference between Bayes MAP (8) and naïve MC (6).

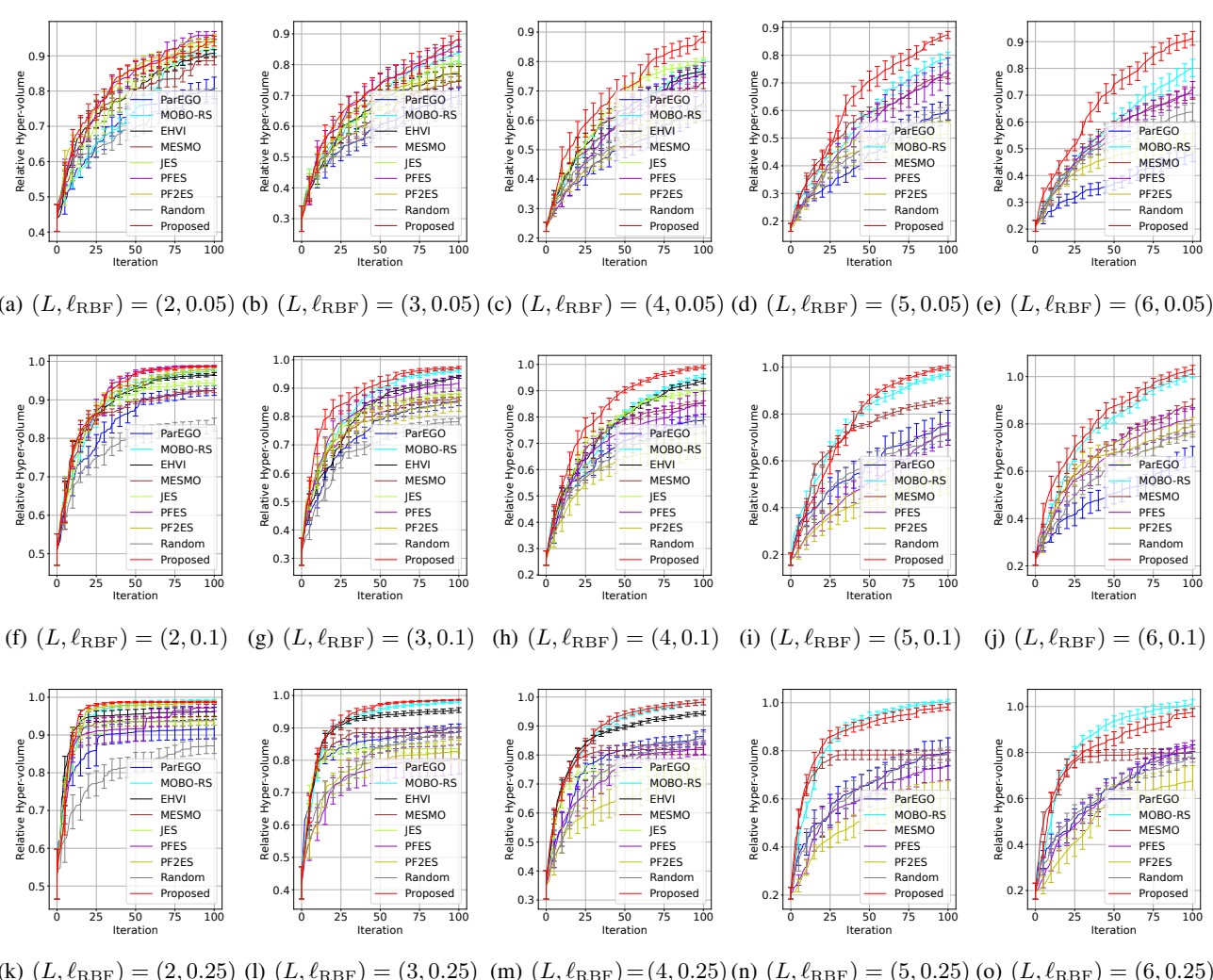

*Figure 15.* Results on GP-derived synthetic function ($d = 2$)

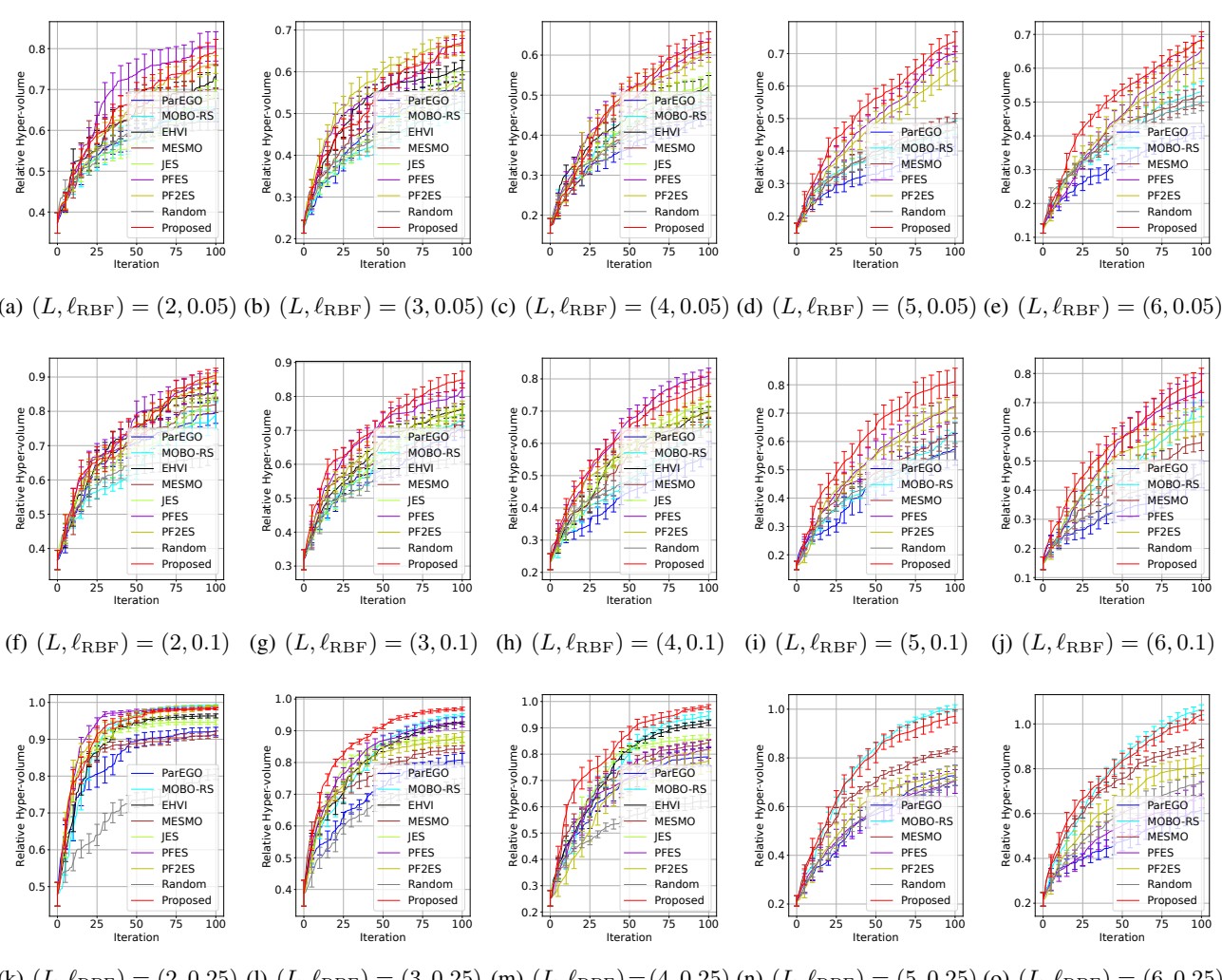

(a) $(L, \ell_{\mathrm{RBF}}) = (2, 0.05)$ (b) $(L, \ell_{\mathrm{RBF}}) = (3, 0.05)$ (c) $(L, \ell_{\mathrm{RBF}}) = (4, 0.05)$ (d) $(L, \ell_{\mathrm{RBF}}) = (5, 0.05)$ (e) $(L, \ell_{\mathrm{RBF}}) = (6, 0.05)$

(f) $(L, \ell_{\mathrm{RBF}}) = (2, 0.1)$ (g) $(L, \ell_{\mathrm{RBF}}) = (3, 0.1)$ (h) $(L, \ell_{\mathrm{RBF}}) = (4, 0.1)$ (i) $(L, \ell_{\mathrm{RBF}}) = (5, 0.1)$ (j) $(L, \ell_{\mathrm{RBF}}) = (6, 0.1)$

(k) $(L, \ell_{\mathrm{RBF}}) = (2, 0.25)$ (l) $(L, \ell_{\mathrm{RBF}}) = (3, 0.25)$ (m) $(L, \ell_{\mathrm{RBF}}) = (4, 0.25)$ (n) $(L, \ell_{\mathrm{RBF}}) = (5, 0.25)$ (o) $(L, \ell_{\mathrm{RBF}}) = (6, 0.25)$

*Figure 16.* Results on GP-derived synthetic function ($d = 3$)

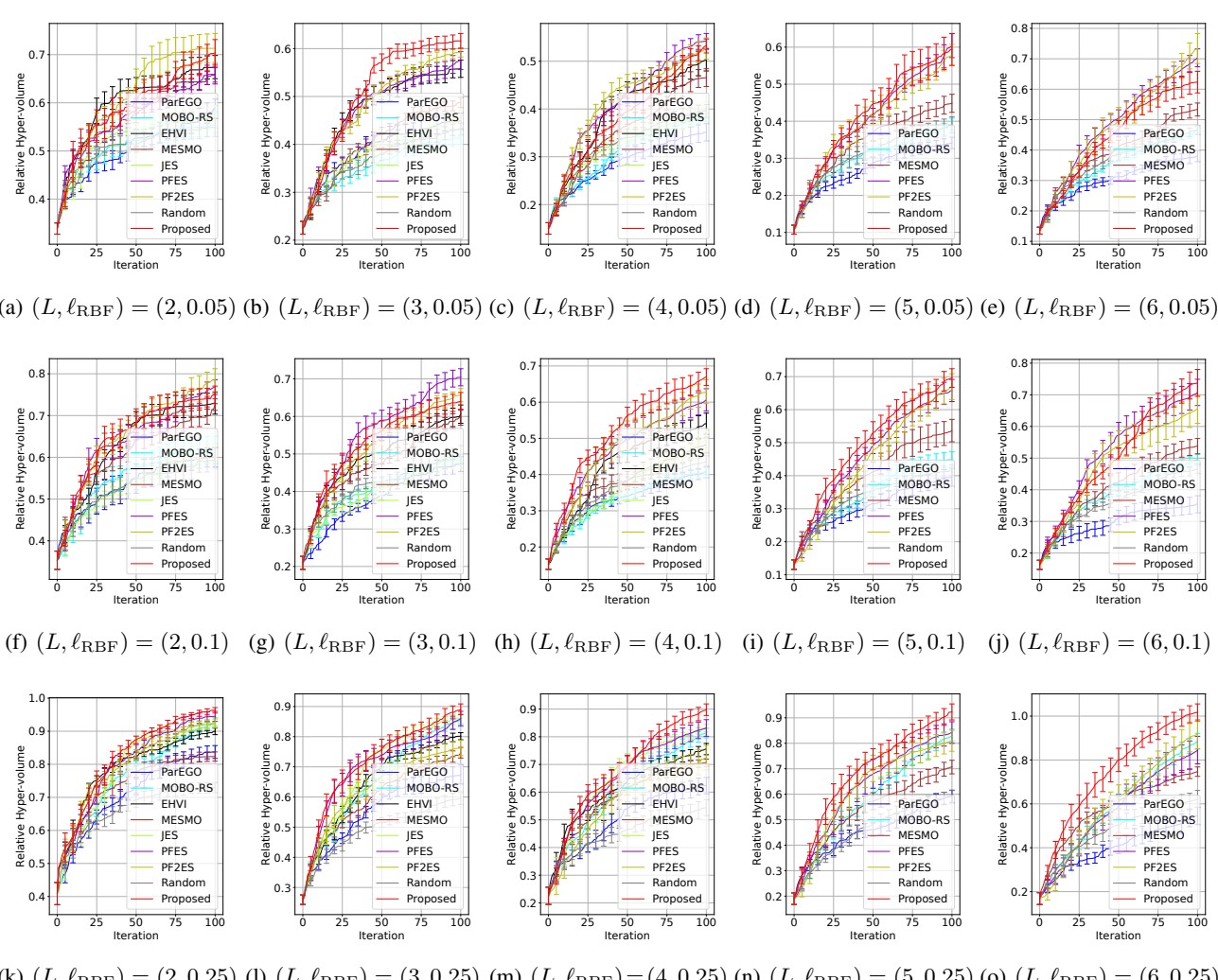

(a) $(L, \ell_{\mathrm{RBF}}) = (2, 0.05)$ (b) $(L, \ell_{\mathrm{RBF}}) = (3, 0.05)$ (c) $(L, \ell_{\mathrm{RBF}}) = (4, 0.05)$ (d) $(L, \ell_{\mathrm{RBF}}) = (5, 0.05)$ (e) $(L, \ell_{\mathrm{RBF}}) = (6, 0.05)$

(f) $(L, \ell_{\mathrm{RBF}}) = (2, 0.1)$ (g) $(L, \ell_{\mathrm{RBF}}) = (3, 0.1)$ (h) $(L, \ell_{\mathrm{RBF}}) = (4, 0.1)$ (i) $(L, \ell_{\mathrm{RBF}}) = (5, 0.1)$ (j) $(L, \ell_{\mathrm{RBF}}) = (6, 0.1)$

(k) $(L, \ell_{\mathrm{RBF}}) = (2, 0.25)$ (l) $(L, \ell_{\mathrm{RBF}}) = (3, 0.25)$ (m) $(L, \ell_{\mathrm{RBF}}) = (4, 0.25)$ (n) $(L, \ell_{\mathrm{RBF}}) = (5, 0.25)$ (o) $(L, \ell_{\mathrm{RBF}}) = (6, 0.25)$

*Figure 17.* Results on GP-derived synthetic function ($d = 4$)

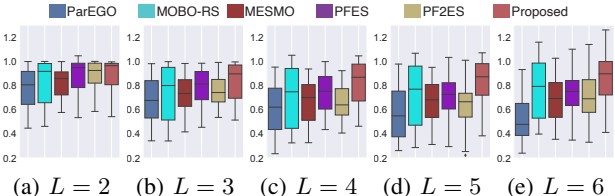

Figure 18. Boxplots of RHV at 100-th iteration.

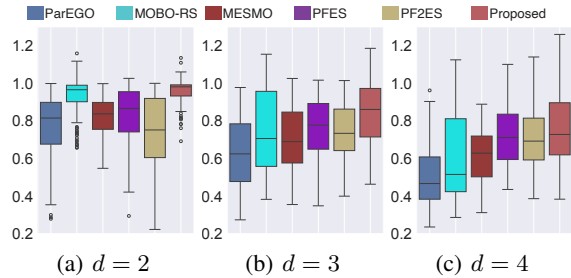

Figure 19. Boxplot of RHV at 100-th iteration for each input dimension.

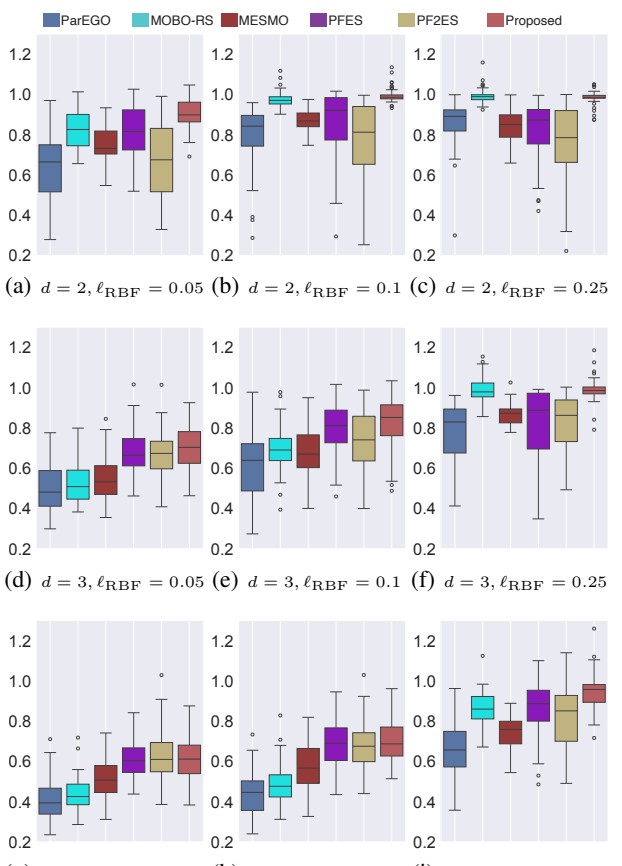

Figure 20. Boxplot of RHV at 100-th iteration for each input dimension and length scale.

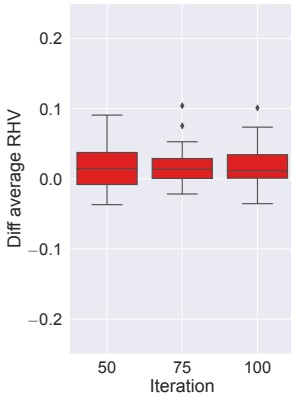

Figure 21. Boxplot of difference of average RHV between Bayes MAP and naïve MC (Bayes MAP minus naïve MC). The differences of the 10 runs average RHV are shown for BO iterations 50, 75, and 100. The GP-derived functions of $d \in \{2, 3, 4\}$, $L \in \{2, 3, 4\}$, and $\ell_{\mathrm{RBF}} \in \{0.05, 0.1, 0.25\}$ were used (i.e., $27 = 3 \times 3 \times 3$ points for each iteration). Since positive values indicate Bayes MAP is better than naïve MC, we see that Bayes MAP shows slightly better performance across iterations.

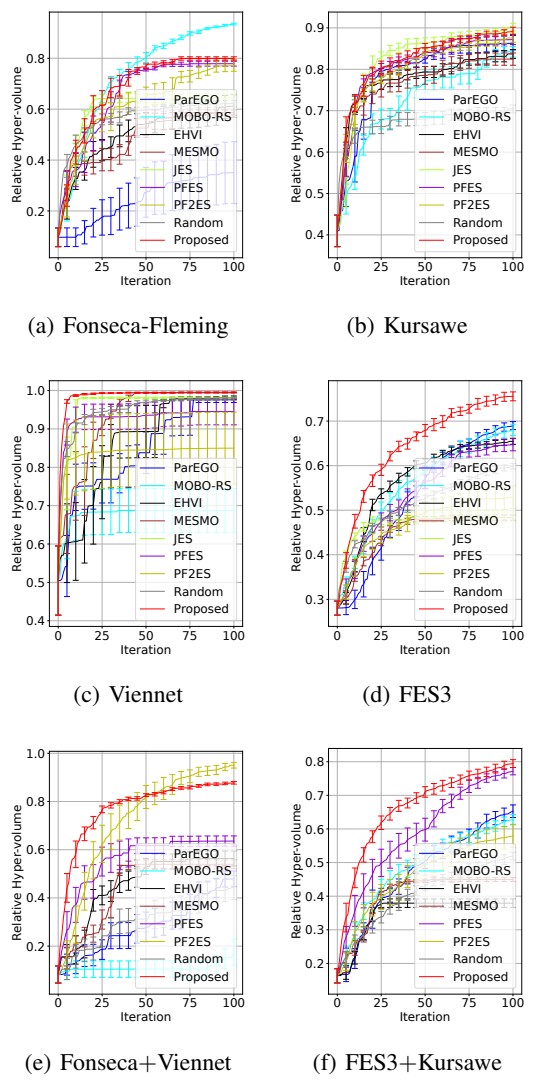

(a) Fonseca-Fleming        (b) Kursawe

(c) Viennet          (d) FES3

(e) Fonseca+Viennet     (f) FES3+Kursawe

*Figure 22.* Performance comparison in benchmark functions (average and standard deviation of 10 runs).

### K.2. Additional Results on Benchmark Functions

Figure 22 presents an extended version of Fig. 5 from the main text, including additional baseline methods (ParEGO, MOBO-RS, and MESMO). Figure 23 shows additional results on benchmark functions. FES1 and FES2 are $(d, L) = (3, 2)$ and $(d, L) = (3, 3)$, respectively. Therefore, FES1+Kursawe and FES2+Kursawe are $(d, L) = (3, 4)$ and $(d, L) = (3, 5)$, respectively. The experimental setup is identical to that in Section 6.1. These additional results further demonstrate our method's superior performance across various problem configurations.

To assess the scalability of the proposed method, we extend our experiments to higher input dimensions. Figures 24, 25, and 26 display the results for FES1, FES2, and FES3 benchmark functions with input dimensions ranging

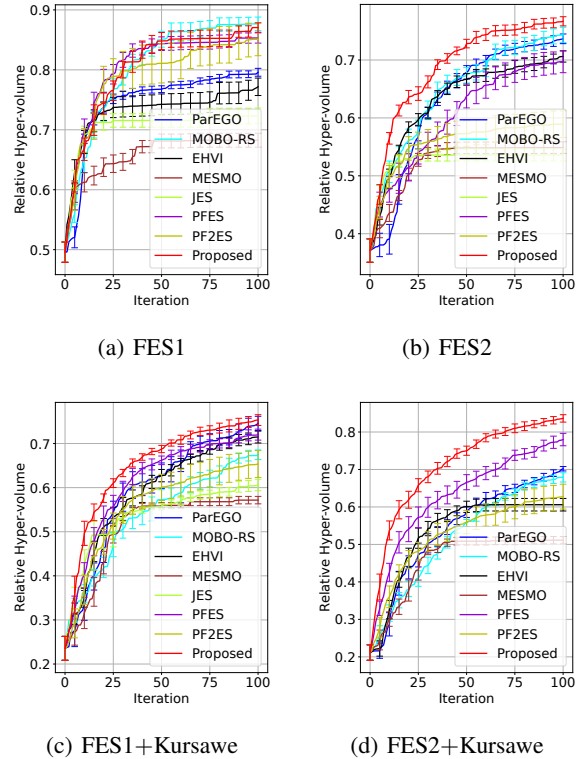

(a) FES1          (b) FES2

(c) FES1+Kursawe     (d) FES2+Kursawe

*Figure 23.* Performance comparison in additional benchmark functions (average and standard deviation of 10 runs).

from $d = 5$ to $d = 10$. All experimental settings except for the input dimension are identical to those in the main text. The results show that our method demonstrates stable optimization performance as the input dimension increases, similar to the cases presented in the main text. Additionally, we evaluated our method on the DTLZ benchmark suite with $(d, L) = (3, 3)$, as shown in Figure 27. On DTLZ2, DTLZ5, DTLZ6, and DTLZ7, our proposed method achieves higher RHV compared to other methods. For DTLZ1 and DTLZ3, all methods show similarly high RHV values (close to 1) with small differences between them. In the case of DTLZ4, EHVI and MESMO demonstrate better performance, while our method performs comparably to the remaining baselines.

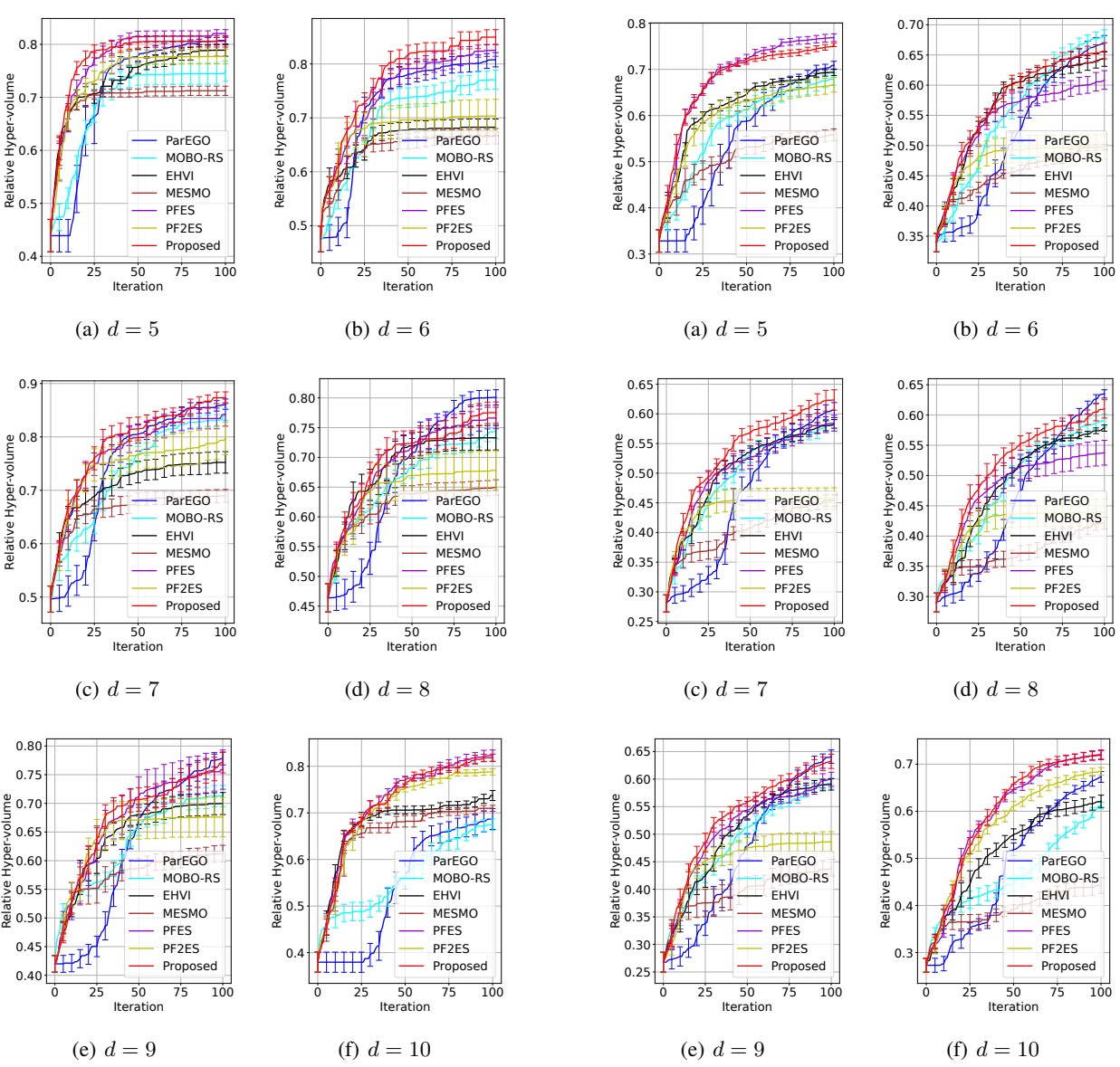

*Figure 24.* Results on FES1.

*Figure 25.* Results on FES2.

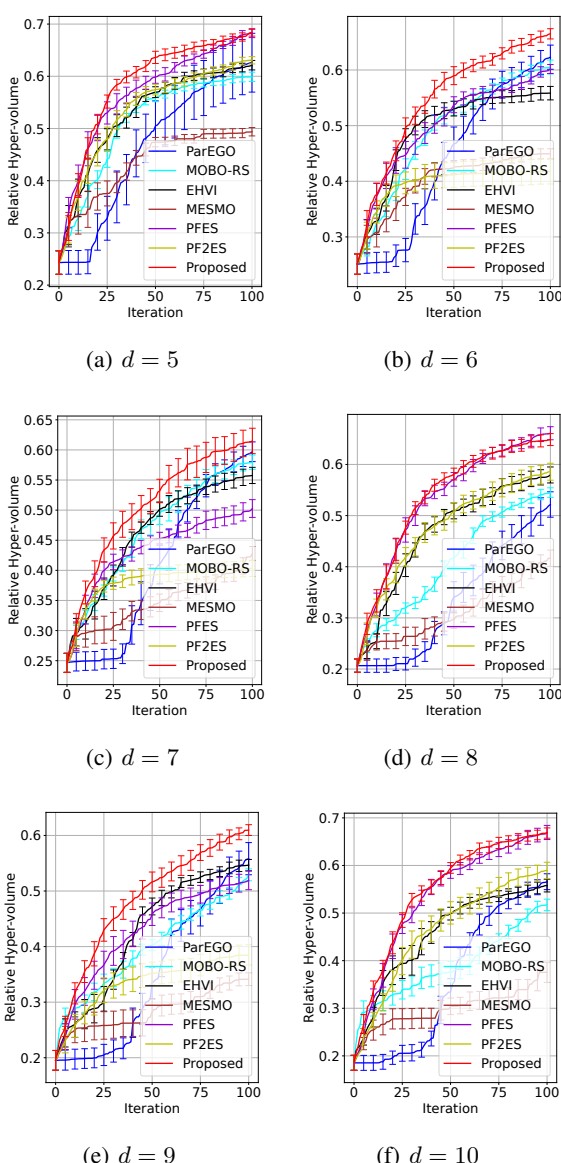

(a) $d = 5$

(b) $d = 6$

(c) $d = 7$

(d) $d = 8$

(e) $d = 9$

(f) $d = 10$

*Figure 26.* Results on FES3.

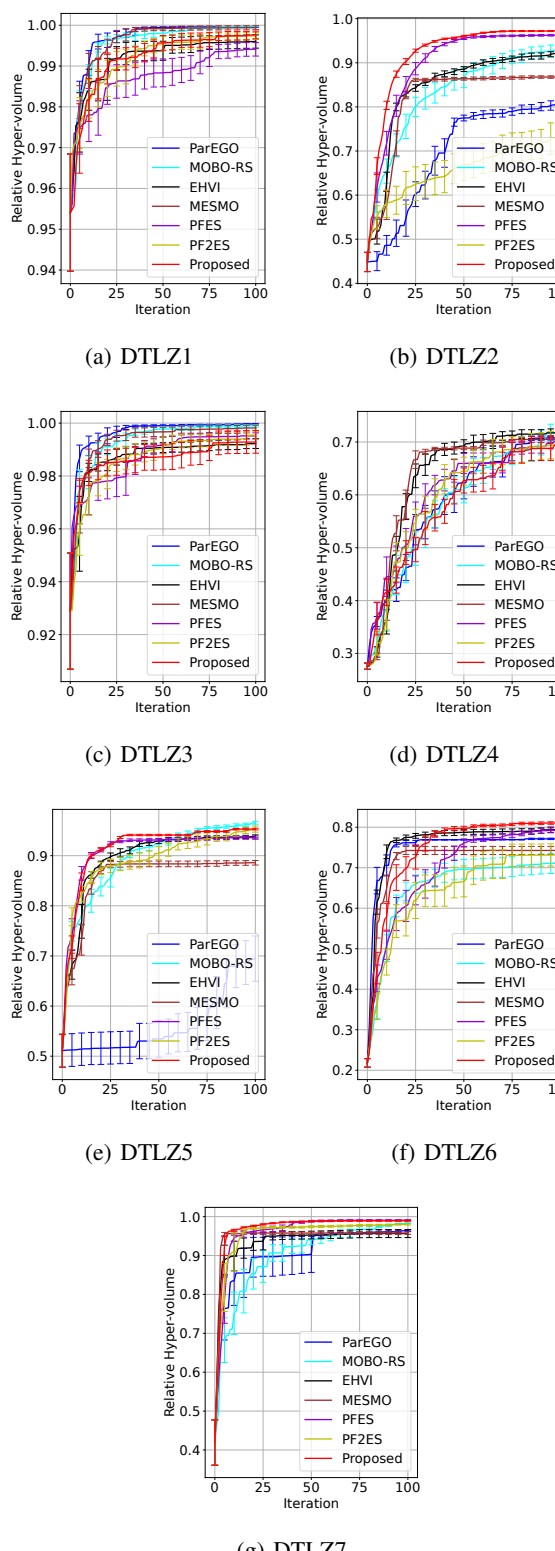

(a) DTLZ1

(b) DTLZ2

(c) DTLZ3

(d) DTLZ4

(e) DTLZ5

(f) DTLZ6

(g) DTLZ7

*Figure 27.* Results on DTLZ ($d = 3, L = 3$).

### K.3. Computational Time

We here examine the computational time of PFEV. We used the GP-derived synthetic function ($d = 3$) from Section 6.1. The training dataset was randomly selected 50 points, and we evaluated the computational time (CPU time) required to calculate the acquisition function values for 100 points generated by the Latin hyper cube sampling. In PFEV, we evaluate time of calculating 11 grid points of $\lambda$ ($10^{-3}, 0.1, 0.2, \ldots, 1.0$) for a given $\boldsymbol{x}$. The results are shown in Table 1, in which we evaluate ParEGO, PFES, EHVI, and PFEV (Proposed).

Obviously, ParEGO was quite fast because it applies the usual single-objective BO to the scalarized value. For $L = 3$ and 4, EHVI was also fast, but it becomes slow at $L = 5$ and we stopped it at $L = 6$ because of the long computational time. The slow computation of EHVI at large $L$ is widely known. PFEV and PFES were similar up to $L = 4$, but for $L = 5$ and $L = 6$, PFEV was slower than PFES. This is because of the computation of QHV. For $L = 3$ and $L = 4$, the computational time of NSGAII was dominant in PFEV (Note that the same procedure is also performed in PFES). However, for $L = 5$, QHV requires the similar cost as NSGAII, and for $L = 6$, the computational cost of QHV became much larger than NSGAII. PFEV requires QHV two times for each sampled Pareto-frontier (see Appendix B) while PFES requires QHV only once for each sampled Pareto-frontier. This was major reason of the difference of PFEV and PFES in $L = 6$.

## L. Difference of Over- and Under-Truncation

Here, we discuss approximation error caused by replacing the dominated region by the entire Pareto-frontier $\mathcal{A}^{\mathcal{F}^*}$ with its counter-part created by $\mathcal{F}_S^*$, i.e., $\mathcal{A}_O^{\mathcal{F}_S^*}$ or $\mathcal{A}_U^{\mathcal{F}_S^*}$. The difference actually can be small if $S$ is sufficiently large, but practically intractable size of $S$ is required even for reasonable size of $L$. To demonstrate this claim, we evaluate differences of hyper-volumes of $\mathcal{A}_O^{\mathcal{F}_S^*}$ and $\mathcal{A}_U^{\mathcal{F}_S^*}$. If the difference of $\mathcal{A}_O^{\mathcal{F}_S^*}$ and $\mathcal{A}_U^{\mathcal{F}_S^*}$ are small, it suggests that the true Pareto-frontier $\mathcal{F}^*$ can be accurately approximated by $\mathcal{F}_S^*$.

In Fig. 28, we empirically investigate the ratio of the volume of under- and over- truncations compared with the true volume. We assume that the Pareto-frontier is the simplex ($\sum_{i \in [L]} f^i = 1$ and $f^i \geq 0$), and approximation points $\mathcal{F}_S^*$ are sampled uniformly in the simplex as shown in Fig. 28 (a). The examples of the volume ratio is shown in Fig. 28 (b). In Fig. 28 (c), we see that, when $L = 2$, the difference between the truncations rapidly decreases, but for $L \geq 3$ shown in Fig. 28 (d) and (e), large differences remain even when the population size is 1,000 (Note that in this example, over-truncation is closer to the true value than under-truncation, but in general, it depends on the shape of the

*Table 1.* CPU time evaluation (sec).

|  |  | $L = 3$ | $L = 4$ | $L = 5$ | $L = 6$ |
|---|---|---|---|---|---|
| ParEGO |  | $0.95 \pm 0.08$ | $1.15 \pm 0.06$ | $1.38 \pm 0.07$ | $1.62 \pm 0.11$ |
| PFES |  | $21.43 \pm 0.20$ | $29.18 \pm 0.39$ | $45.23 \pm 1.41$ | $127.64 \pm 24.44$ |
| EHVI |  | $0.86 \pm 0.06$ | $3.51 \pm 2.24$ | $500.09 \pm 336.18$ | - |
| Proposed | Total | $21.93 \pm 0.42$ | $31.74 \pm 0.64$ | $61.04 \pm 3.26$ | $274.49 \pm 28.99$ |
|  | CALCPFEV | $0.59 \pm 0.05$ | $1.56 \pm 0.13$ | $5.04 \pm 0.49$ | $24.76 \pm 3.71$ |
|  | NSGAII | $19.59 \pm 0.37$ | $25.66 \pm 0.37$ | $31.36 \pm 0.27$ | $37.26 \pm 0.22$ |
|  | RFM | $0.59 \pm 0.00$ | $0.80 \pm 0.01$ | $1.00 \pm 0.00$ | $1.19 \pm 0.00$ |
|  | QHV | $0.42 \pm 0.03$ | $2.72 \pm 0.26$ | $22.41 \pm 2.65$ | $209.81 \pm 25.68$ |

true Pareto frontier which is unknown). The continuous Pareto-frontier is the $(L-1)$-dimensional space, and we conjecture that the required number of points to keep the sufficient density of approximation points $\mathcal{F}_S^*$ increases exponentially when $L$ increases.

About the difference between $\mathcal{F}^*$ and $\mathcal{F}_S^*$, increasing the number of MC samples cannot mitigate the essential problem in principle. In our formulation, the over-truncation corresponds to $\lambda = 0$. Therefore, the over-truncation based lower bound is biased toward a smaller value, because the bound is defined as the maximizer with respect to $\lambda$. This discussion holds even in the population $\mathrm{LB}(\boldsymbol{x}, \lambda)$ (i.e., before introducing the sample approximation), and therefore, increasing the number of samples from the GPs does not solve this intrinsic bias.

## M. Consideration on Gradient-based Optimizer

Figure 29 shows results with qLogNEHVI (Daulton et al., 2020; 2021; Ament et al., 2023) for GP-derived functions ($d = 3, L = 3, 4, 5$) and four benchmark functions. For qLogNEHVI, we used qLogNoisyExpected-HypervolumeImprovement of BoTorch. The reference point is defined by $y_{\min} - 0.1(y_{\max} - y_{\min})$, where $y_{\min}$ and $y_{\max}$ are vectors consisting of the minimum and the maximum of each dimension in training $\boldsymbol{y}_i$, respectively. For qLogNEHVI, we used both the gradient descent (`optimize_acqf` function of BoTorch) and DIRECT for the GP-derived functions. We see that the gradient optimizer slightly improves the results, but DIRECT also has the similar performance. Overall, qLogNEHVI shows good performance, while PFEV also shows comparable performance. Although implementations are not exactly consistent (PFEV is GPy base and qLogNEHVI is BoTorch), we can see that performance in our results is not largely different from the well-known package.

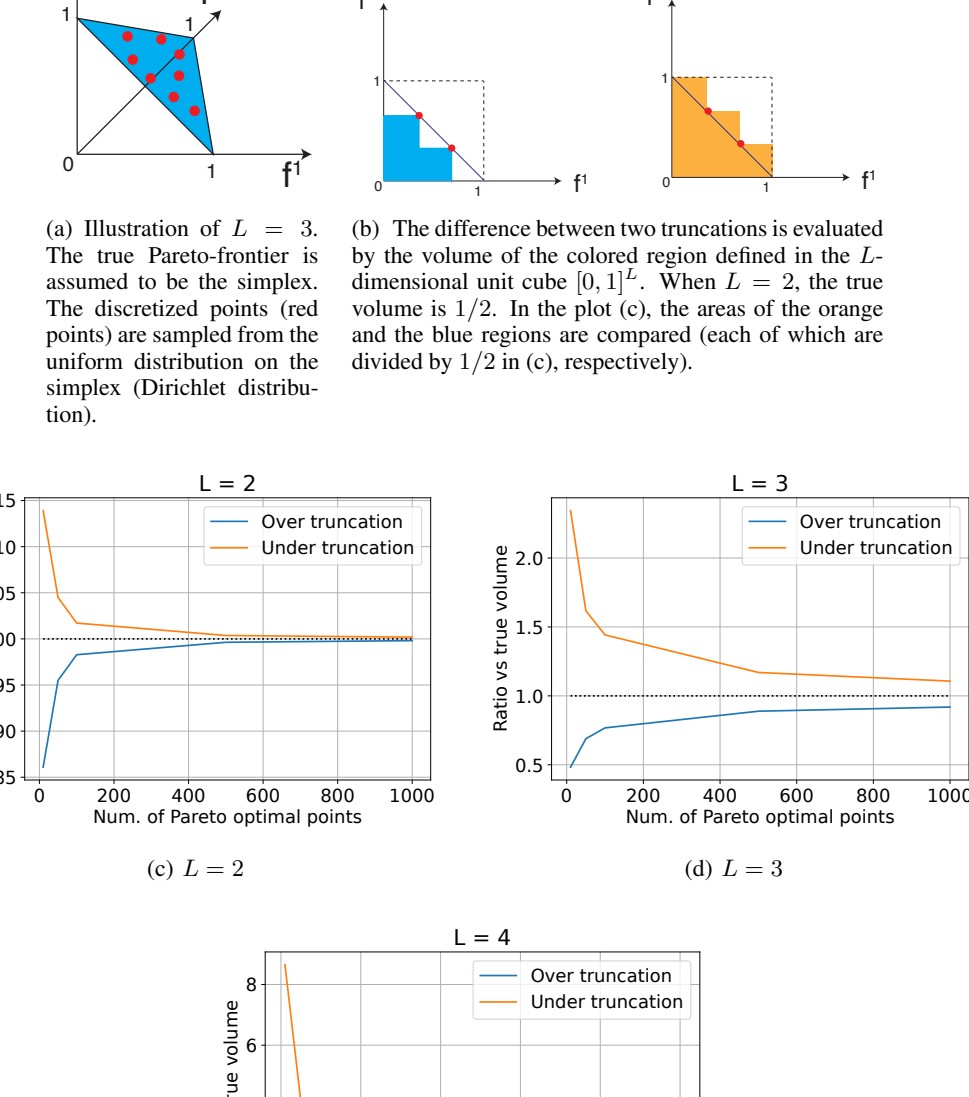

(a) Illustration of $L = 3$. The true Pareto-frontier is assumed to be the simplex. The discretized points (red points) are sampled from the uniform distribution on the simplex (Dirichlet distribution).

(b) The difference between two truncations is evaluated by the volume of the colored region defined in the $L$-dimensional unit cube $[0, 1]^L$. When $L = 2$, the true volume is $1/2$. In the plot (c), the areas of the orange and the blue regions are compared (each of which are divided by $1/2$ in (c), respectively).

(c) $L = 2$

(d) $L = 3$

(e) $L = 4$

*Figure 28.* Hyper-volume relative differences between over- and under- truncations. The volume is defined in the unit cube $[0, 1]^L$. The vertical axis is the ratio compared with the true volume.

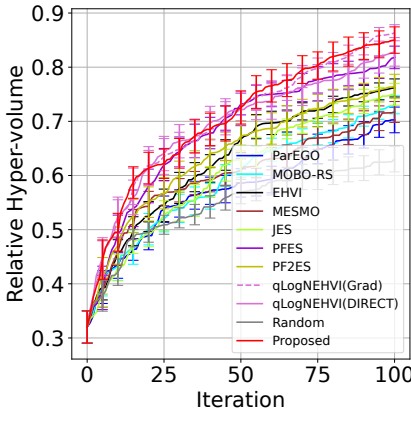

(a) $L = 3$

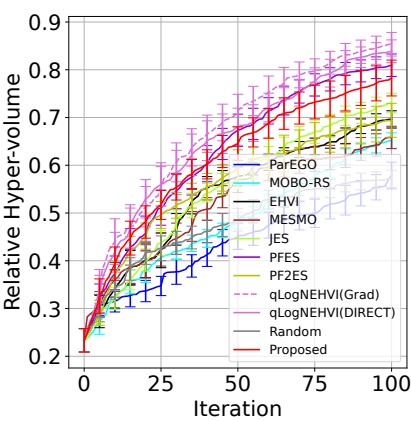

(b) $L = 4$

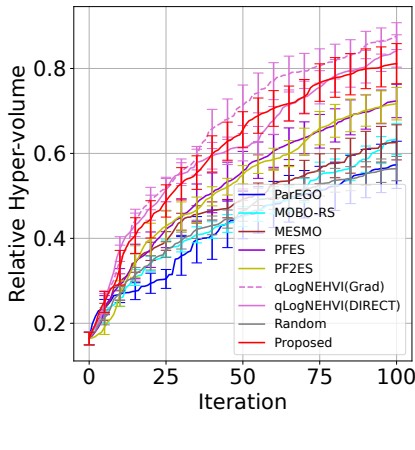

(c) $L = 5$

*Figure 29.* Comparison with qLogNEHVI.

