# OpenReview forum: "Pareto-frontier Entropy Search with Variational Lower Bound Maximization"
_ICML.cc/2025/Conference — ICML 2025 poster_

### Official Review · Reviewer_8xyH · 2025-03-08

**Overall Recommendation:** 4

**Summary:**

This work introduces a multi-objective Bayesian optimization estimation framework that utilizes an information-theoretic acquisition function. It presents an effective approach for conducting variational inference when the continuous Pareto frontier is not fully known. By employing a mixture of distributions with different supports, the method enhances the estimation of the target distribution, demonstrating strong performance across diverse experiments.

**Claims And Evidence:**

Four claims are made by the authors:

>    PFEV is the first approach to continuous space MOBO that is based on a general lower bound of MI

I agree, but this lower bound of MI on MES concept has been proposed in Takeno 2022 in single-fidelity, so I won't consider it as a MAJOR contribution;

>    We newly introduce an under truncation approximation for $p(\boldsymbol{f}(x)\mid\mathcal{F}^\ast). Further, we define the variational distribution as a mixture of distributions with the over and the under truncation, and show how to optimize the mixture weight

The claim is correct. However, I do have some concerns about the lower bound optimization part, which I will elaborate later;

>    We also discuss properties and extensions of PFEV. For our MI lower bound, its relation with PI and a Monte-Carlo approximation are derived. We further discuss several extended settings such as parallel querying.

I agree with the most of the claim, except the part involving PI. Although the authors show that the MI lower bound is an upper bound of PI AF, it does not explain many things. Since for acquisition function we only care finding the x that maximize them, the lower bound does not disclose any intrinsic relation between the optimal x derived from MI lower bound and optimal x derived from PI. Therefore, as I agree the results from Remark 3.2, I won't consider it as a major contribution.

>    Through empirical evaluation on Gaussian process generated and benchmark functions, we demonstrate effectiveness of PFEV. We empirically observed that PFEV shows a particular difference from existing over truncation based methods when output dimension >= 3 , in which the difference of two truncation becomes more apparent.

The authors have clear evidence showing this claim.

**Essential References Not Discussed:**

NaN

**Experimental Designs Or Analyses:**

The experimental designs are standard. I do not see any concerns in this part.

**Methods And Evaluation Criteria:**

Overall, I believe the authors clearly presented their proposed methods and corresponding contributions. The mathematical notations are well-defined, and the experiments used to validate their proposed method follow standard practices.

**Other Comments Or Suggestions:**

NaN

**Other Strengths And Weaknesses:**

Strengths:

- The work proposes a new way to solve multi-objective BO in the variational inference perspective;

- The work is solid, with thorough discussion on previous works and well-defined mathematical notations;

- The experiments are solid, showing clear advantages of their proposed method.

Weaknesses:

- The well-definedness of the optimization problem in Equation (4);

- Lack of motivations for the approximation schemes in Section 3.3.

**Questions For Authors:**

My concerns are listed in the weaknesses section. There is no other questions raised.

**Relation To Broader Scientific Literature:**

This work extends previous research on solving MES with variational inference to a multi-objective setting. It introduces a novel approach to advancing information-theoretic multi-objective Bayesian optimization, making a meaningful contribution to the community.

**Theoretical Claims:**

The main concern is about the design for maximizing the lower bound $\max_{x, \lambda\in(0, 1]} L(x, \lambda)$. The author did not show this is a well-defined optimization setting (optimum exists). What if the maximum happens at $\lambda = 0$, in the sense that $p = q_O$?

Some minor suggestions:

- Line 121: "If $q(f_x\mid\mathcal{F}^\ast) = p(f_x\mid\mathcal{F}^\ast)$", should add "for $\mathcal{F}^\ast$ almost everywhere".

- Equation (4): $L$ has been used for the number of objectives, please use another notation for the lower bound

- Many approximation schemes are proposed between line 237 to line 265. It is a bit hard to follow their motivations. Could you explain more about them?

---

> ### Author Rebuttal · Authors · 2025-04-01
>
> Thank you for your constructive comments and suggestions.
>
> 1) About $\lambda$: We can prove $\lambda = 0$ cannot be the maximizer, and thus, the optimization problem is well-defined. This can be derived from (8). First, from the definition, we see $\theta_{MAP} \in (0,1)$ (Note that $\hat{p} \in (0,1)$). When $\lambda \to 0$, the second term of (8) goes to $\log 0$ (from the definition of $\eta_{\lambda}^{\tilde{{\cal F}}_S^*}$ shown after (4)). On the other hand, the first term is finite when $\lambda \to 0$. Therefore, if $\lambda \to 0$, we see (8) goes to $- \infty$. Further, we already show that (8) is concave in Appendix C.
>
> 2) Motivation of the lower bound approximation: The lower bound approximation can be written as (7). However, in (7),
> $p(\boldsymbol{f_x} \in {\cal A}_O^{\tilde{{\cal F}}^*_S} \mid \tilde{{\cal F}}^*)$
> and
>
> $p(\boldsymbol{f_x} \in {\cal A}_{U \setminus O}^{\tilde{{\cal F}}^*_S} \mid \tilde{{\cal F}}^*)$
> cannot be calculated analytically. A naive approximation of these two $p$ are
> $I( \tilde{\boldsymbol{f_x}} \in {\cal A}_O^{\tilde{{\cal F}}^*_S} )$
> and
>
> $I( \tilde{\boldsymbol{f_ x}} \in {\cal A}_{U \setminus O}^{\tilde{{\cal F}}^*_S} )$,
> respectively (line 245-246), which we used in the estimator called naive MC in Figure 6. Instead of this direct one sample estimation, we propose using approximation with the prior knowledge about
> $p(\boldsymbol{f_x} \in {\cal A}_O^{\tilde{{\cal F}}^*_S} \mid \tilde{{\cal F}}^*)$,
> i.e., this can be approximated by $\hat{p}$.
>
> We will also reflect other comments such as notations and claims about PI.

---

> > ### Comment · Reviewer_8xyH · 2025-04-03
> >
> > Thank you for your comments. We should have $\hat{p} \in [0, 1]$, right? In that case, the $\log 0$ term vanishes (which is equivalent to $q = q_O$). In this scenario, $\lambda = 0$ is the optimal solution. Can this method handle this situation?

---

> > > ### Author Response · Authors · 2025-04-03
> > >
> > > Thank you for your comment.
> > >
> > > We actually have $\hat{p} \in (0,1)$ (not $[0,1]$), because of its definition $\hat{p} = Z_O^{\tilde{{\cal F}}^*_S}(\boldsymbol x) / Z_U^{\tilde{{\cal F}}^*_S}(\boldsymbol x)$.
> > >
> > > Since $\boldsymbol{f_x}$ is the Gaussian distribution (the predictive distribution of the GPs), from the definitions,
> > > $Z_O^{\tilde{{\cal F}}^*_S}(\boldsymbol x) = p(\boldsymbol{f_x} \in {\cal A}_O^{\tilde{{\cal F}}^*_S}) \in (0,1)$
> > > and
> > > $Z_O^{\tilde{{\cal F}}^*_S}(\boldsymbol x) < Z_U^{\tilde{{\cal F}}^*_S}(\boldsymbol x) = p(\boldsymbol{f_x} \in {\cal A}_U^{\tilde{{\cal F}^*_S}}) \in (0,1)$.
> > >
> > > Here, we used
> > > $\emptyset \neq {\cal A}_O^{\tilde{{\cal F}}^*_S} \subset {\cal A}_U^{\tilde{{\cal F}}^*_S} \subset \mathbb{R}^L$
> > > (neither ${\cal A}_O^{\tilde{{\cal F}}^*_S}$ nor ${\cal A}_U^{\tilde{{\cal F}}^*_S}$ is the empty set or the entire output space,
> > > and
> > > ${\cal A}_O^{\tilde{{\cal F}}^*_S} \subset {\cal A}_U^{\tilde{{\cal F}}^*_S}$
> > > holds as far as the sampled Pareto frontier set $\tilde{{\cal F}}_S^*$ is finite while the true Pareto frontier is continuous which is our problem setting).

---

### Official Review · Reviewer_QnYc · 2025-03-12

**Overall Recommendation:** 2

**Summary:**

This paper considers a method for multi-objective Bayesian optimization based on entropy search. Many existing entropy search methods rely on a discrete approximation of the Pareto frontier (which may be continuous in many cases), which typically leads to over-truncation of the predictive distribution. The paper highlights this problem and proposes an improvement that leverages a mixture of two distributions that over/under-truncate, respectively. The paper proposes a variational lower bound, which can be optimized to jointly select the new candidate point to evaluate and the mixture parameter, and the paper gives two ways of approximating the lower bound with different bias/variance trade-offs. The authors evaluate the approach on synthetic problems

**Claims And Evidence:**

Many of the claims are well-supported. However, there is a lack of evidence in a few areas:

- For simplicity consider the case where only one sample from the GP is used (as opposed to using 10 in the paper). The issue of over-truncation issue will be most problematic when the approximate PF has very few points. This is easily avoidable but simply increasing the population size for NSGA-II. This would result in far less truncation/discretization error. Is the over-truncation problem still an issue if one uses a finer approximation of the PF (more points from NSGA-II). I could see there being a case made for this increasing computation cost (of NSGA-II and of the box decomposition), but no such argument is made and no evidence is provided to that end (and there are box decomposition algorithms that often are quite a bit faster (Lacour et al, 2017).

- Is over-truncation still an issue if one uses more samples from the GP? If more samples are used then the acquisition function is integrated over different samples of the (discretized) PF. It is not clear from the paper whether over-truncation is only an issue if a small number of samples are used. Of course, even better performance would be achieve if more points on the PF were used in the discretization and more GP samples were used.

Results on how much over-truncation matters when the number of points used to approximate the PF increases and number of GP samples increases seem critical to understanding the significance of this problem

**Essential References Not Discussed:**

Unexpected Improvements to Expected Improvement for Bayesian Optimization, Ament et al, 2023
Differentiable Expected Hypervolume Improvement for Parallel Multi-Objective Bayesian Optimization, Daulton et al, 2020

**Experimental Designs Or Analyses:**

The main issues with the empirical analysis is 1) lack of hard problems (higher dimensional and more realistic), 2) lack of baselines. At a minimum, random search and qLogEHVI (Ament et al, 2023) should be included as baselines.

In addition, why is DIRECT used to optimize the acquisition functions? Many of these are differentiable and gradient-based optimization is typically far more effective (Daulton et al, 2020) and is widely used in modern Bayesian optimization. 5 random starting points also likely leads to poor global optimization of the acquisition functions.

How are reference points chosen for evaluation and for EHVI? That should be documented as it likely has a large effect on the performance evaluation (i.e. how different parts of the Pareto frontier contribute to the overall hypervolume).

Which estimation of the lower bound is used in the experiments in the main text?

**Methods And Evaluation Criteria:**

The synthetic problems are a good start, but they are all low dimensional and they are all synthetic problems. It would be nice to see performance on more realistic examples and higher dimensional problems.

**Other Comments Or Suggestions:**

In the abstract:
- truncated -> truncated
- truncation -> truncations

Figure 2 is confusing, and I don't understand what is being depicted.

**Other Strengths And Weaknesses:**

These are described above. The main issue is understanding the significance of the over truncation issue and understanding how the contribution of the work compares to the SOTA. Also, no code is provided and it appears the authors implemented all baselines by hand if they are in GPy? Using the implementations by the authors of many of the baselines in BoTorch would instill more confidence in the results.

**Questions For Authors:**

Questions are interspersed above.

**Relation To Broader Scientific Literature:**

This builds on work on information theoretic methods for multi-objective BO.

**Theoretical Claims:**

The derivations looked reasonable.

---

> ### Author Rebuttal · Authors · 2025-04-01
>
> Thank you for your constructive comments.
>
> 1) Population size of NSGA-II: As the reviewer mentioned, if the population size of NSGA-II increases, the approximation error between over truncation and the true truncation decreases. On the other hand, the continuous Pareto-frontier is the $L-1$-dimensional space, for which the required number of points to keep the sufficient density of approximation points increases exponentially when $L$ increases. In the linked figure ([link](https://anonymous.4open.science/r/AuthorResponse-0652/Fig28-hv.pdf)), we empirically investigate the truncation error by comparing the volume of under- and over- truncations. In this figure, we assume that the Pareto-frontier is the simplex ($\sum_{i \in [L]} f^i = 1$ and $f^i \geq 0$), and approximation points are sampled uniformly in the simplex. We see that, when $L = 2$, the difference rapidly decreases, but for $L \geq 3$, the large differences remain even when population size is $1,000$. Note that in this example, over truncation is closer to the true value than under truncation, but in general, it depends on the shape of the true Pareto frontier which is unknown.
>
> 2) Truncation error and the sampling from GP: In our formulation, the over-truncation corresponds to $\lambda = 0$. Therefore, the over truncation based lower bound is biased toward smaller value, because the bound is defined as the maximizer with respect to $\lambda$ (Further, we prove that $\lambda = 0$ never becomes the maximizer in the response #1 of reviewer 8xyH). This discussion holds even in the population $L(\boldsymbol x, \lambda)$ (i.e., before introducing the sample approximation), and therefore, increasing the number of sampling from the GPs does not solve the intrinsic bias.
>
> 2) Realistic problem setting: The linked figure ([link](https://anonymous.4open.science/r/AuthorResponse-0652/Fig29-HPO.pdf)) shows the results on hyper-parameter optimization problems of a machine learning algorithm. These experiments were performed before the review available (so, baselines are same as the first submission). The problem setting is the class weight optimization for multi-class classification. We used LightGBM as a base model. We only focus on the optimization of the class weight parameters and thus the input dimension $d$ of BO is equal to the class size. The objective function is the test classification accuracy of each class (i.e., $L$ is also equal to the class size). Overall, the results on the four datasets indicate that PFEV has sufficient performance compared with other methods.
>
> 3) Higher input dimension: See response #3 of reviewer SDSC.
>
> 4) qLogNEHVI and Random: The linked figure ([link](https://anonymous.4open.science/r/AuthorResponse-0652/Fig30-31-qLogNEHVI.pdf)) shows results with qLogNEHVI and Random for GP-derived functions ($d = 3, L = 3,4,5$) and four benchmark functions. For qLogNEHVI, we used qLogNoisyExpectedHypervolumeImprovement of BoTorch. The reference point is defined by y_min - 0.1 (y_max - y_min), where y_min and y_max are vectors consisting of the minimum and the maximum of each dimension in training $\boldsymbol{y}_i$, respectively. For qLogNEHVI, we used both the gradient descent (optimize_acqf function of BoTorch) and DIRECT for the GP-derived functions. We see that the gradient optimizer slightly improves the results, but DIRECT also has the similar performance. Overall, qLogNEHVI shows good performance, while PFEV also show comparable performance (particularly when comparing with the DIRECT version of qLogNEHVI). Although implementations are not exactly consistent (PFEV is GPy base and qLogNEHVI is BoTorch), we can see that performance in our results are not largely different from the well-known package.
>
> 5) About DIRECT: See response #4 of review SDSC. Note that `5 random $\boldsymbol x$' in the end of the second paragraph of Section 6 is about the initial points of BO (not the initial points of the acquisition function optimization). Sorry for confusing.
>
> 6) Reference point of EHVI: In EHVI, the two reference points are required, shown as v_ref and w_ref in Shah and Ghahramani 2016. The worst point vector v_ref is defined by subtracting $10^{-4}$ from the vector consisting of the minimum value of each dimension of $\boldsymbol y_i$ in the training data. On the other hand, the ideal point vector $w_ref$ is defined by adding $1$ to the vector consisting of the maximum value of each dimension of $\boldsymbol{y}_i$. We will add explanation in the paper.
>
> 7) Lower bound estimator: Throughout the paper, the lower bound estimator (8) is used, except for Section 6.3 in which the performance of two estimators (8) and (6) were compared.
>
> 8) Implementation of HVKG: See response #1 of reviewer SDSC.
>
> 9) Fig2 illustrates examples of under and over truncation in a three dimensional output space. We tried to show there exist large differences in the case of $L \geq 3$.
>
> We will also reflect other comments such as typos and references.

---

### Official Review · Reviewer_w3eW · 2025-03-14

**Overall Recommendation:** 3

**Summary:**

This paper introduces Pareto-frontier Entropy Search with Variational Lower Bound Maximization (PFEV), an acquisition function for multi-objective Bayesian optimization (MOBO). It addresses a key limitation in prior information-theoretic MOBO methods, which rely on over-truncation when approximating the predictive distribution hence leading to inaccurate approximations. Instead, PFEV models this distribution as a mixture of over- and under-truncated approximations and optimizes the mixture weight variationally to minimize approximation error. Empirical results show that PFEV outperforms existing methods, particularly in high-dimensional output spaces ($L ≥ 3$), with superior relative hyper-volume performance on both Gaussian process-simulated and benchmark functions.

**Claims And Evidence:**

- The paper claims the previous approximation is crude

**Essential References Not Discussed:**

All the essential references seem to have been discussed.

**Experimental Designs Or Analyses:**

- The synthetic validation on GP samples and more complicated real problems make sense to me.
- One slightly unconventional thing is the choice of synthetic function to optimize which seems not that common in MOBO, I would be interested in seeing its performance on at least one of the following commonly used benchmarks in contemporary literature:
    - BraninCurrin
    - DTLZ series

**Methods And Evaluation Criteria:**

- The mixtured approach makes sense to me

**Other Comments Or Suggestions:**

the paper in general is very well written, I do not have much further comments.

One only additional comment is it can be helpful to plot the evolvement of $\lambda$ w.r.t BO iteration

**Other Strengths And Weaknesses:**

Strength
- The approach of learning a Pareto frontier truncated conditional distribution is interesting and, to my best of knowledge, novel.
- An extensive problem setting (decoupled setting, noisy observation, parallel query) has been discussed.

Weakness
- I think one issue is that the mixtured approximation, though could improve the original approximation, can still be too crude as there is only one parameter, in practice, it could happen is some region is overestimated, some region is underestimated, but this is not able to be faithfully characterized by this metric, hence its practical utility is not exactly clear.
- While the approach makes sense theoretically, the practical empirical advantage is slightly puzzling me, as hypervolume decomposition is known to be complicated with high objective numbers, which means all of these information theoretical acquisition function, which relies on hypervolume decomposition, ideally is not expected to be leveraged there.

**Questions For Authors:**

- How the learned $\lambda$ evolve over BO iteration, a plot is crucial to validate
- What exactly is line 14-16 used for? it seems the BO has stopped at line 13 already.
- Why $\lambda$ cnnot reach $0$ in parameterization?

**Relation To Broader Scientific Literature:**

The paper proposes a more accurate truncated Pareto frontier distribution approximation, which could be of interest in information-theoretic acquisition functions for Multi-Objective Bayesian Optimization community.

**Theoretical Claims:**

- I have checked Appendix A,B,C they all seems plausible to me.

---

> ### Author Rebuttal · Authors · 2025-04-01
>
> Thank you for your constructive comments.
>
> 1) About benchmark function: The results on DTLZ is shown [here](https://anonymous.4open.science/r/AuthorResponse-0652/Fig26-DTLZ.pdf). Note that these were performed before the review opening. We see that, in DTLZ 2, 5, 6, and 7, the proposed method shows high RHV among compared methods. On the other hand, in DTLZ 1 and 3, all the methods show high RHV (close to 1), and differences among methods are small. In DTLZ4, EHVI and MESMO showed good performance, while the other methods were similar performance.
>
> 2) Mixture approximation: We employ one parameter ($\lambda$) model, because this parameter should be estimated by the small number of samples (i.e., $K$, for which we used 10). Using more complicated model is possible (e.g., defining $\lambda$ as a function of $\boldsymbol x$), but there is a risk of the over-fitting to the small samples. We think that investigating the performance of more complicated variational distribution is one of important directions.
>
> 3) Hyper-volume decomposition: In current our experimental setting, the hyper-volume is exactly calculated by a divide-and-conquer based approach (QHV). Therefore, even if the output dimension increases, the accuracy of the decomposition is maintained. On the other hand, the cell-based decomposition can take longer time for a high dimensional output space (e.g., $L \geq 7$). Then, introducing approximate decompositions is a possible approach. The cell decomposition is performed for calculating cumulative distribution function (CDF) of the predictive distribution. We plan to consider a pruning strategy that stops the divide-and-conquer procedure if the density $p(\boldsymbol f(\boldsymbol x))$ is sufficiently small in the (intermediately) decomposed region.
>
> 4) Transition of $\lambda$: The linked figure ([link](https://anonymous.4open.science/r/AuthorResponse-0652/Fig27-lambda-transition.pdf)) shows the transition of $\lambda$ during BO iterations (the $10$ runs average) for a GP-derived synthetic function ($d = 3, L = 4$). We see that $\lambda$ takes intermediate values in $(0,1]$, indicating that under- and over- truncated distributions are indeed mixed during BO iterations. As far as we examined so far, any increasing or decreasing tendency have not been observed during iterations.
>
> 5) Line14-16: The line 14-16 in Algorithm 1 is the definition of the function `CalcPFEV', which calculates the acquisition function value. This function is used in line 10 of Algorithm 1.
>
> 6) $\lambda$ cannot reach 0: When $\lambda = 0$, the support condition of the variational distribution (2) is not satisfied, which is required to guarantee that $L(\boldsymbol{x})$ is a lower bound.

---

### Official Review · Reviewer_SDSC · 2025-03-15

**Overall Recommendation:** 4

**Summary:**

The paper considers an information-theoretic acquisition function -- namely Pareto Frontier Entropy Search -- for multi-objective Bayesian Optimization. The authors propose a novel "variational" approximation to the predictive distribution given the Pareto frontier based on over and under truncation of the Pareto set. This results in an acquisition function that requires jointly optimizing over the (one-dimensional) variational parameter and the query location(s). The authors describe the implementation and computational details and discuss useful extensions. Finally, the paper demonstrates the empirical performance of the proposed approach in a number of benchmarks against common baselines.

### Update after rebuttal.
My overall assessment of the paper remains the same. The authors' response to my review clarify some questions but it also highlights that relying on a gradient-free optimization method for the acquisition function optimization has its limits. I don't think this should make or break the paper if it's sufficiently discussed in an updated version (the additional results generated for the rebuttal should help with that); ideally providing a continuous relaxation of the indicator as discussed in the authors' response. I think this is a solid paper and
think it should be accepted, but would not revolt if the majority of other reviewers feels differently.

**Claims And Evidence:**

Yes, claims are clear and demonstrated based on convincing evidence.

**Essential References Not Discussed:**

AFAICT the paper discusses all of the essential references in this space as far as the methodology is concerned. But some of the comparisons with the EHVI-based methods do not use state-of-the-art implementations though as discussed in [A,B,C].

**Experimental Designs Or Analyses:**

* Overall the experiments and conclusions drawn are solid.
* The plots in the MT are all **extremely** hard to read. Please consider moving some of the technical details in the MT into the appendix to make room for more readable plots. You may also want to select a few representative results and relegate the rest of the empirical results to the appendix.
* In a similar vein as above, the comparison to HVKG in Appendix F seems potentially problematic due to the simplifications in the implementation: "we evaluate the posterior mean hyper-
volume after adding sampled fl(x) into the GPs only by using the pre-defined grid points in X (uniformly taken 5 points in each dimension is used"). This feels like comparing an optimized implementation of PFEV against a simplified implementation of HVKG. A comparison against the implementation in [A] would be more meaningful.

[A] S. Daulton, M. Balandat, and E. Bakshy. Hypervolume knowledge gradient: A lookahead approach for multi-objective Bayesian optimization with partial information. In  International Conference on
Machine Learning, 2023.

**Methods And Evaluation Criteria:**

Yes, the proposed method is sound and the evaluation is clear and comprehensive (modulo the discussion of higher-dimensional settings, see below)

**Other Comments Or Suggestions:**

* "Note that if VDC is not satisfied, L(x) is not defined because of log 0" <- "not defined because of log 0" is a rather odd statement; this not being defined is simply b/c p would not be absolutely continuous w.r.t. q.
* "number of samplings" -> number of samples

**Other Strengths And Weaknesses:**

**Strengths**
* The paper is well-written and clearly identifies the key contributions.
    * For instance, I really liked the use of Figure 1, this helps a lot with visualizing the over- and under-truncation and makes it a lot easier to follow the paper.
* The authors provide a comprehensive review of related work.
* The empirical evaluation setup is solid, and the baselines considered are meaningful - however, some do not use state-of-the-art implementations (see other comments)
* The performance of the proposed approach especially on problems with many outcomes is demonstrated clearly (albeit only for low input dimensions).
* The paper has very comprehensive supplementary material that goes into detail on many relevant extensions / variants of the main contribution (parallel, decoupled, joint, noisy).

**Weaknesses**
* The primary weakness of the approach is that it uses gradient-free optimization of the acquisition function (in this case the DIRECT algorithm). These methods are known to perform very poorly in higher dimensions, which presumably is the reason why the largest evaluated input dimension is d=3 (d=4 in the supplementary material). This weakness / limitation of the approach is currently not discussed at all in the paper.
    * I would like to see some ablations of PFEV on higher-dimensional inputs compared to approaches that use gradient-based optimization of the acquisition function such as qEHVI/qNEHVI from [B]/[C] (assuming that EHVI as evaluated in the paper also used DIRECT).

[B] S. Daulton, M. Balandat, and E. Bakshy. Differentiable expected hypervolume improvement for parallel multi-objective bayesian optimization. In Advances in Neural Information Processing Systems, volume 33, 2020.

[C] S. Daulton, M. Balandat, and E. Bakshy. Parallel bayesian optimization of multiple noisy objectives with expected hypervolume improvement. In Advances in Neural Information Processing Systems, volume 34, 2021.

**Questions For Authors:**

* Your approach uses a "global" mixture distribution for the PF truncation. Is it possible / would it be advisable to do have this mixture depend on the location (in the outcome space)? I'm thinking of settings in which over-truncation is more accurate in one part and under-truncation is more accurate in another part of the outcome space.
* Positivity of PFES has not been clarified - what does this mean? Is this about the formulation or about the approximation / algorithm?
* The results in Fig 3. show a pattern where PFEV and MOBO-RS perform quite a bit better than PFES in d=2, but the difference is less for d=3. Is this b/c PFEV is more challenging to optimize as the dimension increases?

**Relation To Broader Scientific Literature:**

* The key contribution is a novel variational approximation of the predictive distribution given the Pareto frontier.
* This formulation is quite elegant and effectively addresses an issue with the approximation that becomes more and more problematic as the number of outcomes increases, an issue that the paper identifies and explains nicely.

**Theoretical Claims:**

Yes, I checked the main claims in the MT and spot-checked some of the details in the appendix.

---

> ### Author Rebuttal · Authors · 2025-04-01
>
> Thank you for your constructive comments.
>
> 1) HVKG in Appendix F: The linked figure ([link](https://anonymous.4open.science/r/AuthorResponse-0652/Fig20-Decoupled.pdf)) shows a comparison with BoTorch HVKG (called qHypervolumeKnowledgeGradient, which is based on the well-known 'one shot' optimization of KG) in the decoupled setting (Appendix F). Because of the long computational time, we set the number of Pareto optimal points in the one-step ahead posterior mean as 10 in HVKG. For fair comparison, the proposed method also set the size of the Pareto optimal points in the sampled function (i.e., the NSGA-II population size) as 10. The number of samplings (so called fantasy points) in HVKG is also set as $10$. Although there is the implementation difference (the proposed method is the GPy based implementation), we consider the result shows the performance of the proposed method is sufficient compared with an existing package.
>
> 2) qEHVI: We performed evaluation with qLogNEHVI. See response #4 of reviewer QnYc.
>
> 3) Higher input dimension: For FES1, 2 and 3 benchmark functions, results on the $10$ dimensional input are shown in Figure 19 of Appendix K.2. We further show the results on $d = 5, \ldots, 9$ of the same benchmark functions [here](https://anonymous.4open.science/r/AuthorResponse-0652/FES.pdf), which were performed before the review becomes open. The results indicate that the proposed method stably performs optimization similar to the cases of the main text. Evaluating more higher input problems on realistic scenarios is one of our future work.
>
> 4) About DIRECT: We employed DIRECT because it does not require `initial points' unlike the gradient descent that requires the appropriate setting of the initial points (the number of initial points and locations). Our purpose is to focus more on differences of the acquisition functions and to reduce the other factors effecting the performance. On the other hand, evaluation using gradient-based approaches is also important future work because it is widely used in BO. We partially show comparison with a baseline using the gradient optimization (qLogNEHVI) in response #4 of reviewer QnYc.
>
> 5) About mixture: As the reviewer suggested, it is possible to use a mixture depending on the location (e.g., defining $\lambda$ as a function of $\boldsymbol x$, which can be estimated by the variational lower bound maximization). We employ the simple global mixture only with one parameter because it should be estimated based on the K samples, which is usually quite small ($10$ in the experiments). Introducing more complicated variational distribution is a possible future direction.
>
> 6) Positivity of PFES: PFES calculates an approximate MI through a decomposition as a difference of the entropy. On the other hand, Takeno 2022 et al. show possible negativity of the naive application of MES to the constraint BO, which has a similar formulation to PFES (the entropy difference based MI decomposition and truncation of a multi-dimensional predictive distribution by a box shaped region). This suggests that PFES also might have a risk of having a negative value, but positivity of PFES has not been proven so far.
>
> 7) Relation of the input dimension and performance: As the reviewer suggested, there is a tendency for the differences between methods to decrease as the input dimension increases. We confirmed this tendency by the boxplot ([link](https://anonymous.4open.science/r/AuthorResponse-0652/Fig24-25-boxplot_input_dim.pdf)), created by the same way as Figure 4. According to the figure of the first page, differences of all the methods gradually decrease (not only the proposed method). In the second page, we further plot separately for each length scale setting of the true objective function. We see that, particularly for small length scale problems (which tends to be a multi-modal function), differences become small. We speculate that the differences were less apparent for challenging problems with high-dimension and highly multi-modal functions.
>
> We will also reflect other comments such as plotting and references.

---

> > ### Comment · Reviewer_SDSC · 2025-04-04
> >
> > Thanks for the response and for some of the additional results, especially on the higher dimensional problems and the gradient-based acquisition function optimization - however, ideally there would be some results on the comparison of gradient-based and gradient-free acquisition function optimization for higher-dimensional problems, since this is where we'd expect the biggest difference (the provided results have d=3 as their highest dimension).
> >
> > While I agree that DIRECT does seem to perform quite reasonably, there is a sizable difference between the performance of the DIRECT- and the gradient-based version of qLogNEHVI in the [provided comparison](https://anonymous.4open.science/r/AuthorResponse-0652/Fig30-31-qLogNEHVI.pdf). Also, why is the gradient-based version missing from the results in Figure 31? Also, what computational budgets were used for both approaches / what was the wall time for the optimization in each case?
> >
> > > Our purpose is to focus more on differences of the acquisition functions and to reduce the other factors effecting the performance.
> >
> > While I am sympathetic to this argument, I also think that the performance of the acquisition function cannot be fully decoupled from the way this acquisition function can be optimized. If it does not provide any gradients, and gradient-based optimization can provide significant performance improvements (see above), then that is a limitation of the method.
> >
> > Overall, I continue to believe that this is a good paper around a nice idea (though not an excellent paper) - I will keep my score.

---

> > > ### Author Response · Authors · 2025-04-09
> > >
> > > Thank you for your additional comments and suggestions.
> > >
> > > > why is the gradient-based version missing from the results in Figure 31?
> > >
> > > This is simply because of the time limitation of the author response period (we originally used a GPy-based implementation, but have implemented it from scratch using BoTorch). This time, we added qLogNEHVI with the gradient descent to FES3 with $d = 3$, and both the gradient descent and DIRECT to FES3 with $d = 10$ (FES3 was the only benchmark where the input scale is $[0,1]^d$ among the four benchmarks in Fig. 31, so it could be run first without requiring any implementation adjustments of the scale, as $[0,1]^d$ is the default of optimize_acqf in BoTorch. We will perform the others as well). We updated [Fig. 31](https://anonymous.4open.science/r/AuthorResponse-0652/Fig30-31-qLogNEHVI.pdf) (note that since some results had initial points inconsistency of qLogNEHVI in the previous Fig. 31, we corrected it though the change does not affect our discussion). In $d = 10$ (Fig. 31 (e)), DIRECT was slightly better than the gradient based counter-part of qLogNEHVI. However, we agree with that, in general, the gradient method is better for higher dimensional problems. We do not particularly focus on high input-dimensional problems, and our current interpretation is that DIRECT reasonably worked for the moderate input dimension problems that we performed in the paper.
> > >
> > > We measured the wall-clock time in the GP-derived synthetic data $(d = 3, L = 4)$, and the acquisition function optimization took 6.3 and 31.5 sec for the gradient descent and DIRECT in qLogNEHVI, respectively (The time was the 10 time average when the GPs have randomly selected $50$ points as a training dataset). The settings of optimize_acqf of BoTorch was num_restarts=10 and maxiter=200 (from settings of [qEHVI tutorial](https://botorch.org/docs/tutorials/multi_objective_bo/)) and DIRECT employed the scipy default 1000 max iterations (Note that DIRECT does not have multi-restarts). We conjecture that the gradient method was faster because of the early termination by the gradient convergence.
> > >
> > > Applying a gradient method to the proposed method is an important future work. The proposed acquisition function (8) is mostly differentiable, except for the indicator $I(\boldsymbol{\tilde{f_x}} \in {\cal A}_O^{\tilde{{\cal F}}^*_S})$
> > >
> > > in $\theta_{MAP}(\boldsymbol{\tilde{f_x}})$ of (8).
> > > We consider that possible approaches are simply ignoring this term in the gradient (regarding the gradient of the indicator as 0) or using a continuously approximated gradient. In the continuous approximation of the gradient, we replace
> > > $I(\boldsymbol{\tilde{f_x}} \in {\cal A}_O^{\tilde{{\cal F}}^*_S}) \approx p(\boldsymbol{f} \in {\cal A}_O^{\tilde{{\cal F}}^*_S})$,
> > > where
> > > $\boldsymbol{f} \sim N(\boldsymbol{\tilde{f}_x}, \rho \boldsymbol{I})$ in which $\rho > 0$ is a fixed smoothing parameter.
> > > The right hand side of the approximation is differentiable with respect to $\boldsymbol{x}$ through the similar decomposition to (9) in Appendix B (note that $\boldsymbol{\tilde{f}_x}$ is differential because it is generated from RFM). This can be interpreted as a counter-part of the standard CDF based smoothing approximation of an indicator function, extended to the Pareto dominated region. For the calculation, although the cell-based decomposition is required for ${\cal A}_O^{\tilde{{\cal F}}^*_S}$, we can reuse the cells created in line 7 of Algorithm 1.
> > >
> > > We will include all the discussion so far somewhere in the main paper or the appendix. Once again, thank you for the constructive feedback.

---

### Decision · Program_Chairs · 2025-05-01

**Decision:**

Accept (poster)

**Comment:**

In this paper, the authors study multi-objective Bayesian optimization (MOBO) through the information gain of the Pareto-frontier. To calculate the information gain, a predictive distribution conditioned on the Pareto-frontier is needed. This distribution is a distribution truncated by the Pareto-frontier. However, it is usually impossible to obtain the entire Pareto-frontier in a continuous domain. The authors  consider an approximation of the truncated distribution by using a mixture distribution consisting of two possible approximate truncation obtainable from a subset of the Pareto-frontier. Since the optimal balance of the mixture is unknown beforehand, the authors propose optimizing the balancing coefficient through the variational lower bound maximization framework. The empirical evaluation demonstrates the effectiveness of the proposed method mostly when the number of objective functions is large. The reviewers have indicated the novelty of the proposed approach and the completeness of the experiments considered. They also highlight the thorough discussion on previous works and well-defined mathematical notations. They also say that the experiments are solid, showing clear advantages of their proposed method. The reviewers have also indicated small weaknesses of the proposed method related to the small dimensionality of the experiments and some lack of baselines that the authors should try to address. Overall, I think that this is not an outstanding paper. However, it will receive the attention of the community working on Bayesian optimization.